# Carbon fiber reinforced composite material layup design optimization method and its application in automobile battery box

Dongzhen Lu[1,2], Shuai Zhang[1,2]*, Zhao Li[3], Jiufeng Chen[4], Feng Xiong[5], Yuzhuo Zhang[1], Jinlong Xiao[6]

1 College of Vehicle and Traffic Engineering, Henan University of Science and Technology, Luo yang,China, 2 State Key Laboratory of Structural Analysis, Optimization and CAE Software for Industrial Equipment, Zhengzhou University, Zhengzhou,China, 3 School of Art and Design, Henan University of Science and Technology, Luoyang,China, 4 Ningbo Tuopu Group Co.,Ltd, Zhejiang,China, 5 Key Laboratory of Advanced Manufacturing Technology for Automobile Parts, Ministry of Education, Chongqing University of Technology, Chongqing,China, 6 College of Vehicle Engineering, Henan Industry and Trade Vocational College, Zhengzhou, China

* boreke@126.com

## Abstract

In order to improve the lightweight level of new energy vehicle battery box assembly and its design efficiency and precision, a design scheme of sheet molding compound (SMC) composite box upper plate and carbon fiber composite lower box was proposed. Based on digital twin technology and combined with an improved particle swarm-bacterial foraging (PSO-BFO) algorithm, a fiber reinforced composite layup design and multi-level twin optimization method was proposed. The SMC composite material upper panel was designed and optimized by using the morphology optimization and size optimization methods. The carbon fiber composite material lower panel layup sequence was designed and optimized by using the constructed digital twin multi-level optimization method. Comparing the performance of the battery box assembly before and after optimization, the improvement rates of battery box assembly quality and first-order modal frequency were 40.63% and 33.34% respectively. The stress improvement rates under bumpy conditions, bumpy+braking conditions/bumping+turning conditions are 83.60%; 81.01%; 84.31% respectively, and the displacement improvement rates are 62.80%, 63.21%, and 63.14% respectively. Lightweight and performance improvement The effect is remarkable. The effectiveness of the battery box assembly design scheme and carbon fiber reinforced composite material layup design optimization method proposed in this paper is demonstrated.

## 1. Introduction

Automotive Lightweight technology can significantly improve energy efficiency, reduce emissions, and extend the range of electric vehicles [1–2]. The battery box

**Data availability statement:** All relevant data are within the paper and its Supporting Information files.

**Funding:** This work is supported by the Open Fund Project of State Key Laboratory of Structural Analysis, Optimization and CAE Software for Industrial Equipment (grant number GZ2024A03-ZZU); the National Key Laboratory of Land and Air Based Information Perception and Control, China (grant number B324009); Science and Technology Research Project of Henan Province(grant number 242102241055); the Industry-University-Research Collaborative Innovation Base Project on Au-tomobile Lightweight of "Science and Technology Innovation in Central Plains"(grant number 2024KCZY315);The Project Supported by National Natural Science Foundation of China (grant number 52302408); the National Natural Science Foundation of China (grant number 52202437)

**Competing interests:** The authors have declared that no competing interests exist.

is the energy storage center of electric vehicles, and as the supporting member of the automotive power battery, the stiffness and strength of the battery box should be considered in the design. Against the backdrop of hindered battery technology development, lightweight design of the battery box can significantly increase the specific energy of the EV powertrain, thereby maximizing the EV range. On the premise of meeting various requirements, designing and optimizing a composite battery box assembly can effectively improve the lightweight level and safety performance of the battery box, which has important scientific significance and engineering prospects.

Automotive lightweight technology can be specifically divided into structural optimization design, advanced manufacturing processes, and lightweight material technology. Scholars have achieved many research results in these areas. Material lightweighting is achieved by using lower density and higher strength materials such as aluminum alloy, magnesium alloy, and carbon fiber reinforced composite materials to replace traditional steel, thereby reducing the overall weight. Among the many high-strength, lightweight materials, carbon fiber composites are an important automotive material for reducing vehicle weight, improving performance, and reducing energy consumption due to their excellent properties and broad application prospects. Ma et al. replaced the metal material with carbon fiber composite and optimized the composite design of the battery box, ultimately reducing the weight by 56% [3]. Jiang et al. replaced the steel control arm of the car suspension with a carbon fiber composite material, and proposed a multi-stage optimization strategy to optimize the design of the carbon fiber composite control arm. The optimization results showed that its mass was reduced by 38.34% and all other indicators were significantly improved [4]. Zhang et al. designed a new hybrid material B-pillar assembly by replacing the car's B-pillar reinforcement plate with a carbon fiber composite. The final optimization results showed a weight reduction of 26.44% and a reduction in peak crash force of 43.7% [5]. He et al. proposed a multi-scale optimization method for long carbon fiber nonwoven composites and used the car hood for lightweight design, which reduced the weight by 37% while improving the stiffness and strength [6]. Lv et al. developed a smart body for the elderly made of carbon fiber composites. By combining experiments with simulation methods, they designed a smart car body that meets the lightweight requirements [7]. Sheet molding compound (SMC) is a thermoset composite material typically consisting of chopped glass fibers, resin, fillers and additives. It is widely used in the automotive industry, typically for automotive parts with moderate performance requirements. Li et al. studied the influence of the fiber factor on the properties of sheet molding compound (SMC) to achieve Lightweight and performance improvement of SMC composites. The results showed that the flexural stiffness of SMC composites increased by 147%, and the impact strength increased by 27% [8]. Ding et al. developed and designed SMC composite parts, studied the influence of load distribution on SMC composite parts, so as to achieve maximum weight reduction, obtain a reasonable structure, and improve the efficiency of structural design, providing guidance for the development and application of SMC composite parts [9].

Lightweight design is based on material selection and further weight reduction by optimizing the shape, size, layout, etc. of the body and other structural components. It focuses not only on the selection of materials, but also on how to make the car structure meet both the strength requirements and keep the weight to a minimum through reasonable design. For example, the use of honeycomb structures, thinning designs, topology optimization, etc. Wang et al. achieved a 25.74% weight reduction in an automotive bumper system by combining structural optimization design with material changes [10]. Zhang et al. proposed a combined design method for wheel joint topology optimization to provide a new idea for wheel Lightweight [11]. Dong et al. optimized the structure of the car battery box to a frame structure, and the results showed that the weight of the battery box was reduced by 450 kg, which is a significant Lightweight effect [12].Du et al. used topology optimization to lighten the front rails of the car. The optimization results showed that the front rails of the car were reduced by 6.5%, the peak acceleration in a collision was significantly reduced, and the energy absorbed in a collision was increased [13]. Yu et al. applied the tailor rolled blank (TRB) structure to the front-end components of an all-electric vehicle to improve its lightweight and crashworthiness. A multi-objective optimization algorithm was used for the optimal design. The optimization results showed that the lightweight effect was significant while the crashworthiness was improved [14].Wu et al. comprehensively described the performance of basalt/GRP rebars in harsh environments. Comprehensive analysis showed that the durability of basalt/GRP rebars in alkaline environments could be further improved by improving the types of fiber materials and resin matrix [15]. Justin et al. studied the mechanical and durability properties of fiber-reinforced concrete in acidic environments. The results showed that concrete containing glass fiber and silica powder exhibited excellent mechanical properties and durability in acidic environments, which is a material with broad application prospects in harsh environments [16].

Process lightweight is the further reduction of weight by optimizing production processes after the material and structural design has been determined. In terms of Lightweight processes, Zhang et al. used a method of segmental heating and cooling to hot-stamp parts [17]. Klinke et al. discuss the current shortcomings of TRB and propose a strategy method for optimal part selection based on TRB technology [18]. Hu et al. proposed a roll forming technology for carbon fiber reinforced metal hybrid materials, and the results showed that the specific energy absorption was improved by more than 45.6%, which had a significant Lightweight effect [19]. Zhou et al. studied the process characteristics of lightweight metal-composite structures for automobiles, analyzed the advantages and disadvantages of the current laser welding forming technology, summarized the forming quality of laser welded joints and the methods of regulation, and provided a reference and guidance for the development of dissimilar material joining technology [20].

The battery box, as the carrier of the battery system of an electric vehicle, is responsible for protecting the battery system. The design of the battery box must take into account the performance requirements under different operating conditions. With the rapid development of electric vehicles, range has become a key concern, and lightweight design of the battery pack structure is one of the key technologies to solve the problem. The Lightweight of the battery box of an electric vehicle is mainly achieved by optimizing the structural design and selecting suitable materials. Pan et al. proposed a new strategy for evaluating the crashworthiness of automotive battery cases by investigating how the thickness and material of the battery case affect the crashworthiness of the vehicle. The optimization results show that this strategy can reduce the weight of the battery case while improving its crashworthiness [21]. Kulkarni et al. replaced the aluminum alloy battery box of a car with a carbon fiber composite material and explored the factors affecting the performance of the battery box under crash conditions [22]. Pan et al. proposed a method to optimize the dimensions of high-strength steel battery cases. Simulations and experiments have verified that the weight gain of the battery case is 10.41%, while ensuring the improvement of other performance indicators [23]. Zhao et al. proposed a new type of BF/PLA composite battery case by replacing the original battery case material with BF/PLA composite material. The optimization results show that it can reduce the weight by 40.88%, and all performance indicators meet the requirements [24]. Wang et al. used an improved multi-objective particle swarm optimization algorithm based on numerical simulation and analysis methods to optimize the design of automotive battery boxes. The results showed that the Lightweight of the battery box was ensured while its crashworthiness was improved [25].

Traditional optimization methods are mostly applicable to linear or simple systems, but they have many limitations when dealing with multi-objective optimization problems. Such methods are often time-consuming and complex when dealing with multi-objective optimization, and may ignore the global or near-optimal solution within the feasible domain. With the rapid development of various optimization techniques in engineering applications, intelligent optimization algorithms are becoming increasingly popular for solving multi-objective optimization problems. Zhao et al. proposed a battery box with an outer layer of high-strength steel and an intermediate layer of aluminum alloy honeycomb. A particle swarm algorithm was used to obtain the optimal thickness coefficient of the honeycomb cells, and the final result was a 37.26% reduction in the mass of the optimized battery box [26]. Liu et al. proposed a reliability-optimized design method to solve the Lightweight problem of carbon fiber composite battery cases, and used an improved particle swarm optimization algorithm and surrogate modeling technology to achieve a 22.14% weight reduction in composite materials [27]. The bacterial foraging optimization algorithm is a new intelligent algorithm with strong search capabilities and strong robustness, but it is prone to local optima and has low optimization accuracy [28–30]. To solve this problem, researchers have tried to develop new optimization strategies. For example, the use of hybrid algorithms can solve this problem, and the use of hybrid algorithms can combine accuracy and efficiency to solve multi-objective optimization problems. Zhang et al. proposed a carbon fiber composite laminate design method using a PSO-BFO hybrid algorithm to achieve comprehensive optimization of materials, processes, and structures, which has a significant Lightweight effect [31]. Wang et al. proposed a strategy that combines an improved particle swarm optimization algorithm with principal component analysis, and applied it to the optimization design of the outer panel of the car's B-pillar. The results showed that the weight of the B-pillar was reduced by 20.7%, and this optimization strategy has superior performance [32]. Wang et al. proposed an optimization strategy that combines a particle swarm optimization (PSO) algorithm with a bacterial foraging optimization (BFO) algorithm. The results show that this hybrid algorithm optimization strategy is of great significance for guiding lightweight automotive design [33]. Raju et al. used a hybrid particle swarm-bacterial foraging optimization to perform an optimization design of fused deposition modeling (FDM) process parameters to find the best process parameters [34].Kontoni et al. combined the adaptive neuro-fuzzy inference system with the particle swarm optimization algorithm by using the existing comprehensive test library to predict the maximum axial strain and peak axial stress of circular confined concrete reinforced with fiber-reinforced polymers. The final test results showed that the established model has a good effect in predicting the final axial strain and peak axial stress of confined concrete [35].Ahmadi et al. evaluated the limitations of the design specifications for composite columns and proposed a simplified relationship for the nominal compressive bearing capacity of a new type of square high-strength steel tube concrete short column. The results were compared and this research result can be applied to the design of square high-strength steel tube concrete components [36].

As a cutting-edge technology in the era of industry 4.0, digital twin simulation can realize real-time interaction between virtual models and actual physical entities, and has the unique advantages of timely data collection and closed-loop optimization. In terms of simulation optimization, digital twins help gain a deeper understanding of the impact of different solutions on structural performance by adjusting designs and parameters in a virtual environment to select the most cost-effective and optimal design. Simulation and optimization using digital twins can significantly reduce the number of physical tests, thereby reducing development and maintenance costs and improving the efficiency of engineering project implementation [37–38]. Xie et al. proposed a design framework based on digital twin driving and applied it to the design of elevator fairings. The results of optimization showed that both drag and lateral force were reduced by 18.1% and 11.2%, respectively [39]. Feng et al. proposed a gear health management method driven by digital twins, which effectively reveals the wear propagation characteristics of gears and can accurately predict the rated speed. This method is of great practical significance [40]. Dong et al. used aluminum plates of different thicknesses as research objects, monitored the welding process with the help of digital twins, and tested the parameters after processing to demonstrate the feasibility of the proposed method [41]. Choi et al. proposed a new honeycomb shape and a digital twin-based design framework for the detailed design of semi-symmetric NS curved beams. By using an artificial intelligence model, the error rate is

                                                                    

reduced and the energy absorption capability is improved while ensuring the convergence of the virtual model [42]. Wang et al. applied digital twin technology to offshore wind turbines (OWTs) to achieve real-time monitoring and fault diagnosis of structural members, providing a promising application for reliability analysis of light rail support structures [43].Jia et al. proposed an assembly operation-driven digital twin model synchronization framework based on multicolor set theory for product assembly accuracy prediction [44].In order to save the company's operation and maintenance costs, Sun et al. proposed a remote operation and maintenance method for electromechanical equipment based on digital twin technology. By establishing a digital twin remote operation and maintenance service platform, they reduced labor costs and improved the company's service efficiency [45].Song et al. used digital twin technology to complete the construction and testing of the milling process system, realized data-driven intelligent parameter optimization, and verified the effectiveness of various functions of the system through experiments [46].

By combing through the above literature, the researchers have studied and analyzed the performance of automotive battery box assemblies from different angles, including structural optimization, high-strength and lightweight materials, and optimization algorithms, but have ignored manufacturing costs and universality. The lightweight design of the automotive battery box assembly requires comprehensive consideration of factors such as safety, structural optimization, material application and cost control.

In terms of safety, the design and manufacture of the automotive battery box assembly requires comprehensive consideration of multiple factors to ensure the safe driving of the vehicle under various operating conditions. In terms of structural optimization, factors such as the ply angle, shape, number of layers and sequence of carbon fiber composite materials have an important influence on structural performance and strength. At the same time, facing the problem of multi-variable optimization, the convergence and optimization efficiency of the optimization algorithm need to be improved, and the quantitative ranking of the optimized compromise solutions will help to comprehensively balance the performance requirements of all parties. In terms of cost control, it is difficult for an automotive battery box assembly made of a single lightweight material to have both performance and cost advantages. In response to this situation, this paper proposes a hybrid particle swarm optimization and bacterial foraging optimization strategy (PSO-BFO) based on mathematical twin technology, which can significantly improve the lightweight optimization effect of automotive carbon fiber composite battery boxes and achieve an integrated lightweight design of structure, materials and performance. However, no relevant methods have been applied to the lightweight design of hybrid material automotive battery box assemblies.

Therefore, this paper proposes a novel optimization design method for composite battery cases to address the above issues. Based on structural optimization, material substitution and experimental testing, Digital Twinning technology is used in combination with a hybrid particle swarm optimization and bacterial foraging optimization strategy to improve the global optimization capability and convergence speed of composite material battery cases in the lightweight design process. A real-time simulation and monitoring system is used to provide more physical feedback during the PSO-BFO optimization process provides more physical feedback while jointly optimizing key indicators such as strength, stiffness, and manufacturing cost under multi-objective comprehensive constraints, ensuring that in the pursuit of Lightweight, structural safety and material performance are not compromised, and that an integrated lightweight design of the structure, materials, and performance of the carbon fiber hybrid material battery box can be achieved. This provides an entirely new solution for advanced lightweight design with broad application prospects.

## 2. Methods

### 2.1 Basics of single-layer plate mechanics

Composite materials generally exhibit directionally dependent physical properties, meaning that their mechanical behavior varies depending on the direction of the material. In other words, composites respond differently to forces applied in different directions, exhibiting anisotropic properties:

$$\left\{\begin{array}{c} \sigma_1 \\ \sigma_2 \\ \sigma_3 \\ \tau_{12} \\ \tau_{23} \\ \tau_{13} \end{array}\right\} = \left[\begin{array}{cccccc} C_{11} & C_{12} & C_{13} & C_{14} & C_{15} & C_{16} \\ C_{21} & C_{22} & C_{23} & C_{24} & C_{25} & C_{26} \\ C_{31} & C_{32} & C_{33} & C_{34} & C_{35} & C_{36} \\ C_{41} & C_{42} & C_{43} & C_{44} & C_{45} & C_{46} \\ C_{51} & C_{52} & C_{53} & C_{54} & C_{55} & C_{56} \\ C_{61} & C_{62} & C_{63} & C_{64} & C_{65} & C_{66} \end{array}\right] \left\{\begin{array}{c} \varepsilon_1 \\ \varepsilon_2 \\ \varepsilon_3 \\ \gamma_{12} \\ \gamma_{23} \\ \gamma_{13} \end{array}\right\}$$

(1)

In engineering applications, the internal structure of materials is often designed to be symmetrical to simplify the analysis and manufacturing process. Orthotropic materials are a special type of anisotropic material that have consistent physical and mechanical properties in three mutually perpendicular planes of symmetry. The carbon fiber composite lower box studied in this article has the basic properties of this type of material. Specific relationship:

$$\left\{\begin{array}{c} \sigma_1 \\ \sigma_2 \\ \sigma_3 \\ \tau_{12} \\ \tau_{23} \\ \tau_{13} \end{array}\right\} = \left[\begin{array}{cccccc} C_{11} & C_{12} & C_{13} & 0 & 0 & 0 \\ C_{21} & C_{22} & C_{23} & 0 & 0 & 0 \\ C_{31} & C_{32} & C_{33} & 0 & 0 & 0 \\ 0 & 0 & 0 & C_{44} & 0 & 0 \\ 0 & 0 & 0 & 0 & C_{55} & 0 \\ 0 & 0 & 0 & 0 & 0 & C_{66} \end{array}\right] \left\{\begin{array}{c} \varepsilon_1 \\ \varepsilon_2 \\ \varepsilon_3 \\ \gamma_{12} \\ \gamma_{23} \\ \gamma_{13} \end{array}\right\}$$

(2)

Laminated boards are made by stacking several single-layer boards in a certain order, so the properties of the laminated board depend on the properties of the single-layer boards. When analyzing a single-layer board, it is generally assumed that the direction of thickness is as indicated by number 3 in Fig 1.

In light of the aforementioned research, the thickness of the single-layer board and the longitudinal and transverse dimensions are found to be relatively small. Furthermore, it is determined that the board is in a plane stress state.

$$\sigma_3 = \tau_{23} = \tau_{31}$$

(3)

The relationship between stress and strain can be expressed as follows:

$$\left\{\begin{array}{c} \varepsilon_1 \\ \varepsilon_2 \\ \gamma_{12} \end{array}\right\} = \left[\begin{array}{ccc} S_{11} & S_{12} & 0 \\ S_{12} & S_{22} & 0 \\ 0 & 0 & S_{66} \end{array}\right] \left\{\begin{array}{c} \sigma_1 \\ \sigma_2 \\ \tau_{12} \end{array}\right\} = [S]\{\sigma\}$$

(4)

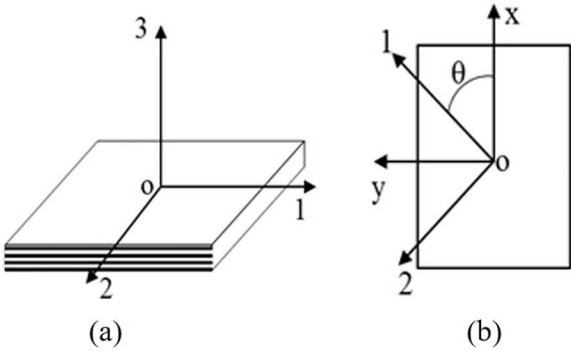

(a)  (b)

**Fig 1. Schematic diagram of single-layer composite material coordinates.** (a) A single-layer coordinate system. (b) Positive rotation of the shaft is observed.

In this context, $[S]$ represents the flexibility matrix. $S_{11} = \frac{1}{E_1}, S_{12} = \frac{\mu_{21}}{E_1} = \frac{\mu_{12}}{E_2}, S_{22} = \frac{1}{E_2}, S_{66} = \frac{1}{G_{12}}$
Equation 4 should be written as a stress-strain relationship:

$$\left\{ \begin{array}{c} \sigma_1 \\ \sigma_2 \\ \tau_{12} \end{array} \right\} = \left[ \begin{array}{ccc} Q_{11} & Q_{12} & 0 \\ Q_{12} & Q_{22} & 0 \\ 0 & 0 & Q_{66} \end{array} \right] \left\{ \begin{array}{c} \varepsilon_1 \\ \varepsilon_2 \\ \gamma_{12} \end{array} \right\} = [Q]\{\varepsilon\}$$

(5)

In this context, the 2D stiffness matrix is represented by the variable $[Q]$ while the variable $[S]$ is obtained through inversion. The variable $Q_{ij}$ is defined as follows:

$$Q_{11} = \frac{E_1}{1 - \mu_{12}\mu_{21}}, Q_{22} = \frac{E_2}{1 - \mu_{12}\mu_{21}}, Q_{33} = G_{12}, \frac{\mu_{12}E_1}{1 - \mu_{12}\mu_{21}} = \frac{\mu_{21}E_2}{1 - \mu_{12}\mu_{21}}$$

(6)

The preceding discussion pertains to the stress-strain relationship of an orthotropic unidirectional material in its principal axis. Given that the x-y coordinate system employed in actual laminated boards differs from the standard system, it is necessary to adjust the coordinate system direction accordingly. As illustrated in Fig 1b, the stress-strain relationship of a unidirectional plate can be expressed as follows:

$$\left\{ \begin{array}{c} \varepsilon_1 \\ \varepsilon_2 \\ \gamma_{12} \end{array} \right\} = \left[ \begin{array}{ccc} \cos^2\theta & \sin^2\theta & \sin\theta\cos\theta \\ \sin^2\theta & \cos^2\theta & -\sin\theta\cos\theta \\ -2\sin\theta\cos\theta & 2\sin\theta\cos\theta & \cos^2\theta - \sin^2\theta \end{array} \right] \left\{ \begin{array}{c} \varepsilon_x \\ \varepsilon_y \\ \gamma_{xy} \end{array} \right\}$$

(7)

Modify the above formula as follows:

$$[T_\sigma] = \left[ \begin{array}{ccc} \cos^2\theta & \sin^2\theta & 2\sin\theta\cos\theta \\ \sin^2\theta & \cos^2\theta & -2\sin\theta\cos\theta \\ -\sin\theta\cos\theta & \sin\theta\cos\theta & \cos^2\theta - \sin^2\theta \end{array} \right]$$

$$[T_\varepsilon] = \left[ \begin{array}{ccc} \cos^2\theta & \sin^2\theta & \sin\theta\cos\theta \\ \sin^2\theta & \cos^2\theta & -\sin\theta\cos\theta \\ -2\sin\theta\cos\theta & 2\sin\theta\cos\theta & \cos^2\theta - \sin^2\theta \end{array} \right]$$

Accordingly, equations 6 and 7 can be transformed into the following:

$$\left\{ \begin{array}{c} \varepsilon_x \\ \varepsilon_y \\ \gamma_{xy} \end{array} \right\} = [T_\varepsilon]^{-1} \left\{ \begin{array}{c} \varepsilon_1 \\ \varepsilon_2 \\ \gamma_{12} \end{array} \right\} \text{ and } \left\{ \begin{array}{c} \sigma_x \\ \sigma_y \\ \tau_{xy} \end{array} \right\} = [\overline{Q}] \left\{ \begin{array}{c} \varepsilon_x \\ \varepsilon_y \\ \gamma_{xy} \end{array} \right\} = \left[ \begin{array}{ccc} \overline{Q}_{11} & \overline{Q}_{12} & \overline{Q}_{13} \\ \overline{Q}_{21} & \overline{Q}_{22} & \overline{Q}_{23} \\ \overline{Q}_{31} & \overline{Q}_{32} & \overline{Q}_{33} \end{array} \right] \left\{ \begin{array}{c} \varepsilon_x \\ \varepsilon_y \\ \gamma_{xy} \end{array} \right\}$$

(8)

In this context, $[T_\sigma]^{-1}$ and $[T_\varepsilon]^{-1}$ represent the inverse matrices of $[T_\sigma]$ and $[T_\varepsilon]$ respectively. The combination of equations 5 and 8 yields the following result:

$$\left\{ \begin{array}{c} \sigma_x \\ \sigma_y \\ \tau_{xy} \end{array} \right\} = [T]^{-1} \left\{ \begin{array}{c} \sigma_1 \\ \sigma_2 \\ \tau_{12} \end{array} \right\} = [T_\sigma]^{-1} [Q] [T_\varepsilon] \left\{ \begin{array}{c} \varepsilon_x \\ \varepsilon_y \\ \gamma_{xy} \end{array} \right\}$$

(9)

The off-axis stiffness matrix is given by the following equation:

$$\left[\overline{Q}\right]=[T_\sigma]^{-1}\,[Q]\,[T_\varepsilon]$$

(10)

The relationship between stress and strain in the x and y coordinate systems can be expressed as follows:

$$\left\{\begin{array}{c}\sigma_x\\\sigma_y\\\tau_{xy}\end{array}\right\}=\left[\overline{Q}\right]\left\{\begin{array}{c}\varepsilon_x\\\varepsilon_y\\\gamma_{xy}\end{array}\right\}=\left[\begin{array}{ccc}\overline{Q}_{11}&\overline{Q}_{12}&\overline{Q}_{13}\\\overline{Q}_{21}&\overline{Q}_{22}&\overline{Q}_{23}\\\overline{Q}_{31}&\overline{Q}_{32}&\overline{Q}_{33}\end{array}\right]\left\{\begin{array}{c}\varepsilon_x\\\varepsilon_y\\\gamma_{xy}\end{array}\right\}$$

(11)

## 2.2 The theory of classical laminates

The limited mechanical properties of single-layer boards allow for the demonstration of the advantages of multi-layer boards through specific combinations of requirements. In light of the intricate nature of the problem, classical laminate theory postulates a number of fundamental assumptions. These include the presence of robust interlayer adhesion, a minimal thickness for the bonding layer, which enables the laminate to behave as a monolithic entity, and the fulfillment of the criterion of a thin sheet. Additionally, the overall thickness is required to be uniform.

As illustrated in Fig 2, the plywood is subjected to surface internal forces, including tension, pressure, and shear, as well as internal moments, such as bending moments and torsion. For the purposes of this discussion, let us assume that the thickness of the plywood is. Integration over the thickness of the plywood yields the magnitude of each internal force and internal moment. Therefore:

$$\left\{\begin{array}{c}N_x\\N_y\\N_{xy}\end{array}\right\}=\int_{-t/2}^{t/2}\left\{\begin{array}{c}\sigma_x\\\sigma_y\\\tau_{xy}\end{array}\right\}dz,\left\{\begin{array}{c}M_x\\M_y\\M_{xy}\end{array}\right\}=\int_{-t/2}^{t/2}\left\{\begin{array}{c}\sigma_x\\\sigma_y\\\tau_{xy}\end{array}\right\}zdz$$

(12)

Equation 12 illustrates the in-plane internal forces and moments of a continuously anisotropic material. Given that laminated boards are composed of multiple discrete components, their stress characteristics lack continuity. In order to obtain the internal forces and moments, it is necessary to integrate the single-layer board and then sum the values of each layer, as demonstrated by the following equation:

$$\left\{\begin{array}{c}N_x\\N_y\\N_{xy}\end{array}\right\}=\sum_{k=1}^{n}\int_{t_{k-1}}^{t_k}\left\{\begin{array}{c}\sigma_x\\\sigma_y\\\tau_{xy}\end{array}\right\}_k dz,\left\{\begin{array}{c}M_x\\M_y\\M_{xy}\end{array}\right\}=\sum_{k=1}^{n}\int_{z_{k-1}}^{z_k}\left\{\begin{array}{c}\sigma_x\\\sigma_y\\\tau_{xy}\end{array}\right\}_k zdz$$

(13)

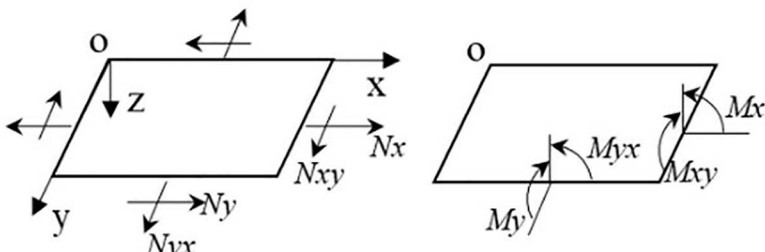

**Fig 2. Schematic diagram of the internal forces and internal moments acting on the laminate.**

The relationship between internal forces and internal moments can be expressed as follows:

$$\left\{ \begin{array}{c} N_x \\ N_y \\ N_{xy} \end{array} \right\} = \sum_{k=1}^{n} \left[ \begin{array}{ccc} \overline{Q}_{11} & \overline{Q}_{12} & \overline{Q}_{13} \\ \overline{Q}_{21} & \overline{Q}_{22} & \overline{Q}_{23} \\ \overline{Q}_{31} & \overline{Q}_{32} & \overline{Q}_{33} \end{array} \right] \times \left\{ \int_{t_{k-1}}^{t_k} \left[ \begin{array}{c} \varepsilon_x^0 \\ \varepsilon_y^0 \\ \gamma_{xy} \end{array} \right] dz + \int_{t_{k-1}}^{t_k} \left[ \begin{array}{c} k_x \\ k_y \\ k_{xy} \end{array} \right] zdz \right\}$$

(14)

$$\left\{ \begin{array}{c} M_x \\ M_y \\ M_{xy} \end{array} \right\} = \sum_{k=1}^{n} \left[ \begin{array}{ccc} \overline{Q}_{11} & \overline{Q}_{12} & \overline{Q}_{13} \\ \overline{Q}_{21} & \overline{Q}_{22} & \overline{Q}_{23} \\ \overline{Q}_{31} & \overline{Q}_{32} & \overline{Q}_{33} \end{array} \right] \times \left\{ \int_{t_{k-1}}^{t_k} \left[ \begin{array}{c} \varepsilon_x^0 \\ \varepsilon_y^0 \\ \gamma_{xy} \end{array} \right] dz + \int_{t_{k-1}}^{t_k} \left[ \begin{array}{c} k_x \\ k_y \\ k_{xy} \end{array} \right] z^2 dz \right\}$$

(15)

As the strain and curvature on the middle surface are independent of the position of the single layer, Equations 14 and 15 can be transformed into the following:

$$\left\{ \begin{array}{c} N_x \\ N_y \\ N_{xy} \end{array} \right\} = \sum_{k=1}^{n} \left[ \begin{array}{ccc} \overline{Q}_{11} & \overline{Q}_{12} & \overline{Q}_{13} \\ \overline{Q}_{21} & \overline{Q}_{22} & \overline{Q}_{23} \\ \overline{Q}_{31} & \overline{Q}_{32} & \overline{Q}_{33} \end{array} \right]_k \times \left\{ (t_k - t_{k-1}) \left[ \begin{array}{c} \varepsilon_x^0 \\ \varepsilon_y^0 \\ \gamma_{xy} \end{array} \right] + \frac{1}{2} (t_k^2 - t_{k-1}^2) \left[ \begin{array}{c} k_x \\ k_y \\ k_{xy} \end{array} \right] \right\}$$

(16)

$$\left\{ \begin{array}{c} M_x \\ M_y \\ M_{xy} \end{array} \right\} = \sum_{k=1}^{n} \left[ \begin{array}{ccc} \overline{Q}_{11} & \overline{Q}_{12} & \overline{Q}_{13} \\ \overline{Q}_{21} & \overline{Q}_{22} & \overline{Q}_{23} \\ \overline{Q}_{31} & \overline{Q}_{32} & \overline{Q}_{33} \end{array} \right]_k \times \left\{ \frac{1}{2} (t_k^2 - t_{k-1}^2) \left[ \begin{array}{c} \varepsilon_x^0 \\ \varepsilon_y^0 \\ \gamma_{xy} \end{array} \right] + \frac{1}{3} (t_k^3 - t_{k-1}^3) \left[ \begin{array}{c} k_x \\ k_y \\ k_{xy} \end{array} \right] \right\}$$

(17)

The aforementioned equations, 16 and 17, can be transformed into the following:

$$\left\{ \begin{array}{c} N_x \\ N_y \\ N_{xy} \end{array} \right\} = \left[ \begin{array}{ccc} A_{11} & A_{12} & A_{13} \\ A_{21} & A_{22} & A_{23} \\ A_{31} & A_{32} & A_{33} \end{array} \right] \left\{ \begin{array}{c} \varepsilon_x^0 \\ \varepsilon_y^0 \\ \gamma_{xy} \end{array} \right\} + \left[ \begin{array}{ccc} B_{11} & B_{12} & B_{13} \\ B_{21} & B_{22} & B_{23} \\ B_{31} & B_{32} & A_{33} \end{array} \right] \left\{ \begin{array}{c} k_x \\ k_y \\ k_{xy} \end{array} \right\}$$

(18)

$$\left\{ \begin{array}{c} M_x \\ M_y \\ M_{xy} \end{array} \right\} = \left[ \begin{array}{ccc} B_{11} & B_{12} & B_{13} \\ B_{21} & B_{22} & B_{23} \\ B_{31} & B_{32} & A_{33} \end{array} \right] \left\{ \begin{array}{c} \varepsilon_x^0 \\ \varepsilon_y^0 \\ \gamma_{xy} \end{array} \right\} + \left[ \begin{array}{ccc} D_{11} & D_{12} & D_{13} \\ D_{21} & D_{22} & D_{23} \\ D_{31} & D_{32} & D_{33} \end{array} \right] \left\{ \begin{array}{c} k_x \\ k_y \\ k_{xy} \end{array} \right\}$$

(19)

In this context, the in-plane stiffness matrix, coupling stiffness matrix, and bending stiffness matrix are represented by the symbols $[A]$、$[B]$ and$[D]$ respectively. Each stiffness coefficient can be expressed as follows:

$$\left. \begin{array}{l} A_{ij} = \sum_{k=1}^{n} \left[ \overline{Q}_{ij} \right] (z_k - z_{k-1}) \\ B_{ij} = \frac{1}{2} \sum_{k=1}^{n} \left[ \overline{Q}_{ij} \right] (z_k^2 - z_{k-1}^2) \\ D_{ij} = \frac{1}{3} \sum_{k=1}^{n} \left[ \overline{Q}_{ij} \right] (z_k^3 - z_{k-1}^3) \end{array} \right\}$$

(20)

## 2.3 PSO-BFO algorithm based on digital twin

**2.3.1 Digital twin.** Digital twin technology creates virtual twins of physical entities by digitizing their structure, performance, and behavior to enable real-time monitoring, simulation prediction, and optimal control of physical systems. The core of the digital twin is data integration and model building, which involves capturing, storing, analyzing, and presenting relevant data about physical entities and their environments. Modeling creates a digital model that is highly

consistent with the physical entity through simulation, computation, and optimization. The digital twin is not only an informational mapping of the physical world, but also a bridge between virtual and real interaction that can respond to changes in the physical world in real time.

In the product design phase, digital twins optimize design, predict performance, and improve product innovation through virtual models. In manufacturing, it improves productivity, reduces defects and optimizes quality control through real-time monitoring and data analysis. During product operation, continuous monitoring, failure alerts and maintenance optimization extend product life, safety and performance. Ultimately, these three are interrelated to achieve total product lifecycle optimization.

**2.3.2 Particle swarm optimization.** Particle swarm optimization (PSO) is an intelligent optimization method inspired by the foraging behavior of bird flocks. In PSO, the position of each particle symbolizes a potential solution within the problem space. These particles move through the solution domain using velocity vectors in order to identify the optimal solution.

Upon completion of each iteration, the particles update their positions and velocities using the optimal extreme values, $Pbest_i$ and $Gbest$. $Pbest_i$ represents the optimal position identified by the ith particle, which is referred to as a local optimal solution. $Gbest$ is the best position found by the particle group, which is called the global optimal solution. The update formula of the position and velocity of the ith particle is as shown in Equation 21.

$$\begin{cases} V_{i,k+1} = wV_{i,k} + c_1 r_1 \left(Pbest_{i,k} - X_{i,k}\right) + c_2 r_2 \left(Gbest_k - X_{i,k}\right) \\ X_{i,k+1} = X_{i,k} + V_{i,k+1} \end{cases} \quad (21)$$

The location of the aforementioned point is as follows: In this context, $V$ represents velocity, $X$ denotes position, and i refers to a specific particle. In this context, the symbol k is used to denote the search generation. The two independent random functions $r_1$ and $r_2$ are defined on the interval $[0, 1]$; $c_1$ and $c_2$ are acceleration factors that assume values within the interval $[0, 2]$ and are employed to modify the local and global optimal position step size, respectively. The variable $w$ represents the inertia weight, which is employed to calibrate the global search and local search capabilities of the particle. An increase in $w$ results in a corresponding enhancement of the global search capability, while a decrease in $w$ yields an equivalent augmentation of the local search capability.

The PSO algorithm is effective in addressing continuous variable optimization; however, it does not directly reflect the spatial position of particles. In the case of discrete optimization of the ply sequence of a composite material, the problem can be converted into a sequence of ply angles. In this representation, the order of access corresponds to the ply sequence of the optimization object, while the length of the path is proportional to the optimization objective. The formulas for updating the position and velocity of particles during the optimization of the ply sequence of a composite material are presented in Equation 22.

$$\begin{cases} V_{i,k+1} = wV_{i,k} \oplus \alpha \left(Pbest_{i,k} - X_{i,k}\right) \oplus \beta \left(Gbest_k - X_{i,k}\right) \\ X_{i,k+1} = X_{i,k} + V_{i,k+1} \end{cases} \quad (22)$$

In this context, the symbol $\oplus$ represents the unary operator. The real numbers $w$, $\alpha$, and $\beta$ are random variables distributed within the interval $[0,1]$. If w is large, then $V_{i,k}$ retains more exchange factors, thereby enhancing its global search ability; conversely, its local search ability is more robust. $\alpha$ and $\beta$ primarily regulate the local optimal step length and global optimal step length, respectively, by influencing the number of exchange factors.

**2.3.3 Bacterial foraging optimization.** Bacterial Foraging Optimization (BFO) is a population-based optimization method initially proposed by Passino. The core mechanism of BFO is trend operation, which determines the direction of bacterial foraging and the strength of their movement, thereby controlling their tumbling and swimming behaviors. Furthermore, the algorithm incorporates cross-breeding and division elimination mechanisms to more accurately simulate the behavior of bacteria. Fig 3 illustrates the two movement modes of BFO in schematic form.

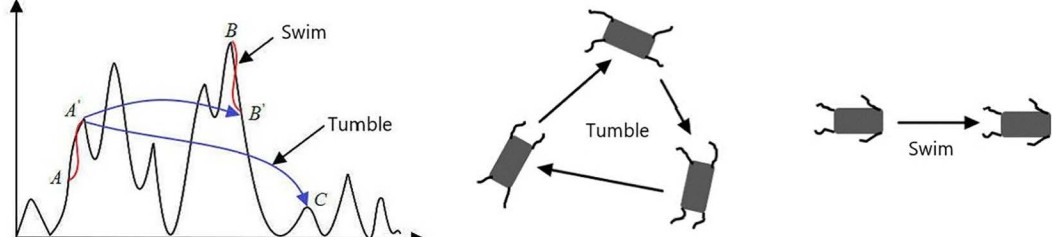

**Fig 3. Illustrates the two distinct movement modes of the BFO.**

The position of the bacteria after rolling and swimming in any direction is as illustrated in Equations 23 and 24.

$$\begin{cases} P(i, q+1, \eta, l) = P(i, q, \eta, l) + c(i) \times \varphi(i) \\ \varphi(i) = \dfrac{\Delta(i)}{\sqrt{\Delta^{T}(i) \times \Delta(i)}} \end{cases} \tag{23}$$

$$P(i, q+1, \eta, l) = P(i, q, \eta, l) + c(i) \tag{24}$$

The location in question is as follows: In this context, $P(i, q, \eta, l)$ represents the position occupied by bacterium $i$ following $q$ trends, $n$ reproductions, and $l$ mitotic eliminations. In this context, $c(i)$ represents the motion step length of the bacterium, $\varphi(i)$ denotes the direction vector of the random motion for each unit step length, and $\Delta(i)$ signifies a random vector that captures the change in the bacterium's direction during tumbling.

**2.3.4 Improved PSO-BFO algorithm.** Although the PSO algorithm has the advantages of high convergence accuracy and efficient calculation, it is susceptible to local optima, which can result in premature convergence and impede the ability to identify the global optimum. To address this issue, this paper presents a hybrid algorithm integrating particle swarm optimization (PSO) and binary foetal osteogenesis (BFO) within a digital twin system. The algorithm effectively circumvents premature convergence by coordinating local search with global search, while enhancing the precision and efficacy of convergence. In this hybrid algorithm, particle swarm optimization (PSO) is employed to determine the optimal location of the flora and to identify both local and global optima. Meanwhile, binary frog leaping optimization (BFO) is utilized to delve deeply into the optima through operations such as elimination, diffusion, tumble, and swim.

During the optimization process, the adaptive value of the particle position $fit(X_i)$ is employed to monitor the state of the particles. When the ratio of the global extreme value fitness $f_g$ to the individual extreme value average fitness $f_{avg}$ approaches 1, it is determined that the particle swarm has reached a state of stasis. At this juncture, the trend operation of BFO is initiated. The formulas utilized to calculate the speed updates for the related fitness $f_g$, $f_{avg}$ and PSO-BFO algorithms are presented in Equations 25 and 26.

$$f_g = fit(Gbest) \tag{25}$$

$$f_{avg} = \frac{\sum\limits_{i=1}^{n} fit(Pbest_i)}{n} \tag{26}$$

$$V_{i,k+1} = wV_{i,k} + b_1 h_1 \left(P_{i,k} - P(i, q, \eta, l)\right) + b_2 h_2 \left(G_k - P(i, q, \eta, l)\right) \tag{27}$$

The optimization process of the PSO-BFO algorithm is illustrated in Fig 4.

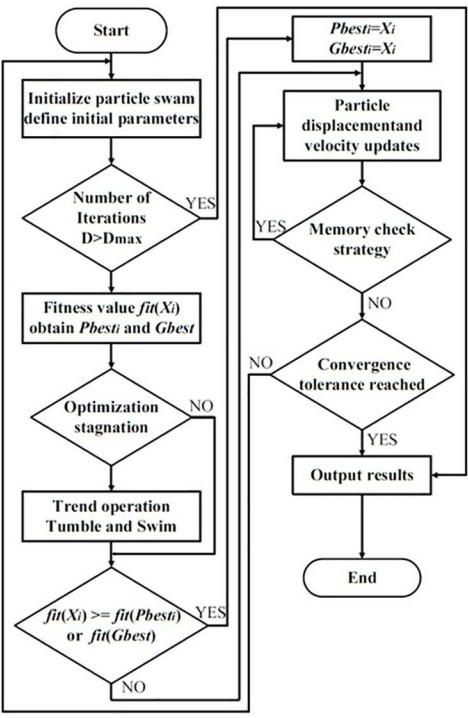

**Fig 4. Digital twin PSO-BFO algorithm optimization process.**

## 3. Finite element modeling and verification of automotive battery boxes

### 3.1 Carbon fiber composite battery box assembly digital twin system

Fig 5 is a flow chart of Carbon fiber reinforced composite battery box assembly digital twin system. First, based on the modal test and modal simulation of the original battery box assembly, the experimental and simulation results are compared and analyzed to verify the accuracy of the established finite element model. On this basis, the original battery box assembly is improved and designed, and the performance analysis and digital modeling of the new battery box are carried out based on the optimized structure. For the upper cover of the battery box, the morphology optimization is carried out, and its material is replaced with sheet molding compound (SMC) composite material. For the lower box part of the new battery box, free size optimization, size optimization and ply sequence optimization are carried out in turn, and the improved particle swarm-bacterial foraging (PSO-BFO) hybrid algorithm is introduced to optimize the composite material ply and improve the calculation efficiency and optimization accuracy. In addition, based on the digital twin data processing system, real-time data transmission and parameter update are realized, thereby improving the feasibility and engineering application value of the optimization scheme.

### 3.2 Finite element modeling of the primary battery box

This section examines a specific battery box model, as illustrated in Fig 6a. In order to facilitate the meshing process using the finite element pre-processing Hypermesh software, it is essential to import the 3D model of the car battery box for geometric cleaning. In order to construct a high-quality simulation model of the battery box, it is necessary to repair and simplify the 3D model of the battery box.

The original battery box is constructed from metal, and the fundamental characteristics of its composition are presented in Table 1 below. The density of the battery box is $7.85 \times 10^{-9}$ t/mm, the elastic modulus is $2.1 \times 10^{5}$ MPa, the Poisson's ratio is 0.3, and the unit size is set to 5 mm.

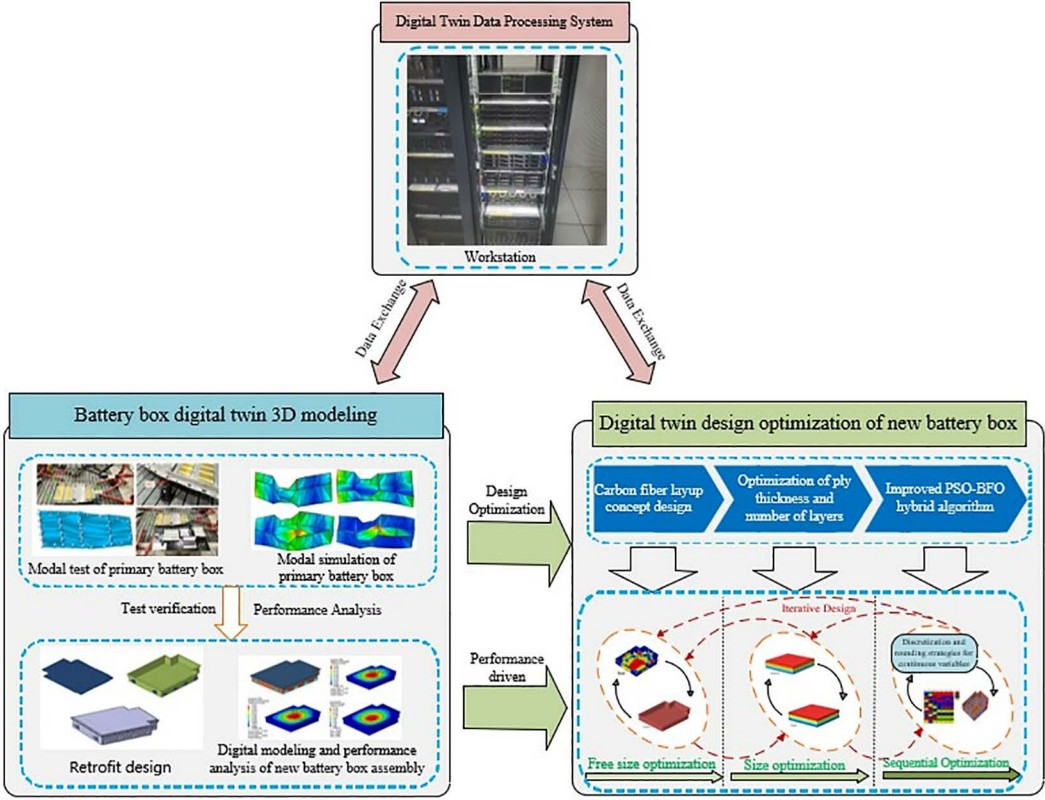

**Fig 5. Carbon fiber reinforced composite battery box assembly digital twin system.**

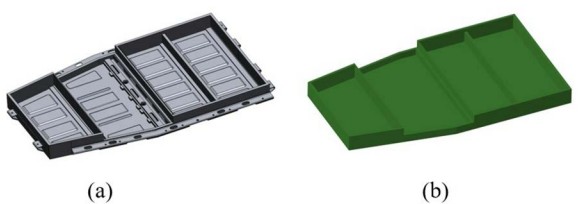

(a)                              (b)

**Fig 6. Primary battery box model.** (a) 3D model. (b) Finite element model.

According to the relevant national standard GB/T 33582−2017 "General Rules for Finite Element Mechanical Analysis of Mechanical Product Structures", the mesh quality of the finite element model of the battery box assembly established in this paper was checked, and the results are shown in Table 1. It can be seen from Table 1 that the mesh quality of the battery box finite element model established in this paper meets the quality standards specified by the regulations. Therefore, the established battery box assembly finite element model can accurately model the actual situation of the battery box assembly.

According to statistics, the number of grid nodes is 175324, the number of grid elements is 805943, and there are 2131 triangular elements, accounting for 0.26% of the total number of shell elements, which meets the requirement that the number of triangular elements does not exceed 5% of the total number of shell elements. The information on the properties of the cell box is shown in Table 2.

The finite element model of the original battery box, obtained following the pretreatment stage, is illustrated in Fig 6b.

**Table 1. Mesh quality evaluation criteria.**

|  | Mesh size | Aspect ratio | Warpage | Skew | Jacobin | Angle quad | Angle tria | percentage |
|---|---|---|---|---|---|---|---|---|
| Regulatory standards | [5,15] | ≤ 5 | ≤ 15 | ≤ 40° | ≥ 0.6 | [40°, 140°] | [30°, 120] | ≤ 5% |
| This model | 5 | 3.2 | 11 | 30 | 0.7 | [70°, 120°] | [50°, 100°] | 0.26 |

**Table 2. Information on the properties of the cell box.**

| Density(t/mm) | Modulus of elasticity | Poisson's ratio | Number of units | Number of nodes | Triangular unit |
|---|---|---|---|---|---|
| $7.85 \times 10^{-9}$ | $2.1 \times 105$ Mpa | 0.3 | 805943 | 175324 | 6124 |

## 3.3 Simulation and verification of the primary battery box

Free mode analysis is employed to ascertain the intrinsic frequency characteristics of a structure in the absence of external constraints. Through free mode analysis, the dynamic response capability of the structure can be evaluated, which facilitates the prediction of the structure's response to external stimuli or other systems.

The mode shape cloud diagram of modal simulation analysis is shown in Fig 7. The first-order mode shape is in a bending state, and its modal frequency is 66.32 Hz; the second-order mode shape is in a torsional state, and its modal frequency is 104.68 Hz; the third-order mode appears as bending and torsion, and its modal frequency is 137.74 Hz; the fourth-order mode appears as oscillating in the transverse direction, and its modal frequency is 150.26 Hz.

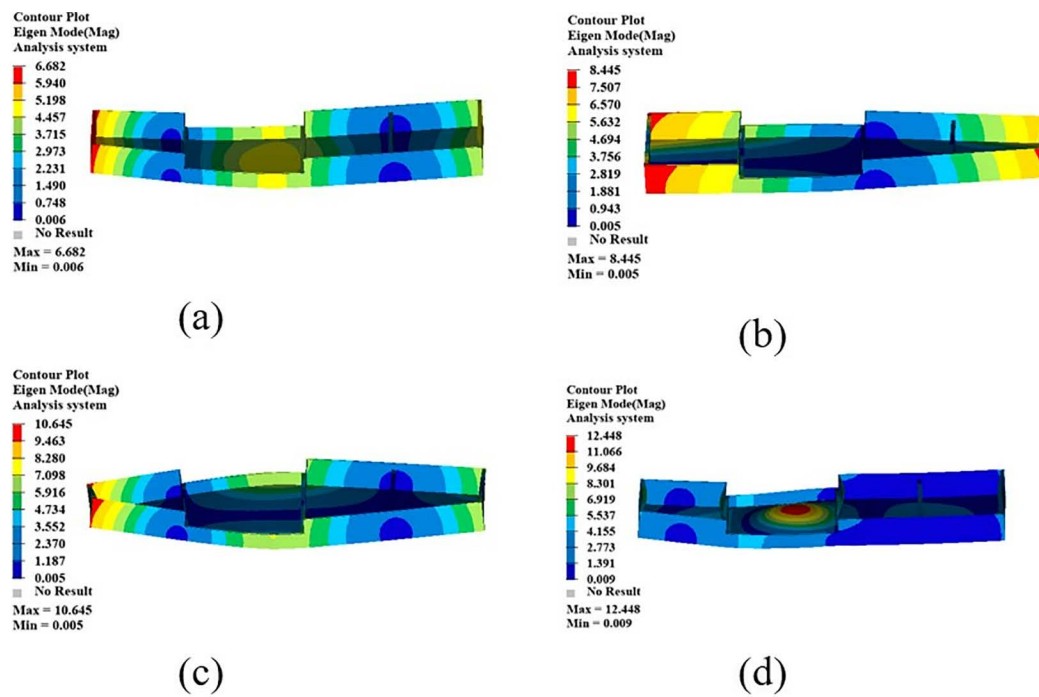

**Fig 7. First four modal shapes of the original battery box.** (a)1st order mode shape. (b) 2nd order mode shape. (c) 3rd order mode shape. (d) 4th order mode shape.

## 3.4 Primary battery case test verification

In the modal test, the frequency sweep signal is transmitted to the exciter through the signal generator and the signal amplifier, and the vibration response data of the battery box is collected by the signal acquisition instrument. The modal characteristics such as the natural frequency, vibration mode and damping ratio of the battery box are obtained by analyzing the collected data. This modal test uses the DHDAS dynamic signal acquisition instrument, 1A312E three-axis acceleration sensor, DH40100 series exciter and DH5872 power amplifier. The test instrument used is shown in Fig 8.

In this modal test, the specific test process includes determining the box support method and suspension height, the location of the sensor measuring point; selecting the excitation point, connecting the equipment, arranging the sensors, etc.

In this test, rubber ropes were used to connect the battery box to simulate the free boundary of the battery box, and four hydraulic cranes were used to control the hanging height of the battery box to keep the battery box in a horizontal state during the test. The specific hanging method is shown in Fig 9a. The excitation point should be selected to avoid being near the structural modal node line. In addition, it is also necessary to consider whether the exciter is convenient to install at the selected excitation point. In this test, the middle edge of the bottom of the battery box is selected as the excitation point, as shown in Fig 9b.

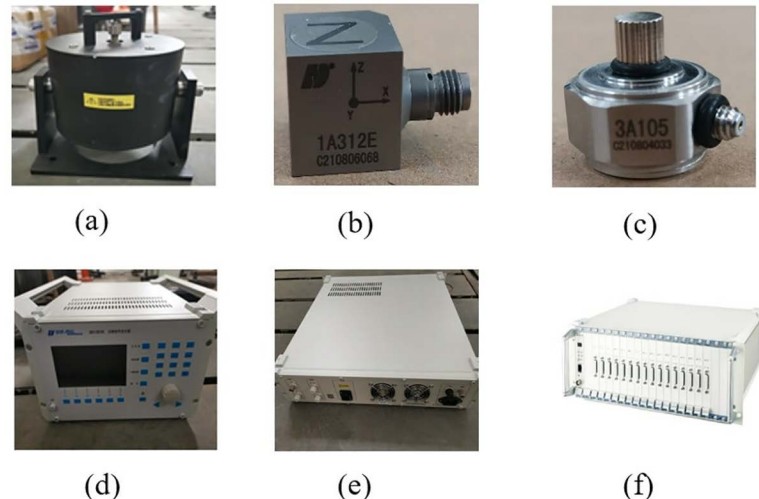

**Fig 8. Test equipment diagram.** (a) Vibrator.(b) Three-axis acceleration sensor.(c) Force sensor.(d) Signal generator.(e) Power Amplifier.(f) Data acquisition instrument.

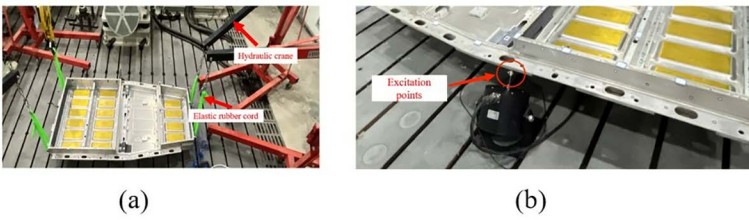

**Fig 9. Support method and selection of excitation point.** (a) Box support method.(b) Selection of the excitation point.

In order to accurately describe the modal shape of the test structure, sufficient acquisition points are required, but the more acquisition points, the more work is involved in the test process. Therefore, when selecting the acquisition points, it is only necessary to consider a reasonable distribution. The distribution of detection points in the selected area of this test is shown in Fig 10a. Equipment Connection The test equipment connection diagram is shown in Fig 10b, where the 24 channels of the sensor and the data acquisition device are connected in sequence; the signal generator and the power amplifier are connected with BNC cables, and the data acquisition device and the computer are connected with a network cable. Several groups of experimental controls were carried out for this test to ensure the credibility of the free modal test results.

### 3.5 Analysis of the results of the free modal test

The first four test modal shapes of the battery box are shown in Fig 11. Compared with the simulated free mode shape of the battery box, the test modal shape is basically the same as the simulated modal shape, which further verifies the correctness of the established finite element simulation model.

The results of the comparison and error analysis of the first four modal frequencies of the battery box free modal simulation analysis and the modal test are shown in Table 3.

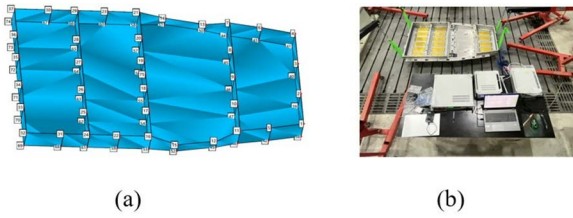

(a) (b)

**Fig 10. Selection of the collection point and connection of the device.** (a) Selection of collection points. (b) Equipment connection.

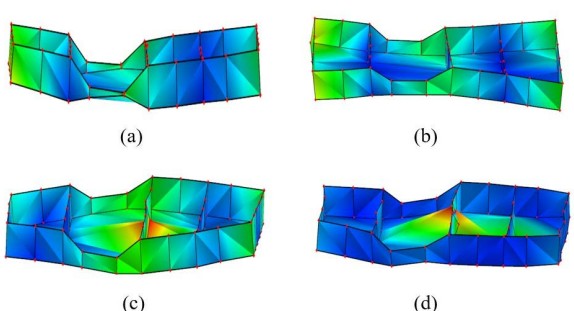

(a) (b)

(c) (d)

**Fig 11. The first 4 modal vibration diagrams of the battery box.** (a)1st order mode shape. (b) 2nd order mode shape. (c) 3rd order mode shape. (d) 4th order mode shape.

**Table 3. Simulation and test results of the first four modal frequencies of the battery box.**

| Frequency order | Simulated frequency/Hz | Test frequency/Hz | Error/% |
|---|---|---|---|
| 1 | 66.32 | 63.97 | 3.67 |
| 2 | 104.68 | 101.34 | 3.29 |
| 3 | 137.74 | 134.16 | 2.26 |
| 4 | 150.26 | 146.48 | 2.58 |

Among them, the maximum error between the free modal test frequency and the simulated modal frequency is 3.67%. The possible reason is that there is a deviation between the real coordinate angle of the three-axis acceleration sensor and the modeling angle, which leads to a maximum error of 3.67% between the simulated modal frequency and the free modal test modal frequency, but meets the accuracy requirements. It can be seen that the established finite element model can better replace the actual battery box for modal analysis research, which further verifies the accuracy of the finite element model and the feasibility of the analysis method used.

The working condition settings and analysis methods in the free modal analysis in this chapter provide some guidance for the constrained modes of the new battery box in the following text, and also provide a reliable modeling environment and analysis methods for the subsequent finite element modeling of the new battery box.

## 4. New battery box design optimization

Given the one-piece molding solution adopted for the body floor in this study, the design of the original battery box needs to be modified. Considering that most electric vehicles currently use square battery boxes, which have high space utilization and are safer and more reliable in structure, and based on the structure of the white body in this project and the current mainstream trend of automotive battery boxes, and based on the optimization design method of the original battery box in the previous section, this section redesigns the original battery box.

### 4.1 New battery box structural design

The specific design of the new battery box is as follows.

(1) Upper cover: The upper cover of the battery box is located at the top of the box and is mainly used for sealing. It does not directly support the weight of the battery module, so it only needs to meet the requirements of the vibration mode frequency. Considering the ease of maintenance of the battery box, it is recommended that the upper cover be bolted to the lower box to facilitate disassembly and to ensure good flatness to ensure the tightness and reliability of the joint.

(2) Lower box: The lower box of the battery box mainly supports the battery module and is a critical part of the structure. Its strength and weight directly affect the safety and service life of the battery box. Since the lower box is installed at the bottom of the vehicle, the working environment is relatively harsh, so special attention must be paid to its structural design.

(3) Reinforced bracket: The reinforced bracket in the battery box is designed to improve the bending and torsional rigidity of the box, preventing deformation and damage to the battery box due to compression or impact. At the same time, the bracket is designed to take into account the rational use of internal space, avoiding the impact on the layout of the battery module due to its excessive size or improper location.

(4) Space design: Based on ensuring that the performance requirements are met, the internal space of the battery box should be maximized to provide more installation locations for the battery modules, thereby improving the range of the electric vehicle.

### 4.2 Reconstruct the battery box model

The parts design module of the 3D modeling software Catia is used to model the components of the battery box one by one according to the dimensions of the components determined in the structural design stage. Then, the assembly design module of Catia is used to assemble the established battery box parts according to the designed assembly relationship to establish a 3D geometric model of the new battery box, as shown in Fig 12a.

Referring to the finite element modeling method and ideas of the original battery box, the finite element modeling of the redesigned battery box is established. Different types of meshing are performed according to the different structures of the

battery box: the battery box is meshed using shell elements; structural parts such as battery modules are meshed using solid elements to establish a finite element model of the new battery box, as shown in Fig 12b.

This paper takes the upper cover and lower box of the battery box as the main research targets, uses shell elements to mesh the battery box, and sets the mesh size to 5 mm. There are 584342 units in total, of which triangular units account for 0.004%, meeting the requirement that the number of triangular units does not exceed 5% of the total number of shell units. The detailed attribute parameters of the new battery box are shown in Table 4.

The next step is to use different types of joints for different parts, as shown in Fig 13a. For the 2D grid, it is also necessary to check the continuity and repeatability of the grid. The result of the mesh continuity check for the battery box is shown in Fig 13b. It can be seen that the quality of the finite element model grid meets the analysis requirements.

### 4.3 Carbon fiber composite test specimen production

In this paper, the composite reinforcement is made of high-strength, high-modulus T700 unidirectional carbon fiber, and the matrix is made of WP-R2300 epoxy resin specially used for prepregs. It has good heat resistance, good surface quality of the molded product and excellent overall performance. Table 5 below lists the relevant parameters of T700 unidirectional carbon fiber and WP-R2300 epoxy resin used in this paper.

**(1) 0° and 90° uniaxial tensile test on laminate.** The 0° and 90° tensile standard specimens are shown in Fig 14a and 14b, which are executed according to ASTM D3039/D3039M-08. The 0° ply standard specimen was laid according to the [0]8 laying method, that is, 8 layers were laid, and the specimen size was 250 mm × 15 mm × 2 mm. The 90° tensile lamination standard specimen was laid according to the laying method of [90]16, that is, 16 layers were laid, and the specimen size was 175 mm × 25 mm × 2 mm. To ensure the accuracy and reliability of this test, the test was conducted

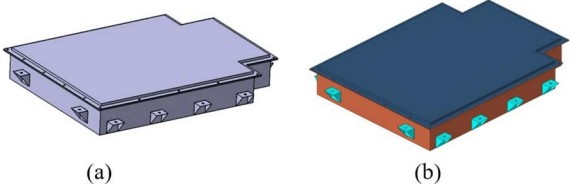

(a)　　　　　　　　　　(b)

**Fig 12. New battery box model after improvement.** (a) 3D model. (b) Finite element model.

**Table 4. New battery compartment properties.**

| Density(t/mm³) | Modulus of elasticity | Poisson's ratio | Number of units | Number of nodes | Triangular unit |
|---|---|---|---|---|---|
| 7.85 × 10⁻⁹ | 2.1 × 105 Mpa | 0.3 | 584342 | 531903 | 2386 |

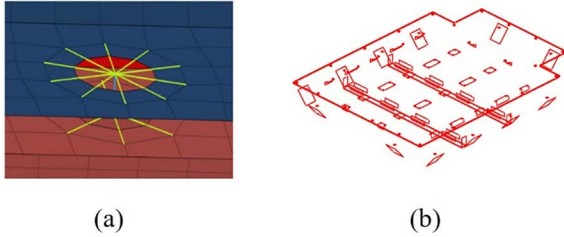

(a)　　　　　　　　　　(b)

**Fig 13. Finite element main structure of the battery box.** (a) Bolted joint. (b) Grid continuity check.

**Table 5. Parameters related to fiber and matrix.**

| Component | Model | Density (g/cm³) | Tensile strength/MPa | Stretch modulus/GPa |
|---|---|---|---|---|
| Disposable tape | T700 | 1.80 | 4900 | 230 |
| Epoxy resin | WP-R2300 | 1.19 | 82 | 2.7 |

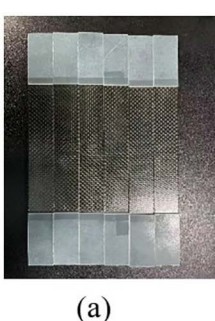 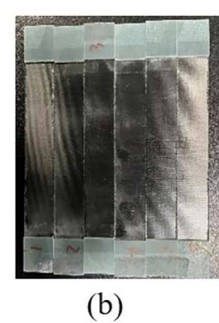 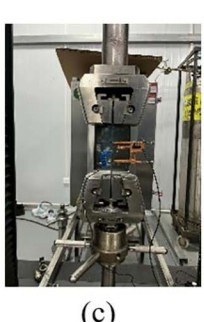

(a)                    (b)                    (c)

**Fig 14. 0° and 90° laminate tensile test.** (a) 0° uniaxial tensile test specimen. (b) 90° laminate tensile specimen. (c) Uniaxial tensile properties test.

more than five times and then compared with the failure form in the implementation standard. The tensile test process is shown in Fig 14.

During the test, the specimen was loaded at a loading rate of 2 mm/min until it was damaged. The damage forms included interlayer delamination and matrix cracking. The damage form of the specimen is shown in Fig 15. The load and displacement data are continuously recorded by the sensors and data acquisition system on the testing machine. After sorting, the test stress-strain curve is shown in Fig 16.

**(2) 0° and 90° uniaxial compression tests on laminates.** The 0° and 90° compression test specimens are shown in Fig 17a and 17b, which were performed according to ASTM D6641/D6641M-09. The 0° and 90° uniaxial compression standard specimens were laid up according to the [0]16 laying method, with 16 layers laid, and the specimen dimensions were 140 mm × 12 mm × 2 mm. During the test, the specimen was loaded at a loading rate of 1.3 mm/min until it was destroyed. To ensure the accuracy and reliability of this test, the test was carried out more than five times and then compared with the destruction form in the implementation standard. The compression test process is shown in Fig 17c, the sample failure form is shown in Fig 18, and the test stress-strain curve after sorting is shown in Fig 19.

**(3) ±45° Shear test within the pavement layer.** The sample photo is shown in Fig 20a, and the shear test process is shown in Fig 20b. According to the ASTM D3518/D3518M-13 standard, the shear standard specimen in the ±45° ply layer is laid according to the [45/-45]4s laying method, and the specimen size is 200 mm × 25 mm × 5 mm. During the test,

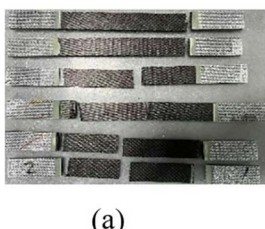 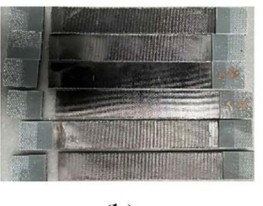

(a)                    (b)

**Fig 15. Specimen failure mode.** (a) 0° uniaxial tensile test specimen. (b) 90° laminate tensile specimen.

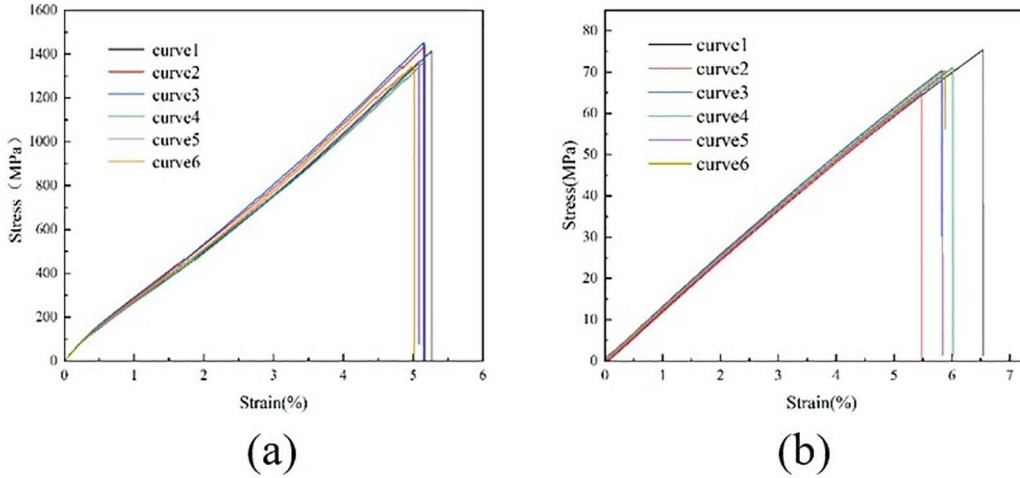

**Fig 16. Stress-strain curves for tensile tests at 0º and 90º.** (a) 0º tensile test stress-strain curve. (b) Stress-strain curve for the 90º tensile test.

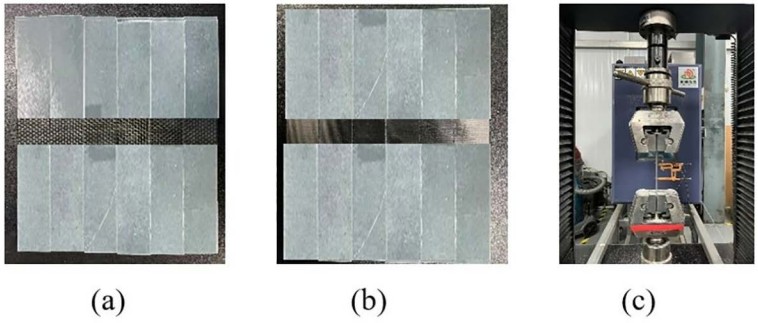

**Fig 17. 0º and 90º laminate compression test.** (a) 0º uniaxial compression specimen. (b) 90º laminate compression specimen. (c) Single-axis compression performance test.

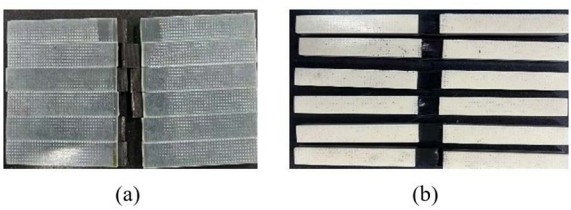

**Fig 18. Specimen failure mode.** (a) 0º uniaxial compression specimen. (b) 90º laminate compression specimen.

the specimen was loaded at a loading rate of 1.3 mm/min until it was damaged, and the damage forms included interlayer delamination and matrix cracking. The load and displacement data continuously recorded by the sensors and data acquisition system on the testing machine were used to draw the stress-strain curve, as shown in Fig 20. After sorting, the test stress-strain curve is shown in Fig 20c.

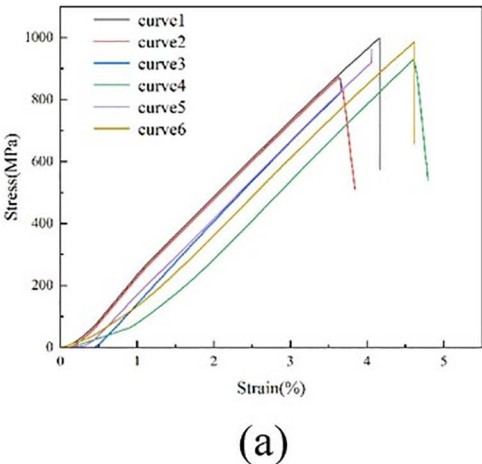
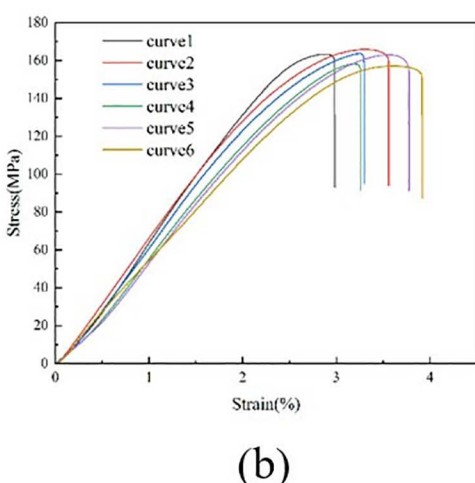

**Fig 19. Stress-strain curves for compression tests at 0º and 90º.** (a) 0º compression test stress-strain curve. (b) 90º compression test stress-strain curve.

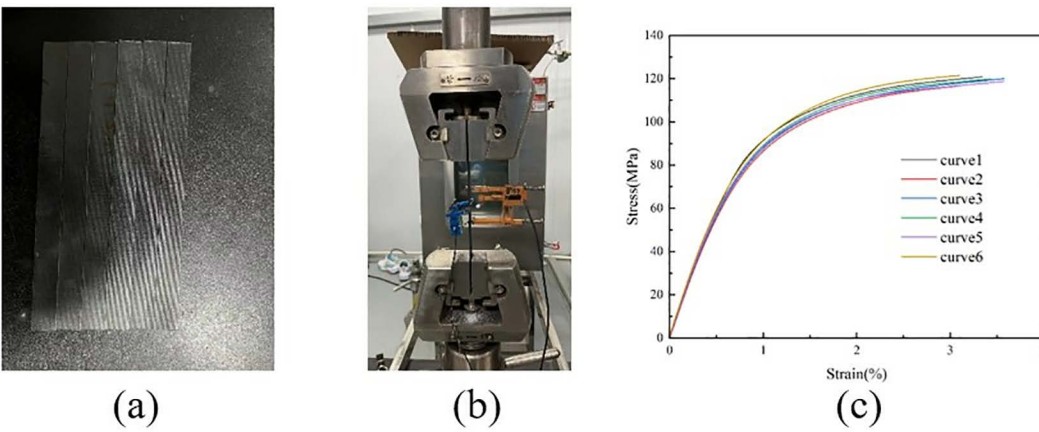

**Fig 20. ±45º Shear test on laminate.** (a) ±45º ply specimens. (b) Shear test within ±45º plane. (c) Stress-strain curve.

The T700/WP-R2300 carbon fiber epoxy composite used in the carbon fiber composite battery case was subjected to a series of tests such as sample making, cutting, stretching, compressing and shearing, and finally obtained its mechanical properties as shown in Table 6. In each test, each test uses 6 groups of repeated tests as controls, and the test results are averaged to ensure the credibility of the test results obtained.

### 4.4 Static performance analysis of the battery box

**4.4.1 Static analysis and setting.** During vehicle operation, the structure of the battery box is primarily subjected to bumps, turns, acceleration, and braking. Based on relevant national standards and related research literature, this paper determines the following three combined working conditions.

(1) Bumpy conditions; (2) Bumpy conditions+sharp left/right turns; (3) Bumpy conditions+emergency braking. The constraints and loading settings for the three combined operating conditions are shown in Table 7.

**Table 6. Parameters related to carbon fiber composite materials.**

| Material parameters | Parameter value | Material parameters | Parameter value |
|---|---|---|---|
| $\rho/(t/mm^3)$ | $1.58\times10^{-9}$ | $E_3/GPa$ | 10.3 |
| $X_t/MPa$ | 1496 | $G_{12}/GPa$ | 6.4 |
| $X_c/MPa$ | 956 | $G_{23}/GPa$ | 3.8 |
| $Y_t/MPa$ | 40 | $G_{13}/GPa$ | 6.4 |
| $Y_c/MPa$ | 249 | $\mu_{12}$ | 0..28 |
| $S/MPa$ | 67 | $\mu_{23}$ | 0.3 |
| $E_1/GPa$ | 127.6 | $\mu_{13}$ | 0.28 |
| $E_2/GPa$ | 13 | | |

**Table 7. Boundary conditions for static analysis.**

| Working Condition | Constraint | Loading settings |
|---|---|---|
| Bumpy | Restrict all degrees of freedom | 2g downward |
| Bumpy+emergency braking | Restrict all degrees of freedom | 2g downward, 1.5g in the direction of braking |
| Bumps+sharp turns | Restrict all degrees of freedom | 2g downward, 1g in the direction of the turn |

The constraints and loading methods in the above-mentioned static analysis conditions are set up, and each condition is solved to obtain the displacement and stress cloud diagram of the new battery box under each condition.

As shown in Fig 21, the maximum displacement occurs at the center of the upper cover with a displacement of 1.22 mm. Under normal circumstances, when the battery box is subjected to an acceleration of 2g on a bumpy road, the maximum displacement does not exceed 2 mm. Therefore, this battery box model meets the displacement requirement. Under this working condition, the maximum stress of the battery box occurs around the box lug, with a stress value of 126.94 MPa, which is caused by stress concentration.

As shown in Fig 22, in the bump+brake condition, the maximum displacement of the battery box occurs at the center of the upper cover, with a maximum displacement value of 0.405 mm, and the maximum stress occurs at the hanging lug, with a maximum stress value of 43.78 Mpa. The maximum stress is much smaller than the yield stress of steel, so there is a lot of room for lightweight.

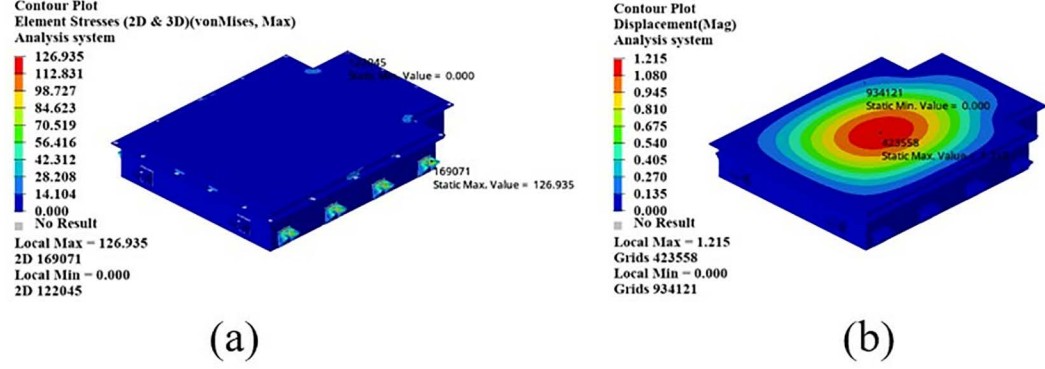

**Fig 21. Bumping conditions cloud chart.** (a) Front side. (b) Reverse side.

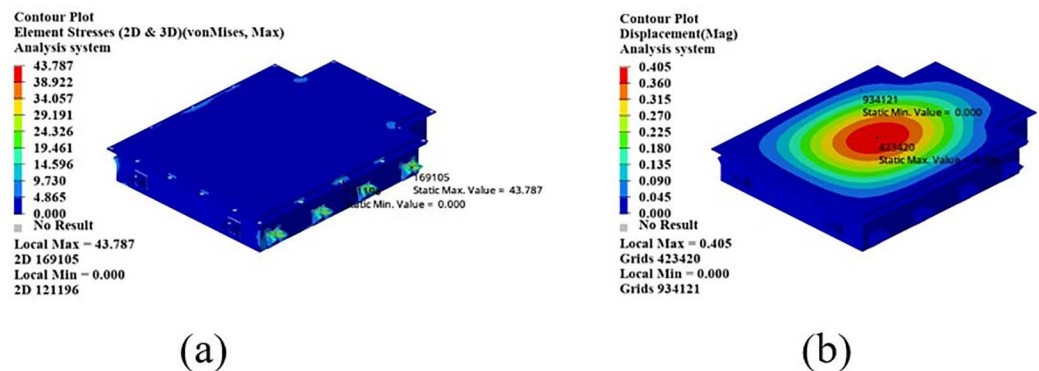

**Fig 22. Cloud diagram of the bumpy＋braking conditions.** (a) Front side. (b) Reverse side.

As shown in Fig 23, under the bump＋turning condition, the maximum displacement occurs at the center of the upper cover of the battery box, and its value is 0.407 mm, which is close to the displacement value of the bump＋braking condition; the maximum stress is 47.32Mpa, which occurs at the lifting ear, and the maximum stress is much smaller than the yield stress of the material.

From the above analysis, it can be seen that the maximum stress under bumping＋braking conditions and bumping＋turning conditions are 43.78Mpa and 47.31Mpa respectively, and the maximum stress values are quite close. The maximum displacement occurs at the center of the upper cover, and the displacement sizes are 0.405 mm and 0.407 mm respectively, which are basically the same and meet the design requirements, with potential for further optimization.

**4.4.2 Modal condition setting and analysis.** In order to simulate the vibration conditions that the battery box is subjected to in actual work, this section performs constrained modal analysis on the battery box. The Optistruct solver is used to solve the statics and constrain the 6 degrees of freedom of all the eye bolt holes. The results of the constrained modal analysis are shown in Fig 24. From the first four modal vibration shapes, it can be seen that the stiffness of the upper cover is relatively weak, which lays the foundation for the subsequent optimization of the three-dimensional morphology of the upper cover of the battery box. Therefore, the morphology of the upper cover of the battery box will be optimized later to improve the stiffness of the upper cover of the new battery box.

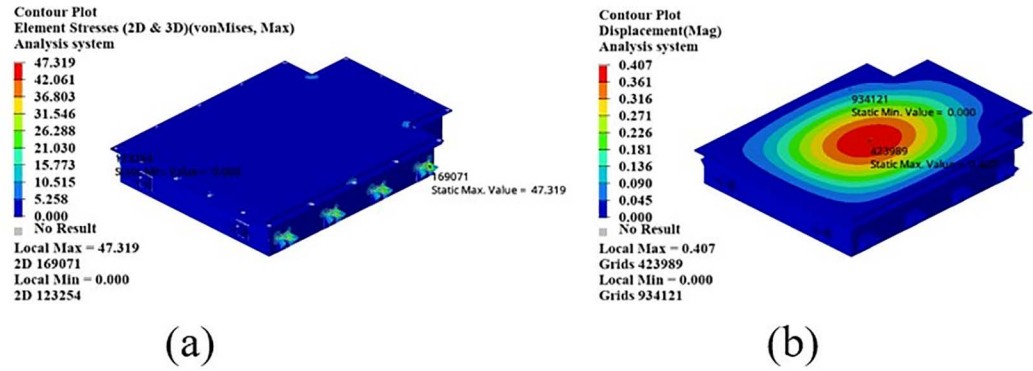

**Fig 23. Bumping＋turning condition cloud map.** (a) Front side. (b) Reverse side.

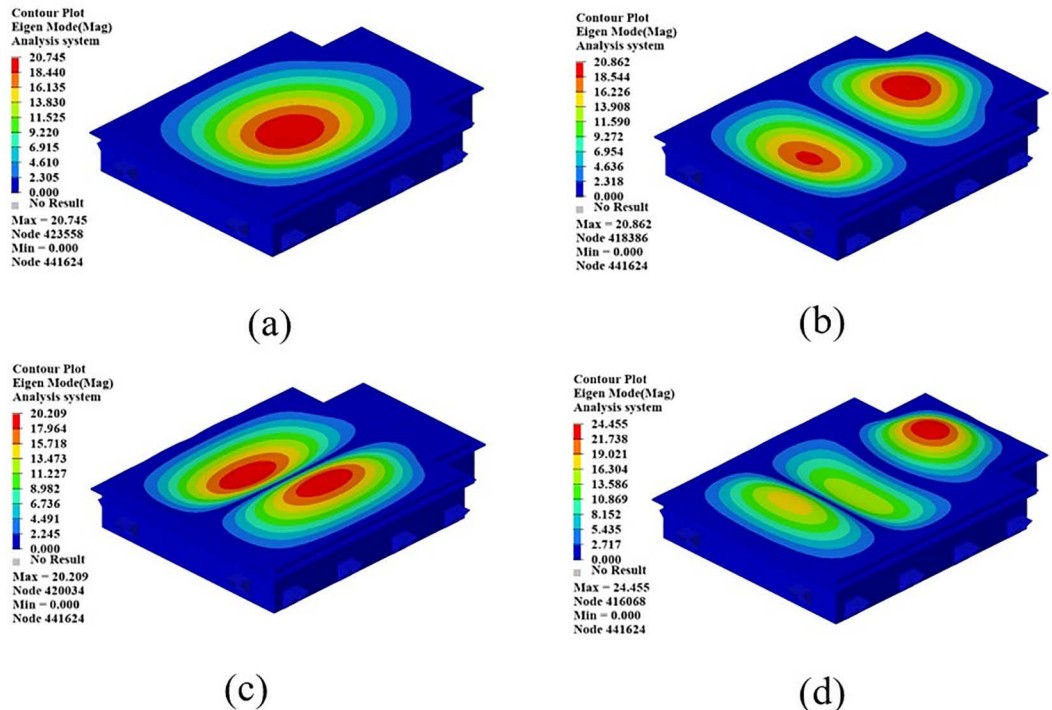

**Fig 24. The first four modal vibration shapes of the battery box.** (a)1st order mode shape. (b) 2nd order mode shape. (c) 3rd order mode shape. (d) 4th order mode shape.

Table 8 shows the first six modal frequencies of the battery box. From the analysis in this table, it can be seen that the first constrained mode of the battery box is 31.59 Hz, and the first constrained mode frequency is 27.78 Hz higher than the highest common road excitation frequency. The modal frequencies of the battery box all meet the requirements.

Analyzing the cloud diagram of the first six constrained mode shapes of the battery box, it can be seen from the Fig that the first six resonance regions of the constrained mode of the battery box all occur in the upper cover area. The upper cover is subject to relatively simple loading conditions and has great potential for weight reduction. In this paper, a sheet molding compound (SMC) composite material is used for subsequent simulation analysis and optimization to consider replacing its material with a lighter composite material. This material can be considered as an isotropic material, and its basic material parameters are similar to those of steel and aluminum.

**Table 8. First six modal frequencies of the battery box.**

| Modal order | Frequency/Hz | Mode shape description |
|---|---|---|
| 1 | 31.59 | The upper cover resonates, with the center showing alternating vibration of a single-peak valley |
| 2 | 38.79 | The upper cover resonates, with alternating peaks and valleys in the front and back. |
| 3 | 50.42 | The upper cover resonates, and the cover vibrates alternately with valleys and peaks on the left and right sides. |
| 4 | 65.37 | The upper cover resonates, with alternating peaks and valleys in the front, middle and back. |
| 5 | 73.42 | The upper cover vibrates, with alternating peaks and valleys in the left front and right front; alternating valleys and peaks in the left rear and right rear. |
| 6 | 89.37 | The upper cover resonates, and the left and right sides of the cover and the center of the cover vibrate alternately with peaks and valleys. |

We suggest the following optimization strategy: First, the shape of the upper cover of the battery box is optimized. While ensuring that the overall quality of the upper cover of the battery box remains unchanged, the size of the upper cover of the battery box is optimized to maximize the first-order modal frequency of the upper cover of the battery box. Then, after the shape optimization and size optimization, the upper cover model is re-modelled by finite elements and replaces the original upper cover in the battery case model to improve the performance of the new upper cover of the battery case.

The lower box, which is subject to complex loading and operating conditions, is designed to be lightweight. According to the fourth section, the original steel battery box has a high safety factor under static conditions, which indicates that the original battery box has an unreasonable material distribution in the lower box, low utilization, and room for further light-weight. Therefore, it is considered to replace the lower box with carbon fiber composite material.

## 5. New battery compartment with optimized design

### 5.1 Optimizing the shape of the upper cover

In this paper, for the battery box upper cover with relatively simple load conditions, it is decided to use SMC composite materials to replace the material of the battery box upper cover. The material parameters are shown in Table 9 below. In addition, the parameters measured in this paper for the SMC material sheet test used for the upper cover of the battery pack box are shown in Fig 25, which can be regarded as an isotropic material.

According to the modal analysis of the battery box in the previous chapter, it can be seen that the resonance area is prone to appear at the position of the upper cover of the battery box. Therefore, the first consideration is to strengthen the

**Table 9. SMC material parameter properties.**

| Density ρ/(g/cm³) | Modulus of elasticity E/GPa | Tensile strength σ_b/MPa | Poisson's ratio |
|---|---|---|---|
| 1.80 | 7.58 | 85.60 | 0.39 |

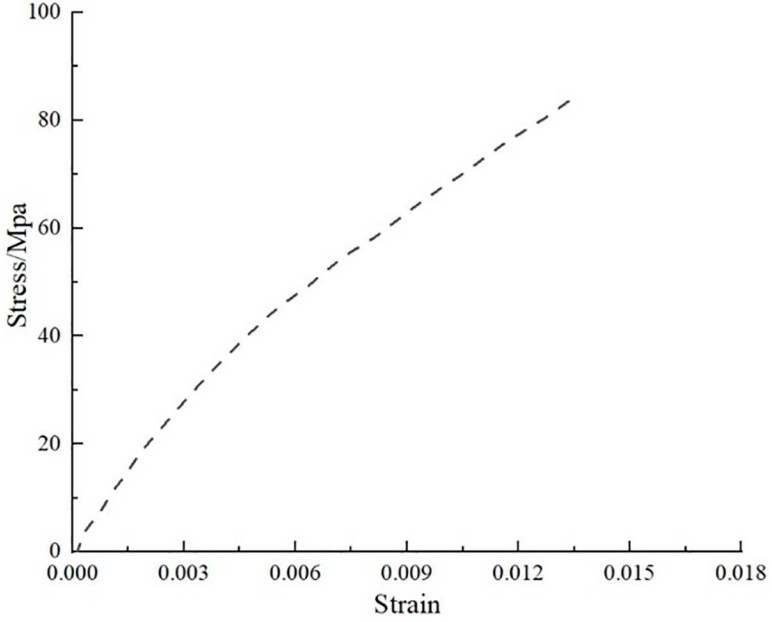

**Fig 25. SMC composite material parameter curve.**

rib structure of the SMC upper cover through morphology optimization to increase its natural frequency. As shown in Fig 26, fix all degrees of freedom at the junction between the upper cover and the lower box. Set the top of the upper cover of the battery box as the design domain and the junction between the side wall of the upper cover and the lower box as the non-design domain.

The optimization objective is the maximum of the constrained first-order modal frequency, the shape parameters are used as constraints, and the cell shape is used as a design variable. The optimization process converges after the 22nd solution iteration, as shown in Fig 27.

The shape and size parameters of the ribs were extracted using the OSSmooth post-processing module in the Optistruct solver. Combined with a process feasibility analysis, the geometry of the ribs generated during the upper cover optimization process was modified in 3D modeling software. The shape and position distribution of the modified upper cover ribs are shown in Fig 28.

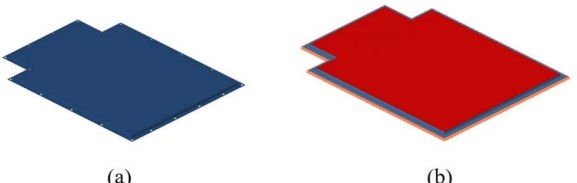

(a)  (b)

**Fig 26. Shape optimization design space division.** (a) Battery box upper cover. (b) Divide the area.

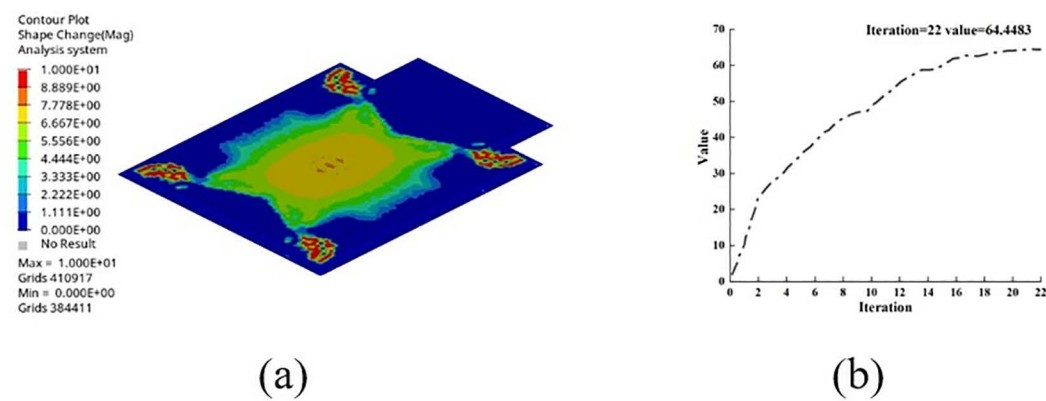

(a)  (b)

**Fig 27. Morphology optimization results.** (a) Optimization results. (b) Optimization iteration graph.

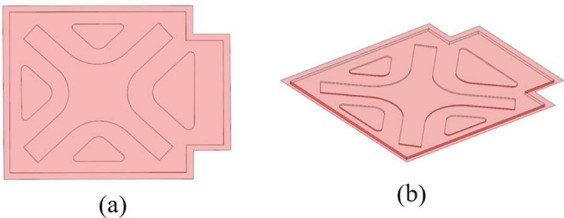

(a)  (b)

**Fig 28. Shape and position distribution of stiffening ribs.** (a) Front side (b) Reverse side.

## 5.2 Optimization of the size of the upper cover of the battery box

In order to further improve the lightweight effect of the battery box upper cover, the size of the battery box upper cover after morphology optimization is optimized to find the optimal thickness of the upper cover. The thickness of the upper cover is used as a design variable, and the thickness variation range is set to 1–3 mm. A first-order modal frequency higher than 30 Hz is used as a constraint, and the minimum mass is used as the optimization target. After seven iterations, the optimal thickness of the upper cover is 1.83 mm, as shown in Fig 29. After comprehensively considering the manufacturing process, the thickness of the upper cover is determined to be 2 mm. After optimizing the design of the upper cover, the mass is reduced by 35.6%.

## 5.3 New battery compartment lower compartment optimized design

### 5.3.1 Optimization of the free dimensions of the lower box.
Units with the same laminate direction are grouped into a set, which is considered a single laminate and is used as a design variable for free size optimization. Laminates with the same laminate angle are referred to as super laminates. In this paper, the initial thickness of the super laminate is set to 2.5 mm. As shown in Fig 30a.

Considering the manufacturing process and design requirements, this paper sets the superlayer with four ply angles of 0°, 90°, +45° and −45°, as shown in Fig 29a. In the dimensional optimization stage, the thickness of the superlayer at each angle is used as the design variable, and the design of each superlayer is continuously optimized. By changing the thickness of each layer and the fiber direction of each unit, the total thickness of the laminated sheet can be continuously varied throughout the structure, as shown in Fig 30b.

During the free-size optimization process, certain manufacturing constraints must be met, such as the proportion of each angle of the laminate is not less than 10%, the thickness and shape of the ±45° superlayer are consistent, and the optimization elements for the free-size optimization of the lower box of the new battery box are as follows:

(1) Design variables: superlayer thickness for four layer angles of 0°, 45°, −45° and 90°

(2) Optimization goal: minimum weighted deflection for three operating conditions: bumps, bumps + turns, and bumps + braking

(3) Restrictions: The mass of the lower box is less than 25 kg, and the first-order modal frequency is greater than 30 Hz.

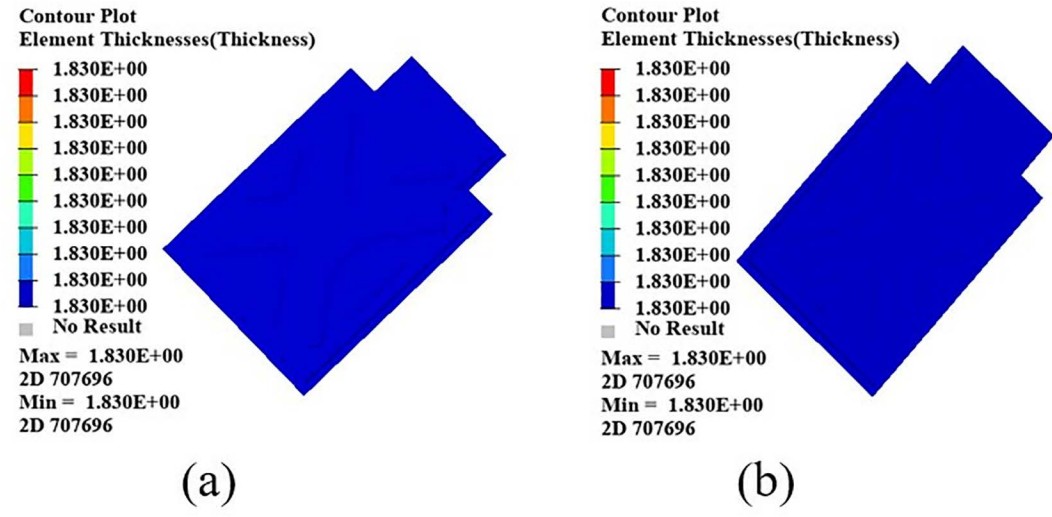

**Fig 29. Upper cover size optimization.** Optimization result. (a) Front side (b) Reverse side.

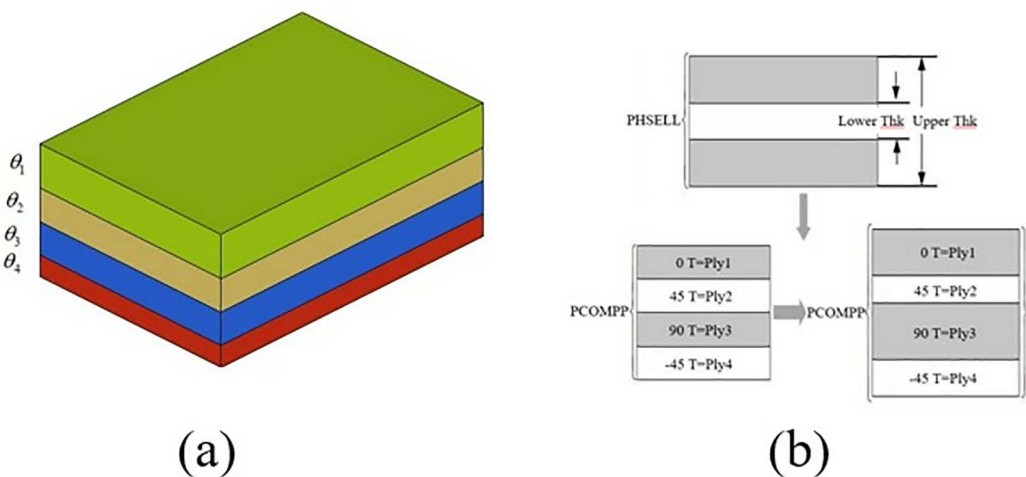

**Fig 30. Schematic diagram of the super-layer.** (a) Schematic of the superlayer. (b) Schematic diagram of superlayer optimization.

After 17 iterations of optimization convergence, the resulting composite thickness distribution is shown in Fig 31. The values in the Fig represent the total thickness distribution of the lower box of the carbon fiber composite battery box. After free size optimization, the total thickness of the lower box of the carbon fiber composite battery box changes from 10 mm to 6.4 mm.

There are four angles: 0°, +45°, −45°, and 90°. After free size optimization, a total of 16 plies are generated, as shown in Fig 32. After free size optimization, each super ply is divided into four plies with unique shapes. Since the ±45° super ply is symmetrical in the previous optimization, the shapes of the plies at these two angles are the same.

As a consequence of the symmetry balance constraint, the new battery box lower box plies are symmetrically balanced, resulting in an actual number of plies in each ply block of 2N. Table 10 illustrates the outcomes of the half-thickness ply of the lower box of the carbon fiber composite material. The lower box of the new battery box comprises a total of 15.4×2 unidirectional plies. The first digit in the table represents the design area; the second digit represents the ply angle, with 1, 2, 3, and 4 representing 0°, +45°, 90°, and −45° plies, respectively; the third digit represents the shape of the optimized ply block; and the fourth and fifth digits represent different plies of the same shape.

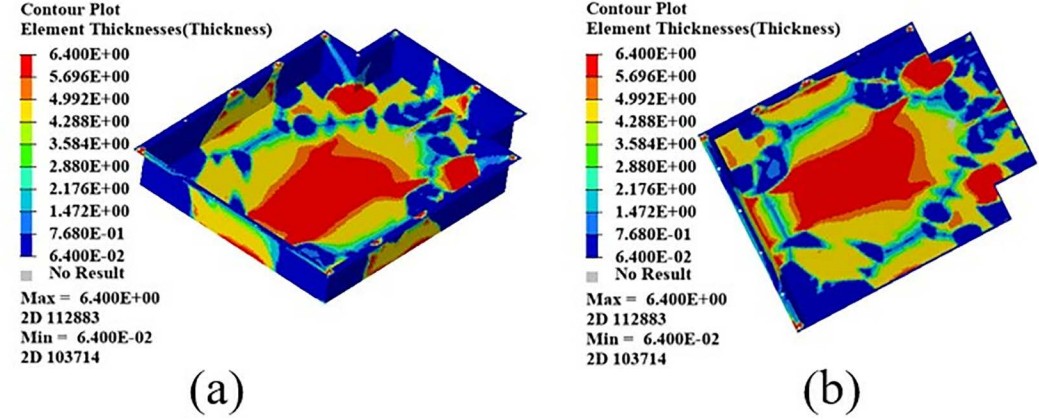

**Fig 31. Free size optimized thickness distribution.** (a) Front side (b) Reverse side.

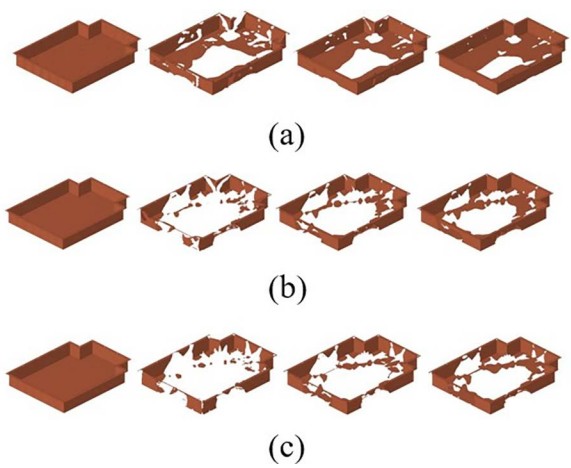

(a)

(b)

(c)

**Fig 32. Shape of the layer blocks.** (a) 0° ply block. (b) ±45° ply block. (c) 90° ply block.

**Table 10. Results of the optimization of the thickness of the lower box lining.**

| Lamination number | Thickness/mm | Layer | Lamination number | Thickness/mm | Layer |
|---|---|---|---|---|---|
| 61100 | 0.0521 | 0.4168 | 63100 | 0.0546 | 0.4368 |
| 61200 | 0.1913 | 1.5304 | 63200 | 0.0794 | 0.6352 |
| 61300 | 0.1426 | 1.1408 | 63300 | 0.2047 | 1.6376 |
| 61400 | 0.2144 | 1.7152 | 63400 | 0.0612 | 0.4896 |
| 62100 | 0.0546 | 0.4368 | 64100 | 0.0589 | 0.4712 |
| 62200 | 0.0794 | 0.6352 | 64200 | 0.1773 | 1.4184 |
| 62300 | 0.2047 | 1.6376 | 64300 | 0.0768 | 0.6144 |
| 62400 | 0.0612 | 0.4896 | 64400 | 0.2128 | 1.7028 |

## 5.4 lower box size optimization

The shape of the ply block that results from free-size optimization is typically highly irregular, which presents significant challenges for subsequent manufacturing processes. Consequently, normalization is essential to enable subsequent industrial ply cutting. Fig 33 illustrates the appearance of the super ply block following cutting at various angles.

The mathematical model for optimizing the size of the lower box of the carbon fiber composite battery box is described as follows:

(1) Design variables: single-layer thickness

(2) Optimization goal: minimum mass of the lower box of the battery compartment

(3) Restrictions: The first mode is greater than 30 Hz; The maximum displacement under bumpy and turning conditions is less than 2 mm; the maximum displacement in the bumpy braking condition is less than 2 mm; the composite material failure factor is less than 1 under each working condition.

In the design optimization of carbon fiber composite materials, size optimization often leads to uneven thickness of each layer, which increases the difficulty of manufacturing and makes it difficult to apply in practice. Traditional methods ensure consistency by setting manufacturing constraints in simulation, but lack flexibility. In this study, digital twin technology is used to ensure uniform ply thickness through real-time monitoring and dynamic adjustment of the design, and

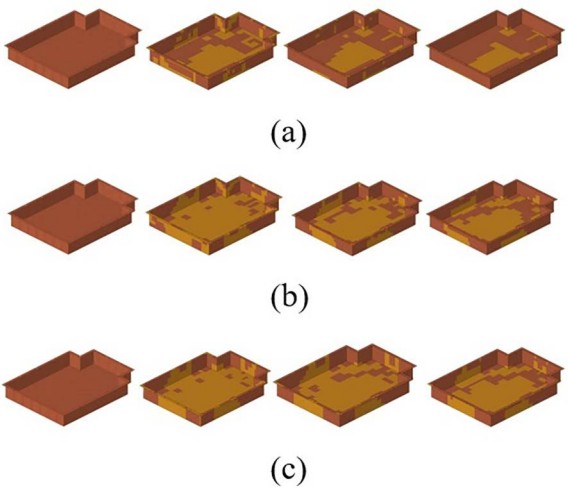

**Fig 33. Shape of the laminate block after regularization.** (a) 0° ply block. (b) ±45° ply block. (c) 90° ply block.

continuously optimize based on manufacturing feedback. This not only improves the accuracy and reliability of the design and reduces manufacturing defects, but also shortens the development cycle, reduces costs, and makes the optimized design more suitable for actual engineering. By introducing digital twin technology, the dynamic management of manufacturing constraints is realized in the design optimization of carbon fiber composite materials, which significantly improves the practical application value and manufacturing feasibility of the optimized design of carbon fiber composite materials. In this study, the layer thickness is set to 0.125 mm, and the process of optimizing manufacturing constraints based on digital twin technology is shown in Fig 34.

**5.4.1 Continuous variable discretization rounding strategy ply.** In order to obtain the discrete ply numbers of each ply angle in the lower box of the carbon fiber composite battery box, a downward rounding strategy is employed to discretize the ply thickness of each ply angle in the lower box of the carbon fiber composite. While this inevitably diminishes the stiffness of the laminate, it is necessary to incorporate supplementary layers into the laminate to ensure the load-bearing capacity of the structure. This must be done in a manner that also considers the objective of reducing the overall weight of the structure.

The ply design employed in this study utilized four angles, specifically 0°, 45°, −45°, and 90°. It should be noted that the application of the down-rounding strategy may result in the loss of up to four plies.

The number of additional layers for the 0° and 90° paving angles can be expressed as a combination of 0, 1, 2, 3, or 4, respectively. Due to the symmetry and balance constraints, the number of layers for the 45° and −45° paving angles appear in pairs. Consequently, the combinations of the number of additional layers for ±45° are 0, 1, and 2. The process of the rounding strategy for discretizing continuous variables is illustrated in Fig 35.

The outcomes of Table 9 are employed as a foundation for the discretization of laminate data. The lower compartment of the composite battery case is subjected to a continuous variable discretization strategy. Among them, Table 11 shows the thickness of the laminate obtained after discretization and rounding, and the lower box module of the battery box is 15.4 × 2 unidirectional laminates. First, continuous dimensional optimization is employed to ascertain the optimal ply thickness for each layer. Subsequently, in conjunction with the manufacturable size of 0.125 mm for single-layer boards in engineering, the manufacturable ply thickness for each layer is determined through a discretization and rounding strategy.

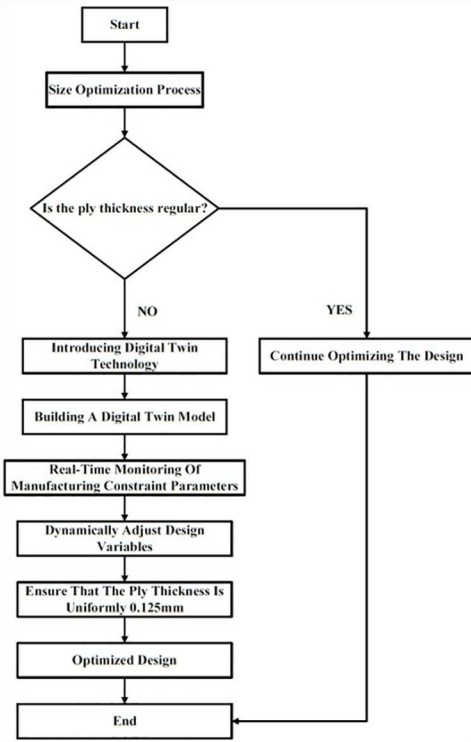

**Fig 34. Optimizing manufacturing constraints using digital twin technology.**

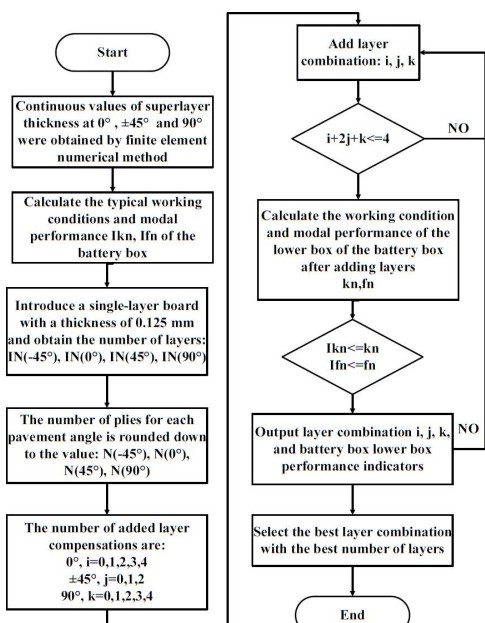

**Fig 35. Rounding strategy for discretization of continuous variables.**

**Table 11. Manufacturable ply thickness.**

| Lamination number | Thickness/ mm | Layer | Number of discrete layers | Lamination number | Thickness/ mm | Layer | Number of discrete layers |
|---|---|---|---|---|---|---|---|
| 61100 | 0.0521 | 0.4168 | 0 | 63100 | 0.0546 | 0.4368 | 0 |
| 61200 | 0.1913 | 1.5304 | 2 | 63200 | 0.0794 | 0.6352 | 1 |
| 61300 | 0.1426 | 1.1408 | 1 | 63300 | 0.2047 | 1.6376 | 2 |
| 61400 | 0.2144 | 1.7152 | 2 | 63400 | 0.0612 | 0.4896 | 0 |
| 62100 | 0.0546 | 0.4368 | 0 | 64100 | 0.0589 | 0.4712 | 0 |
| 62200 | 0.0794 | 0.6352 | 1 | 64200 | 0.1773 | 1.4184 | 2 |
| 62300 | 0.2074 | 1.6376 | 2 | 64300 | 0.0768 | 0.6144 | 1 |
| 62400 | 0.0612 | 0.4896 | 0 | 64400 | 0.2241 | 1.7928 | 2 |

## 5.5 Optimization of ply sequence

### 5.5.1 Optimization of the design process for the lower box plywood sequence.
By employing free-form sizing and sizing optimization techniques for the lower section of the carbon fiber composite battery box, a variable thickness structure was devised, resulting in a continuous laminate thickness distribution. In order to maintain the rigidity of the lower box, the thickness of the continuous laminate is converted into a discrete number of layers using a continuous variable discrete rounding strategy, which is employed as an input parameter for the optimization of the ply sequence design. Subsequently, the global ply sequence was optimized using the PSO-BFO optimization algorithm, with optimization progress monitored using a digital twin. The optimization design process of the ply layup sequence of the lower box of the composite battery box is shown in Fig 36.

The composite battery box lower box scheme, designed using continuous variable thickness optimization, meets the performance requirements of the battery box. However, in actual production, factors such as processability limitations, the adverse effect of the fiber layup sequence on internal forces, and constraints to ensure fiber continuity must also be considered. The mathematical model is described as follows:

(1) Design variables: different permutations of carbon fiber composite materials

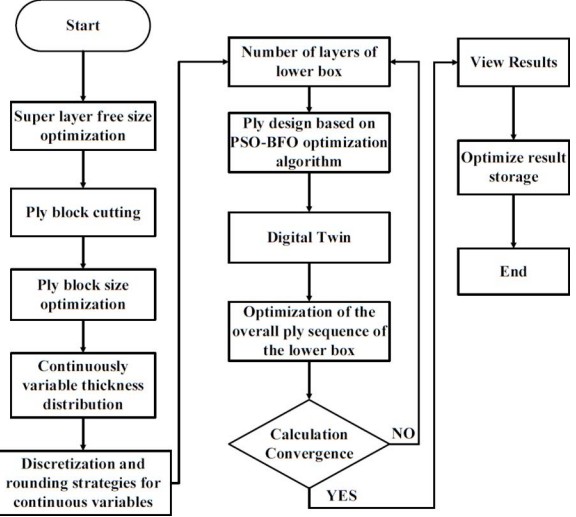

**Fig 36. Digital twin design process for optimizing the ply sequence.**

(2) Optimization goal: First-order modal frequency is maximized.

(3) Optimization constraints: The same direction of ply cannot appear more than 2 layers in a row, and ±45° ply appears in pairs.

Once the optimization model had been conFigd in accordance with the aforementioned optimization criteria and solved using the OptiStruct solver, the carbon fiber ply sequence exhibiting the greatest first-order modal frequency was optimized following six iterations, as illustrated in Fig 37. As can be observed, the angles 45° and −45° manifest in pairs, in accordance with the previously specified requirements, thereby satisfying the constraints associated with the symmetrical ply process for ply manufacturing. At this point, the layup angle, thickness, and sequence of the lower box of the carbon fiber composite battery box have been optimized. The number of plies is 30, and the layup sequence of the lower box can be expressed as [90/90/0/0/90/0/0/90/45/-45/0/45/-45/45/-45]s.

Fig 38 shows the convergence characteristics of the three optimization algorithms, PSO, PSO-BFO and BFO. There are significant differences in convergence speed, final fitness value and stability. PSO-BFO reached stability with the fastest convergence speed of 6500 iterations and achieved the lowest fitness value, indicating that it has achieved a good balance between global search and local optimization. PSO converged second and tended to be stable after 7500 iterations. Its fitness value was slightly higher than PSO-BFO, indicating that it has strong optimization ability but is prone to fall into local optimality. BFO converged the slowest and did not stabilize until 9000 iterations. It had the highest fitness value, showing that it has strong global search ability but low optimization efficiency. The fluctuation characteristics after

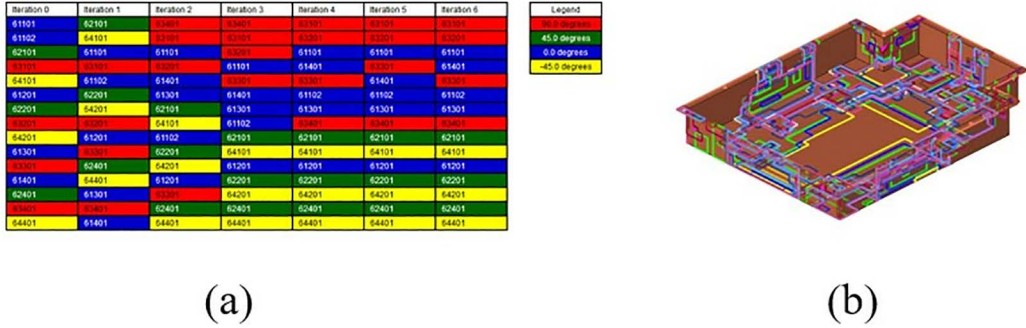

(a)                                                                    (b)

**Fig 37. Results of the ply sequence optimization.** (a) Results of ply sequence (b) Laminate model.

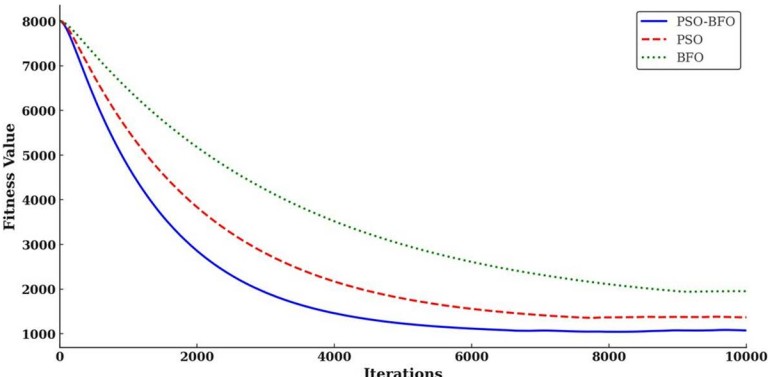

**Fig 38. Comparison of Convergence Characteristics of PSO, PSO-BFO, and BFO Algorithms.**

convergence are also different. PSO has the smallest change, PSO-BFO has moderate fluctuations, and BFO has large oscillations, indicating that PSO-BFO still has a certain local search ability, while BFO is not stable enough in the convergence process due to the characteristics of the search strategy. In general, PSO is suitable for optimization tasks that require high computational efficiency, PSO-BFO is suitable for achieving a balance between optimization quality and computational time, and BFO is more suitable for complex optimization problems that require strong global search capabilities.

**5.6 Comparison of different optimization algorithms**

In order to verify the effectiveness of the hybrid optimization algorithm proposed in this paper, under the same conditions, a single optimization algorithm and a hybrid optimization algorithm are used to optimize the ply design of the lower box of the battery box, and the results are compared and analyzed. Table 12 is a comparison of the results of using different optimization algorithms. From the results, it can be seen that different optimization methods can improve the performance of the battery box and reduce its quality to a certain extent, but using a single optimization algorithm is prone to fall into a local optimal solution. In contrast, the hybrid optimization algorithm can produce more perfect results. The proposed PSO-BFO algorithm improves the mass M of the battery box assembly, the first-order modal frequency of the battery box, the stress under bumpy conditions, and the displacement under bumpy conditions by 40.63%, 33.34%, 83.60%, and 62.80%, respectively. The PSO-GA algorithm improves the mass of the battery box assembly, the first-order modal frequency of the battery box, the stress under bumpy conditions, and the displacement under bumpy conditions by 33.50%, 29.31%, 83.04%, and 57.87%, respectively. Although both the proposed PSO-BFO algorithm and PSO-GA algorithm can obtain the global optimal solution of complex optimization problems, the results obtained by the PSO-BFO algorithm are more balanced than those of the PSO-GA algorithm. Therefore, the PSO-BFO algorithm proposed in this paper can effectively solve the multi-objective optimization problem of the ply layup sequence of the lower box of the carbon fiber composite battery box.

# 6. Compare and verify performance before and after optimization

This chapter verifies the feasibility of the lightweight design method proposed in this paper by comparing the performance parameters of the upper cover, lower box, and battery box assembly of the battery box before and after optimization.

**Table 12. Comparison of results using different optimization algorithms.**

| Optimization methods | Optimization Technology | Optimize results | Improvement rate |
|---|---|---|---|
| Single optimization algorithm | PSO | Mass = 62.14 kg<br>Modal frequency = 45.21 Hz<br>Bump stress = 23.245 Mpa<br>Bump displacement = 0.652 mm | Mass = 29.85<br>Modal frequency = 30.13%<br>Bump stress = 81.69%<br>Bump displacement = 56.3% |
| | BFO | Mass = 60.47 kg<br>Modal frequency = 43.21 Hz<br>Bump stress = 22.842 Mpa<br>Bump displacement = 0.648 mm | Mass = 31.73%<br>Modal frequency = 26.89%<br>Bump stress = 82.00%<br>Bump displacement = 46.67% |
| Hybrid Optimization Algorithm | PSO-BFO | Mass = 52.82 kg<br>Modal frequency = 47.39 Hz<br>Bump stress = 20.821 Mpa<br>Bump displacement = 0.452 mm | Mass = 40.63%<br>Modal frequency = 33.34%<br>Bump stress = 83.60%<br>Bump displacement = 62.80% |
| | PSO-GA | Mass = 58.92 kg<br>Modal frequency = 44.69 Hz<br>Bump stress = 21.524 Mpa<br>Bump displacement = 0.512 mm | Mass = 33.50%<br>Modal frequency = 29.31%<br>Bump stress = 83.04%<br>Bump displacement = 57.87% |

## 6.1 Performance comparison of the upper cover before and after optimization

### 6.1.1 Comparison of the static analysis of the upper cover.
According to the previous setting of the boundary conditions for the static analysis, this section compares and analyzes the stress and displacement of the upper cover of the battery box under three static working conditions before and after optimization. Fig 39 shows the displacement and stress cloud diagrams of the upper cover before and after optimization under the bumping working condition. In the stress cloud diagram, it can be seen that the stress distribution of the upper cover of the battery box has changed after optimization. In the displacement cloud diagram, it can be seen that the displacement distribution of the upper cover of the battery box is basically the same before and after optimization.

Table 13 compares the static performance of the upper cover under bumpy conditions. As illustrated in the table, following optimization under challenging conditions, the maximum stress on the upper cover of the battery box is reduced from 14.85 MPa to 13.67 MPa, representing a 7.95% improvement; the maximum displacement is reduced from 1.193 mm to 0.452 mm, or a 62.11% improvement.

Under the bump + braking conditions, the static performance comparison cloud chart of the battery box upper cover is shown in Fig 40. In the stress cloud diagram, it can be seen that the stress distribution of the upper cover of the battery box has changed after optimization. In the displacement cloud diagram, it can be seen that the displacement distribution of the upper cover of the battery box is basically the same before and after optimization.

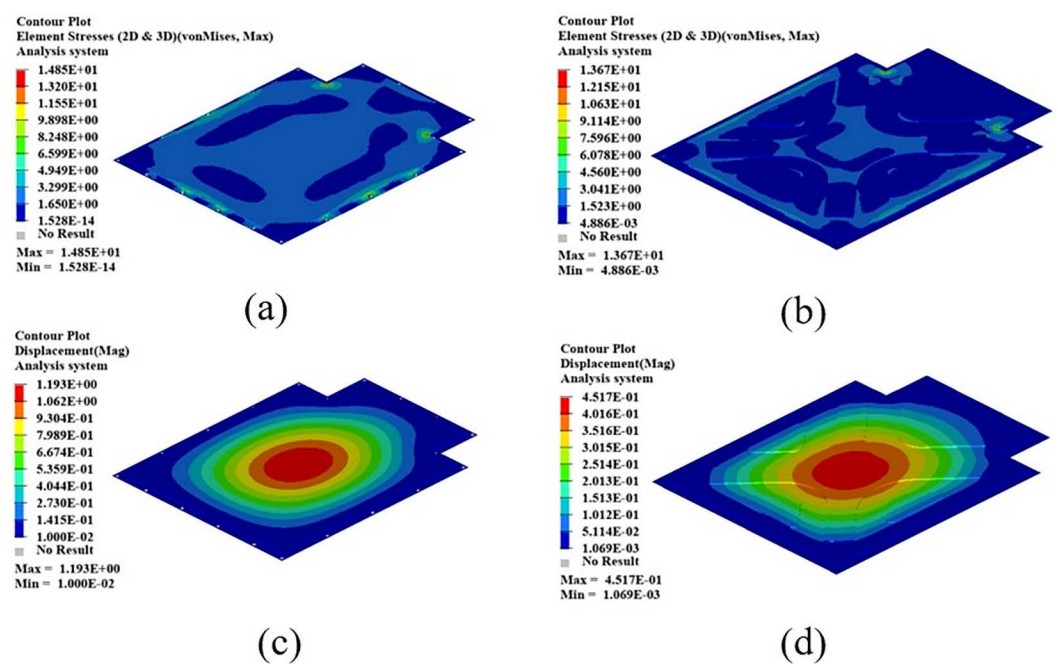

**Fig 39. Performance cloud of upper cover under bumpy conditions.** (a) Stress cloud map before optimization. (b) Stress cloud map after optimization. (c) Displacement cloud map before optimization. (d) Displacement cloud map after optimization.

**Table 13. Comparison of the performance of the upper cover under bumpy conditions.**

| Performance indicator | Before optimization | After optimization | Improvement rate (%) |
|---|---|---|---|
| Stress (Mpa) | 14.85 | 13.67 | 7.95 |
| Displacement (mm) | 1.193 | 0.452 | 62.11 |

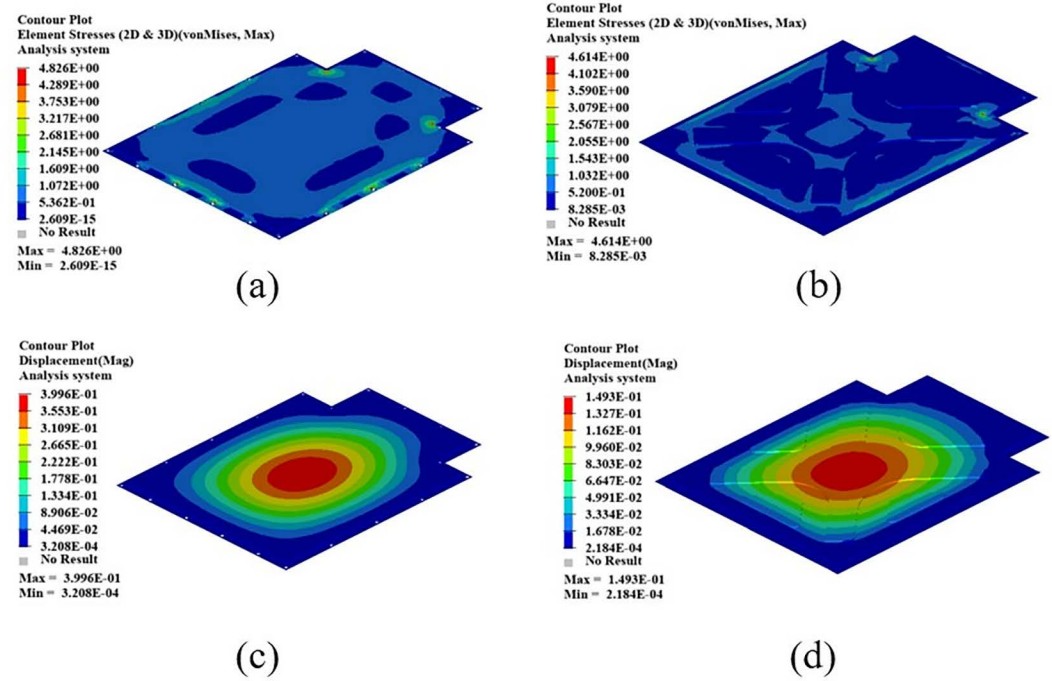

**Fig 40. Performance cloud diagram of upper cover under bumpy + braking conditions.** (a) Stress cloud map before optimization. (b) Stress cloud map after optimization. (c) Displacement cloud map before optimization. (d) Displacement cloud map after optimization.

Table 14 presents a comparison of the static performance of the upper cover under the bump + braking conditions. As evidenced by the data presented in Table 12, optimization of the upper cover of the battery box has resulted in a notable reduction in maximum stress, from 4.826 MPa to 4.614 MPa. This represents a 4.35% improvement in performance. Additionally, the maximum displacement has also decreased, from 0.399 mm to 0.149 mm, indicating a 62.66% enhancement in performance.

Under the bumpy + turning conditions, the static performance comparison cloud chart of the battery box upper cover before and after optimization is shown in Fig 41. In the stress cloud diagram, it can be seen that the stress distribution of the upper cover of the battery box has changed after optimization. In the displacement cloud diagram, it can be seen that the displacement distribution of the upper cover of the battery box is basically the same before and after optimization.

Table 15 presents a comparative analysis of the static performance of the upper cover under the specified bumpy + turning conditions. As illustrated in the Fig, following optimization, the maximum stress of the upper cover of the battery box under the bumpy + turning conditions decreased from 5.6 MPa to 4.553 MPa, representing an improvement of 18.70%. Similarly, the maximum displacement decreased from 0.398 mm to 0.15 mm, indicating a 62.31% enhancement in performance.

**Table 14. Comparison of the performance of the upper cover under the bump + braking conditions.**

| Performance indicator | Before optimization | After optimization | Improvement rate (%) |
|---|---|---|---|
| Stress (Mpa) | 4.826 | 4.614 | 4.35 |
| Displacement (mm) | 0.399 | 0.149 | 62.66 |

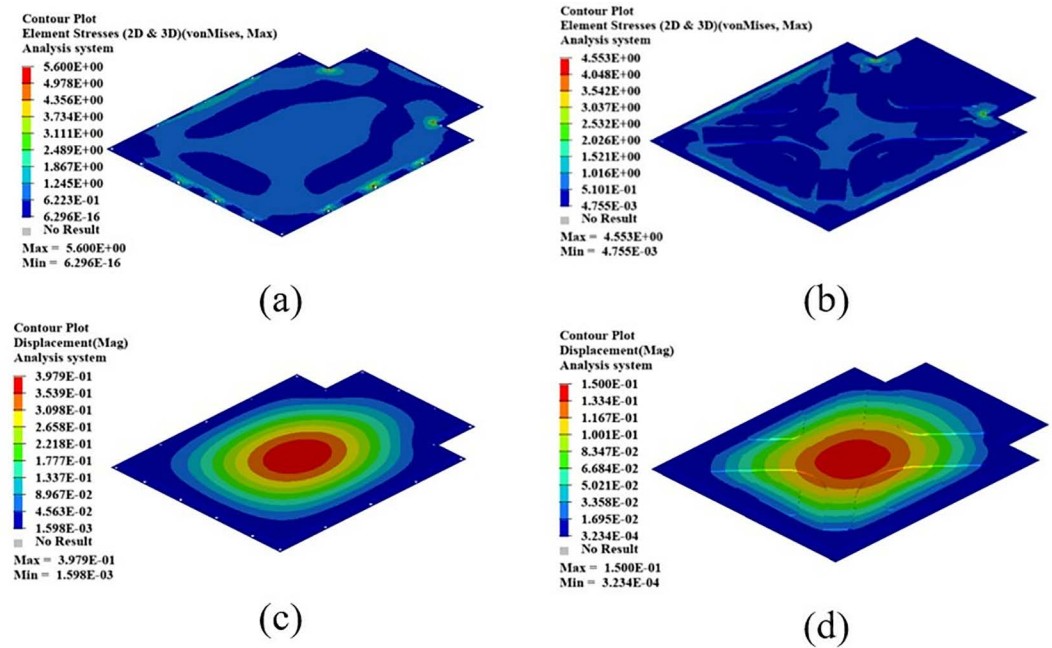

**Fig 41. Cloud diagram of upper cover performance under bumpy and turning conditions.** (a) Stress cloud map before optimization. (b) Stress cloud map after optimization. (c) Displacement cloud map before optimization. (d) Displacement cloud map after optimization.

**Table 15. Comparison of upper cover performance under bumpy + turning conditions.**

| Performance indicator | Before optimization | After optimization | Improvement rate (%) |
|---|---|---|---|
| Stress (Mpa) | 5.6 | 4.553 | 18.70 |
| Displacement (mm) | 0.398 | 0.15 | 62.31 |

**6.1.2 Comparison of free modal analysis of upper cover.** The free modal test of the SMC composite upper cover can verify the reliability of the established upper cover finite element model. In this test, rubber ropes are used to connect the SMC composite upper cover to simulate the free boundary of the battery box upper cover. Four hydraulic cranes are used to control the hanging height of the battery box upper cover to keep the battery box upper cover in a horizontal state during the test. The specific hanging method is shown in Fig 42. In this free modal test of the battery box upper cover, a force hammer is used as the excitation device. The signals of each sensor are collected through the signal acquisition software. The modal test natural frequency of the SMC composite material upper cover is obtained through analysis and processing. Several groups of experimental controls were carried out for this test to ensure the credibility of the free modal test results.The free modal test process of the SMC composite material upper cover is shown in Fig 42.

The comparison between the free modal test and finite element simulation of the SMC composite upper cover is shown in Table 16. Among them, the maximum error between the free modal test frequency and the simulated modal frequency is 3.15%. The possible reason is that there is a deviation between the real coordinate angle of the three-axis acceleration sensor and the modeling angle, which leads to a maximum error of 3.15% between the simulated modal frequency and the free modal test modal frequency, but meets the accuracy requirements. It can be seen that the established finite element model can better replace the actual battery box cover for modal analysis research, which further verifies the accuracy of the finite element model and the feasibility of the analysis method used.

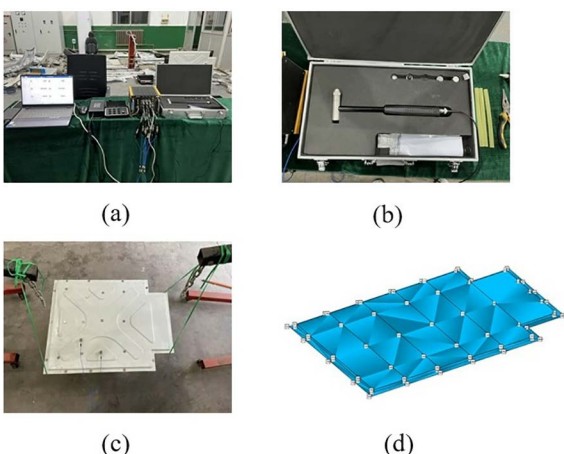

(a) (b)

(c) (d)

**Fig 42. Free modal test of SMC upper cover. (a)** Connection of test instrument **(b)** Test hammer equipment **(c)** Support method of upper cover **(d)** Selection of sampling points.

**Table 16. Comparison of simulation and test results of the first four modal frequencies of the upper cover.**

| Frequency order | Simulation frequency/Hz | Test frequency/Hz | Error % |
|---|---|---|---|
| 1 | 35.82 | 34.98 | 2.40 |
| 2 | 59.71 | 58.14 | 2.70 |
| 3 | 66.14 | 64.12 | 3.15 |
| 4 | 85.30 | 83.46 | 2.20 |

After shape optimization and size optimization, the lightweight design of the upper cover is completed. In order to verify the improvement of the modal performance by the optimization results, a free modal analysis was performed on the SMC upper cover according to the boundary conditions of the previous modal analysis conditions to explore its stiffness changes. The comparison of the first six free modal vibration shapes of the upper cover before and after optimization is shown in Fig 43.

Table 17 presents a comparison of the initial six natural frequencies of the upper cover, both before and after optimization. As evidenced by the comparison, the modal frequency exhibited a notable increase following optimization, with the first-order modal frequency rising from 20.98 Hz to 35.82 Hz, representing a 70.73% surge. This evidence substantiates the efficacy of the proposed optimization strategy for the upper cover of the battery box.

### 6.2 Comparison of battery box lower box performance before and after optimization

**6.2.1 Comparison of static analysis.** In conducting a performance analysis of the carbon fiber composite lower box, the working and boundary conditions are established in accordance with the findings of the preceding static analysis. This section presents a comparative and analytical assessment of the stress and displacement of the battery box under three static working conditions, both before and after optimization. The comparison of the static performance before and after optimization of the battery box lower box under bumpy conditions is shown in Fig 44. In the stress and displacement cloud diagram, it can be seen that the stress and displacement distribution of the lower box of the battery box have changed after optimization.

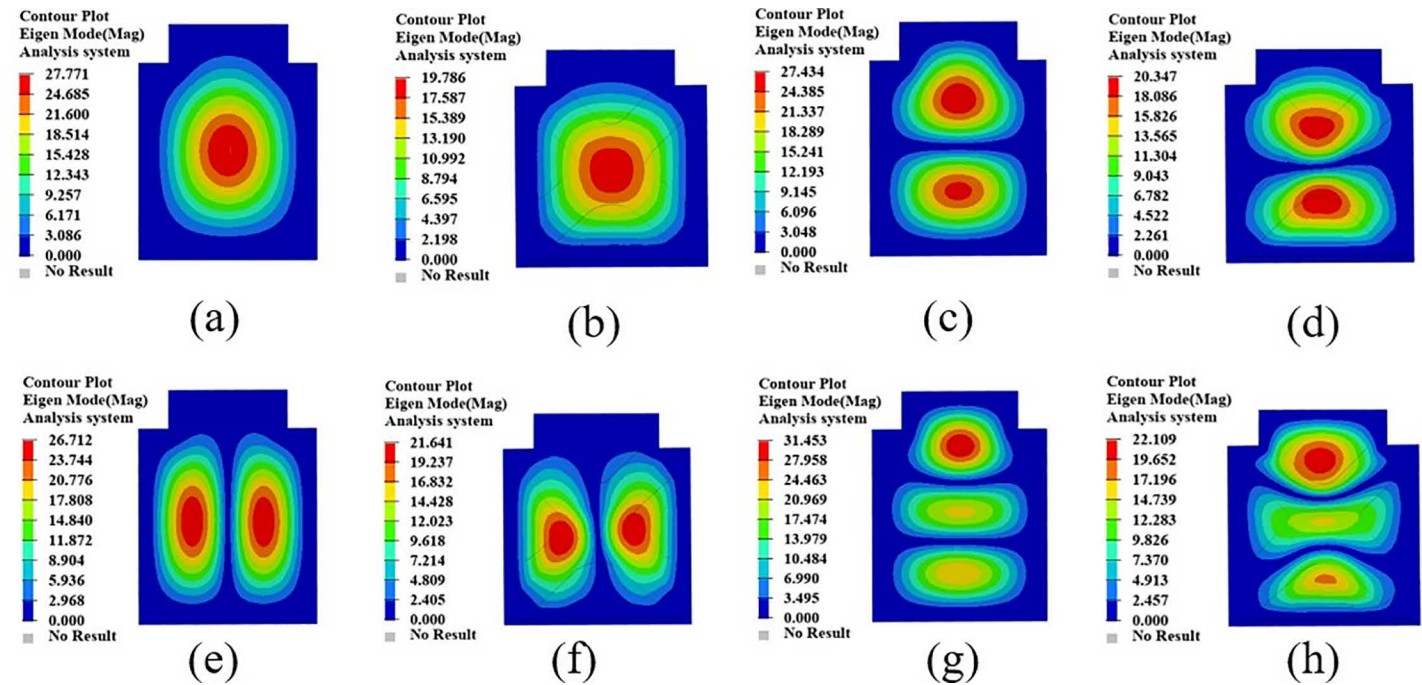

**Fig 43. Free mode cloud diagram of upper cover before and after optimization.** (a) The first vibration mode before optimization. (b) The first-order vibration mode after optimization. (c) The second-order vibration mode before optimization. (d) The second-order vibration shape after optimization. (e) The third-order vibration mode before optimization. (f) The third-order vibration mode after optimization. (g) The fourth-order vibration shape before optimization.(h) The fourth-order vibration mode after optimization.

**Table 17. Comparison of the first 6 modal frequencies of the upper cover.**

| Modal Order | Before optimization/Hz | After optimization/Hz | Rate of change (%) |
| --- | --- | --- | --- |
| 1 | 20.98 | 35.82 | 70.73 |
| 2 | 36.59 | 59.71 | 63.19 |
| 3 | 48.14 | 66.14 | 37.39 |
| 4 | 59.82 | 85.30 | 42.59 |
| 5 | 65.47 | 105.30 | 60.84 |
| 6 | 87.42 | 112.83 | 29.07 |

Table 18 shows the comparison of static performance of the lower box of the optimized battery box under bumpy conditions. The maximum stress experienced a reduction from 49.51 MPa to 47.41 MPa, representing a 4.24% improvement. The maximum displacement increased from the original 0.042 mm to 0.046 mm, representing a 9.52% improvement.

The comparison of the static performance of the front and rear lower boxs under the bump+braking conditions is shown in Fig 45.

Table 19 presents a comparative analysis of the static performance of the lower battery box under bump and braking conditions. As evidenced by the data presented in Table 16, optimization of the lower box of the battery box under bump+braking conditions has resulted in a notable reduction in maximum stress, from 27 to 24.21 MPa. The maximum stress decreased from 75 MPa to 24.21 MPa, representing an improvement rate of 12.76%. Similarly, the maximum displacement decreased from 0.0184 mm to 0.0168 mm, indicating an improvement rate of 8.7%.

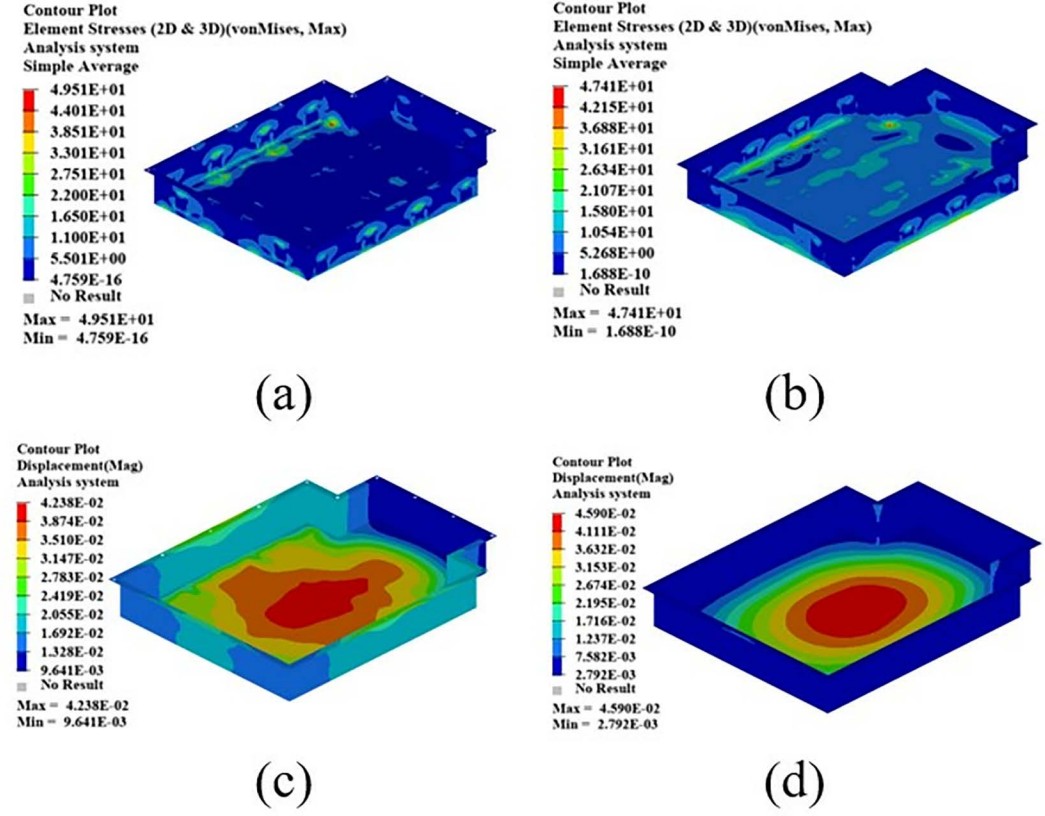

**Fig 44. Cloud diagram of lower box performance under bumpy conditions.** (a) Stress cloud map before optimization. (b) Stress cloud map after optimization. (c) Displacement cloud map before optimization. (d) Displacement cloud map after optimization.

**Table 18. Performance comparison of the lower box under bumpy conditions.**

| Performance indicator | Before optimization | After optimization | Improvement rate (%) |
|---|---|---|---|
| Stress (Mpa) | 49.51 | 47.41 | 4.24 |
| Displacement (mm) | 0.042 | 0.046 | −9.52 |

The optimized front and rear static performance cloud map of the lower box under the bumpy + turning conditions is shown in Fig 46. In the stress and displacement cloud diagram, it can be seen that the stress and displacement distribution of the lower box of the battery box have changed after optimization.

Table 20 presents a comparative analysis of the static performance of the lower box under the bump + turn conditions. As illustrated in the Fig, following the optimization of the upper cover of the battery box under the bump + turn conditions, the maximum stress decreased from 20.34 MPa to 18.42 MPa, representing a 9.44% improvement. The maximum displacement exhibited a reduction from 0.0173 mm to 0.0155 mm, representing a 10.40% improvement. The optimization results of the lower box of the carbon fiber composite battery box are remarkable.

**6.2.2 Comparison of modal analysis.** The lower box body is designed to be lightweight with the upper cover after the carbon fiber composite material layup design is optimized. In order to verify the improvement degree of modal performance brought by the optimization results, free modal analysis was performed on the carbon fiber composite lower

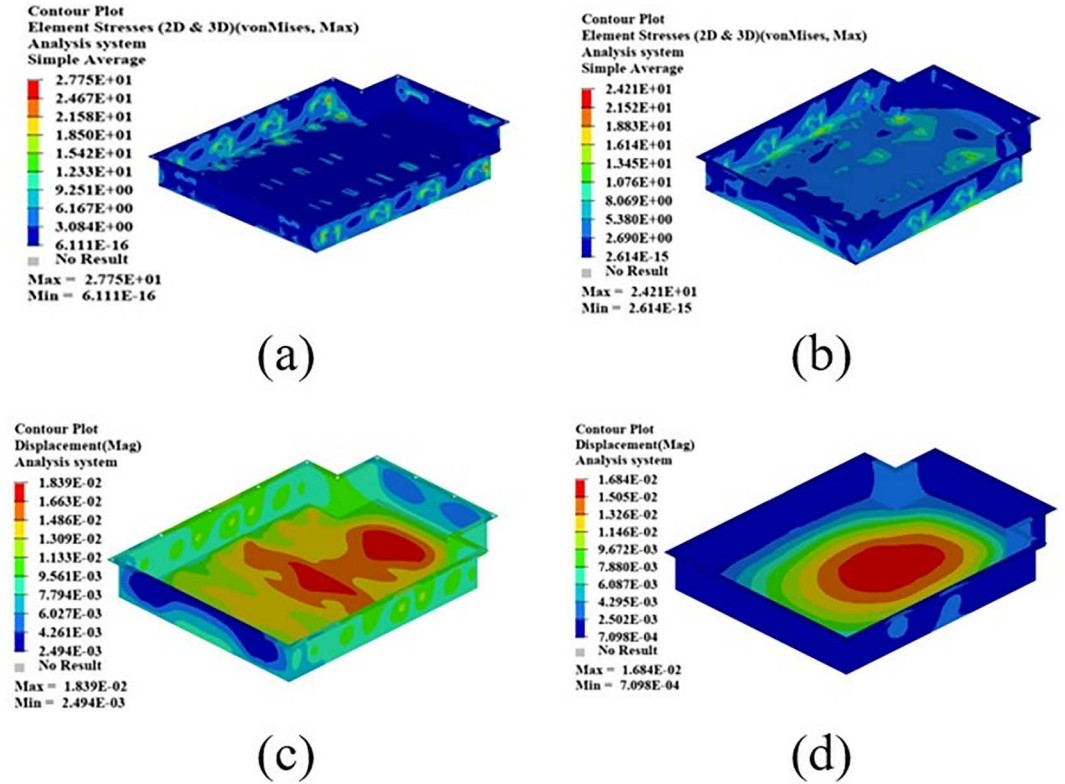

**Fig 45. Cloud diagram of lower box performance under bump＋braking conditions.** (a) Stress cloud map before optimization. (b) Stress cloud map after optimization. (c) Displacement cloud map before optimization. (d) Displacement cloud map after optimization.

**Table 19. Performance comparison of the lower box under bumpy＋braking conditions.**

| Performance indicator | Before optimization | After optimization | Improvement rate (%) |
|---|---|---|---|
| Stress (Mpa) | 27.75 | 24.21 | 12.76 |
| Displacement (mm) | 0.0184 | 0.0168 | 8.70 |

box separately according to the boundary conditions of the previous modal analysis condition to explore its stiffness improvement effect. The comparison of the first six free mode vibration shapes of the upper cover before and after optimization is shown in Fig 47.

The comparison of the first six modal frequencies of the lower box before and after optimization is shown in Table 21.

Through comparison, it can be seen that the modal frequency of the optimized battery box lower case is significantly improved, the first-order modal frequency of the composite lower case is greatly improved from 23.28 to 60.24 Hz, the first-order modal performance is improved by 61.35%; the second-order modal frequency is changed from 42.32 Hz to 152.90 Hz, the second-order modal performance is improved by 73.32%, and the rest of the modal frequencies of the other orders can be improved, the optimization result of the carbon fiber lower case is remarkable. The optimization results of carbon fiber composite lower case are remarkable.

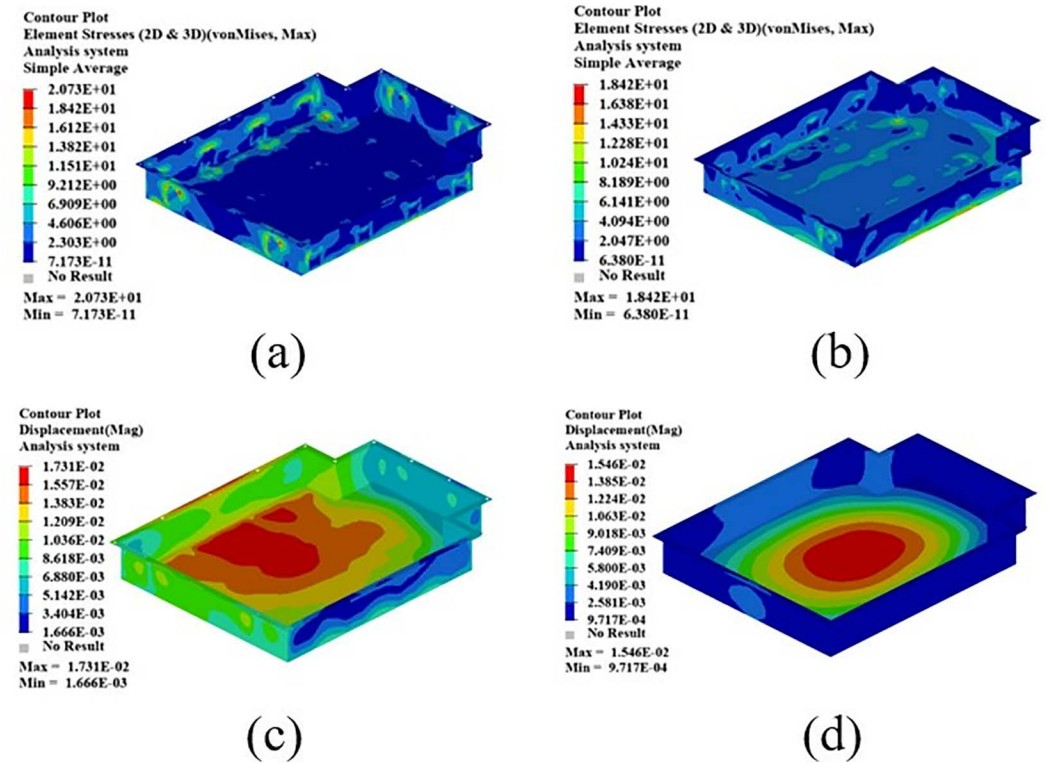

**Fig 46. Cloud diagram of lower box performance under bumpy and turning conditions.** (a) Stress cloud map before optimization. (b) Stress cloud map after optimization. (c) Displacement cloud map before optimization. (d) Displacement cloud map after optimization.

**Table 20. Performance comparison of the lower box under bumpy + turning conditions.**

| Performance indicator | Before optimization | After optimization | Improvement rate (%) |
|---|---|---|---|
| Stress (Mpa) | 20.34 | 18.42 | 9.44 |
| Displacement (mm) | 0.0173 | 0.0155 | 10.40 |

### 6.3 Comparison of battery box assembly performance

**6.3.1 Comparison of static analysis.** The SMC upper cover and the carbon fiber composite lower box are bonded by adhesive. The material parameters of the structural adhesive are shown in Table 22. the Young's modulus of the structural adhesive is 1190 MPa, Poisson's ratio is 0.41, and density is 1.15 g/cm³.

According to the previous static analysis working condition boundary conditions, this section compares and analyzes the stresses and displacements of the battery box assembly before and after optimization under three static working conditions, and the performance cloud diagrams are shown in Fig 48.

Table 23 shows the comparison of the static mechanical performance of the battery box assembly under bumpy conditions. From the Fig, it can be seen that the maximum stress of the battery box of the optimized battery box under bumpy condition decreases from 126.935Mpa to 20.821Mpa, which improves the performance by 83.60%; and the maximum displacement decreases from 1.215 mm to 0.452 mm, which improves the performance by 62.80%.

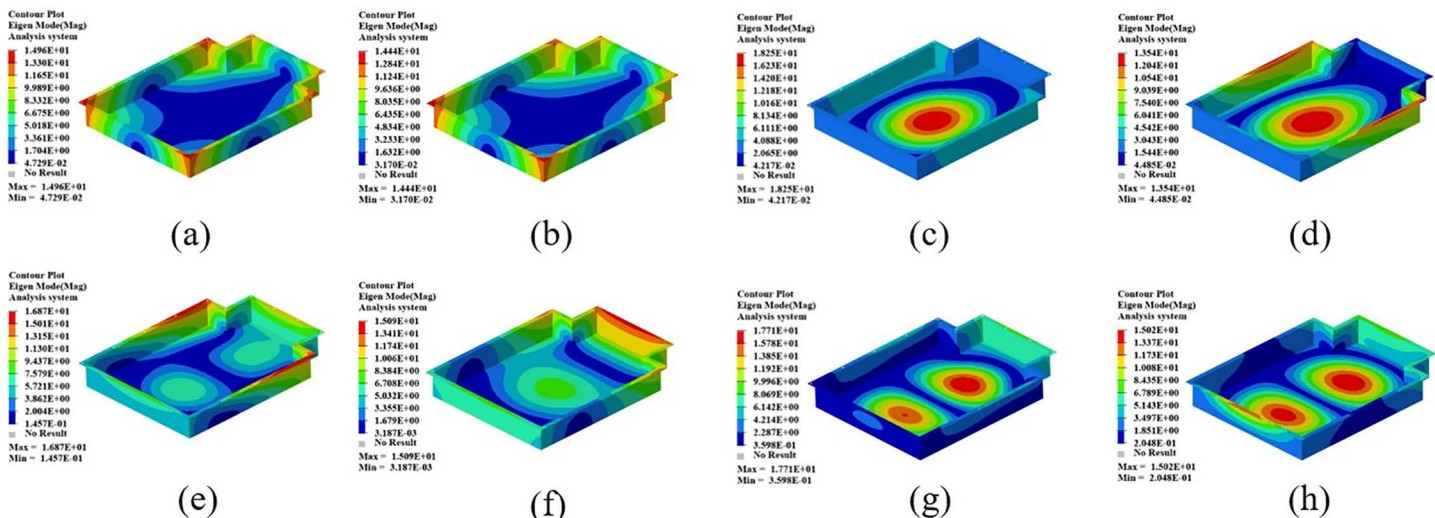

**Fig 47. Free mode cloud of lower box before and after optimization.** (a) The first vibration mode before optimization. (b) The first-order vibration mode after optimization. (c) The second-order vibration mode before optimization. (d) The second-order vibration shape after optimization. (e) The third-order vibration mode before optimization. (f) The third-order vibration mode after optimization. (g) The fourth-order vibration shape before optimization. (h) The fourth-order vibration mode after optimization.

**Table 21. Comparison of the first six orders of modal frequencies of the lower box.**

| Modal Order | Before optimization/Hz | After optimization/Hz | Rate of change (%) |
|---|---|---|---|
| 1 | 23.28 | 60.24 | 61.35 |
| 2 | 42.32 | 152.90 | 72.32 |
| 3 | 68.34 | 162.56 | 57.96 |
| 4 | 71.60 | 220.34 | 67.50 |
| 5 | 87.07 | 241.53 | 63.95 |
| 6 | 109.72 | 320.68 | 65.79 |

**Table 22. Structural adhesive material parameters.**

| Modulus of elasticity | Poisson's ratio | Densities |
|---|---|---|
| 1190Mpa | 0.41 | 1.15 g/cm³ |

A comparison of the static mechanical performance of the battery box assembly before and after optimization is shown in Fig 49 under bumpy + braking conditions.

Table 24 shows the comparison of the static performance of the battery box assembly under bump + braking conditions. It can be seen from the Fig that after optimization under bump + braking conditions, the maximum stress of the battery box upper cover is reduced from 43.787Mpa to 8.313Mpa, and the performance is improved by 81.01%; the maximum displacement is reduced from 0.405 mm to 0.149 mm, and the performance is improved by 63.21%.

Under the bumpy + turning conditions, the static performance comparison of the battery box assembly before and after optimization is shown in Fig 50.

Table 25 shows the comparison of the static mechanical performance of the battery box assembly under bumps and turns, and it can be seen from the Fig that after optimizing the battery box assembly under bumps and turns, the

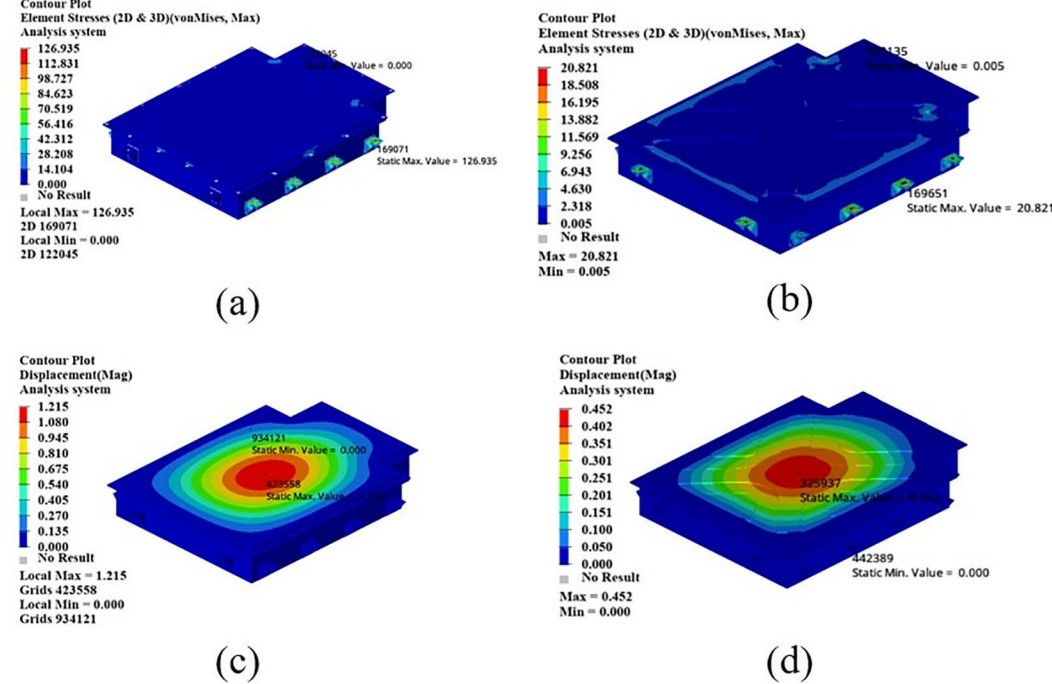

**Fig 48. Battery box assembly performance cloud under bumpy conditions.** (a) Stress cloud map before optimization. (b) Stress cloud map after optimization. (c) Displacement cloud map before optimization. (d) Displacement cloud map after optimization.

**Table 23. Comparison of battery box assembly performance under bumpy conditions.**

| Performance indicators | Before optimization | After optimization | Improvement rate (%) |
|---|---|---|---|
| Stress (Mpa) | 126.935 | 20.821 | 83.60 |
| Displacement (mm) | 1.215 | 0.452 | 62.80 |

maximum stress decreases from 5.6Mpa to 4.553Mpa, which improves the performance by 84.31%, and the maximum displacement decreases from 0.407 mm to 0.15 mm, which improves the performance by 63.14%.

From the above analysis, it can be seen that the joint between the SMC upper cover and the carbon fiber composite lower box is replaced by the bonding of the original bolts, which reduces the stress concentration at the bolt holes. Due to the material replacement, the overall mass of the box is reduced, and from the overall stress performance after optimization, the overall stress of the battery box is significantly reduced, and the performance is greatly improved.

**6.3.2 Comparison of modal analysis.** After a series of design optimizations, the upper cover and lower box of the battery box complete the lightweight design of the battery box assembly. In order to verify the improvement degree of modal performance brought by the optimization results, a free modal analysis of the composite battery box assembly was carried out according to the previous modal analysis working condition boundary conditions. The comparison of the first four free mode vibration shapes of the box assembly before and after optimization is shown in Fig 51.

Comparison of the first six orders of modal frequencies of the composite battery box assembly before and after optimization is shown in Table 26, in which the first order modal frequency is changed from 31.59 Hz to 47.39 Hz, which is a performance improvement of 33.34%, and after optimization of the battery box in the second order modal frequency is significantly increased, the reason for the increase in modal frequency is due to the change in the characteristics of the

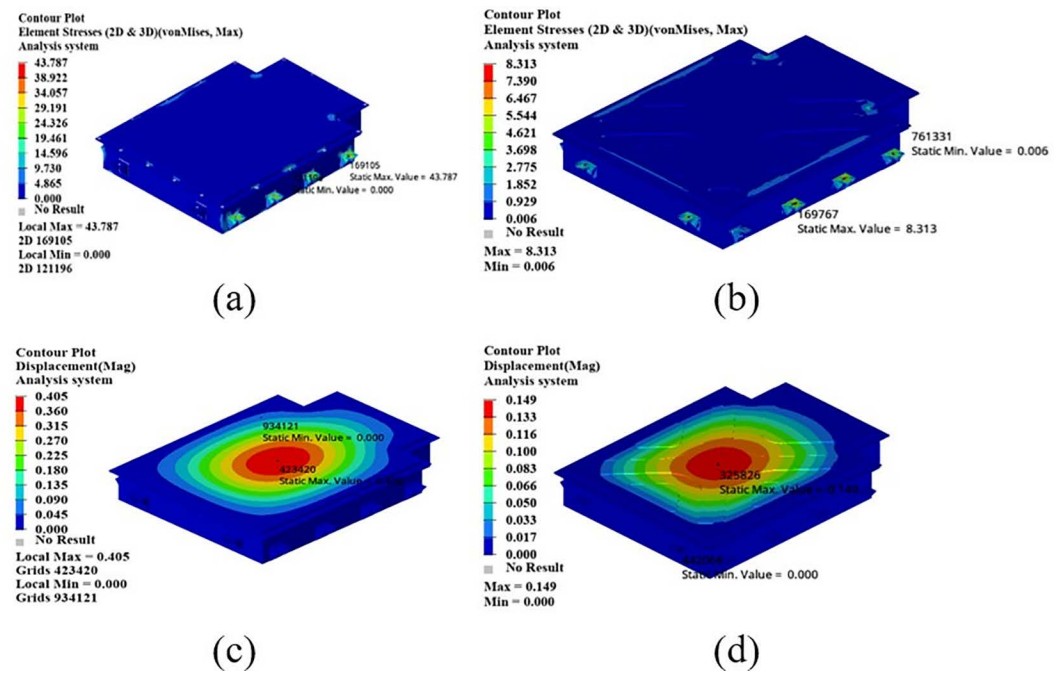

**Fig 49. Battery box assembly performance cloud under bump + braking conditions.** (a) Stress cloud map before optimization. (b) Stress cloud map after optimization. (c) Displacement cloud map before optimization. (d) Displacement cloud map after optimization.

**Table 24. Comparison of battery box assembly performance under bumpy + braking conditions.**

| Performance indicators | Before optimization | After optimization | Improvement rate (%) |
|---|---|---|---|
| Stress (Mpa) | 43.787 | 8.313 | 81.01 |
| Displacement (mm) | 0.405 | 0.149 | 63.21 |

material used, i.e., the replacement of carbon fiber composite material. Comparison of the vibration patterns shows that the first 4 orders of the modal vibration patterns are similar, and the 5th order resonance region occurs on the lower case after optimization. The optimization further improves the safety performance of the automotive battery case as the modal frequency of the battery case increases.

## 6.4 Dynamic performance analysis of battery box assembly

**6.4.1 Random vibration conditions.** The optimized model is used to perform performance analysis and verification of random vibration conditions of the battery box according to relevant standards to ensure that the product meets the use requirements and further verify the feasibility of the optimization method. The vibration excitation that the battery box is subjected to mainly comes from the vibration caused by uneven road surface, which is transmitted through the connection between the vehicle and the battery box. Due to the uncertainty of road conditions, the excitation of the battery box is also random. Therefore, the study of random vibration is of great significance for the performance evaluation of the battery box assembly. Through random vibration analysis, the structural response can be effectively evaluated and the fatigue life can be predicted. According to the GB 38031–2020 regulatory standard, a 21-hour vibration test will be carried out in each axis during the test to observe whether there are abnormal phenomena such as leakage, fire or damage. If the above occurs, the battery box is deemed to be non-compliant with the standard requirements. This article focuses on the rupture

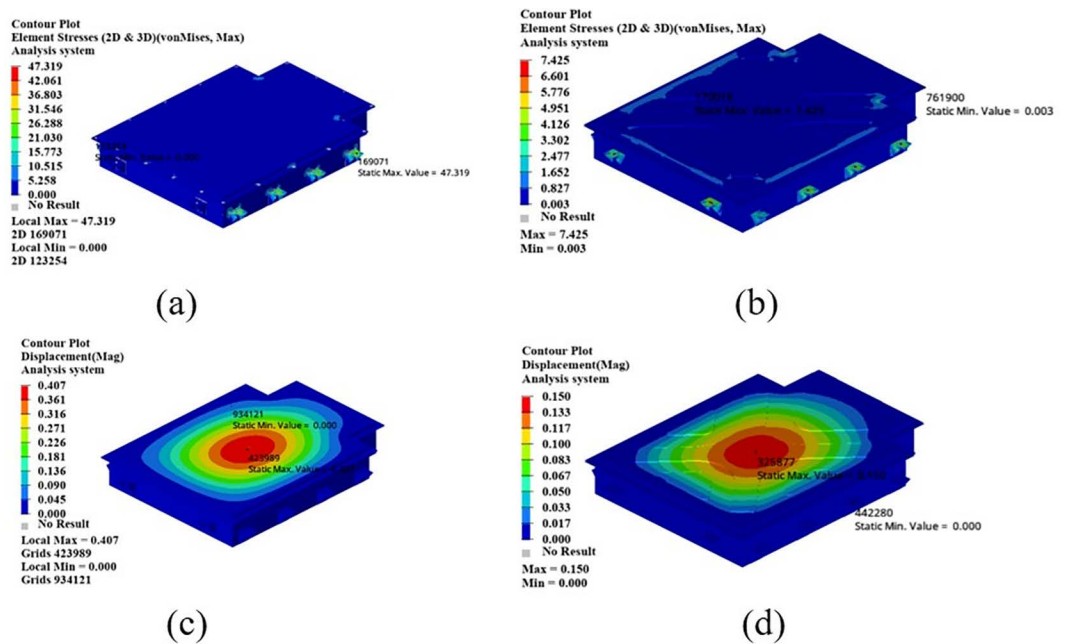

**Fig 50. Battery box assembly performance cloud chart for bumpy and turning conditions.** (a) Stress cloud map before optimization. (b) Stress cloud map after optimization. (c) Displacement cloud map before optimization. (d) Displacement cloud map after optimization.

**Table 25. Comparison of battery box assembly performance under bumpy and turning conditions.**

| Performance indicators | Before optimization | After optimization | Improvement rate (%) |
|---|---|---|---|
| Stress (Mpa) | 47.319 | 7.425 | 84.31 |
| Displacement (mm) | 0.407 | 0.15 | 63.14 |

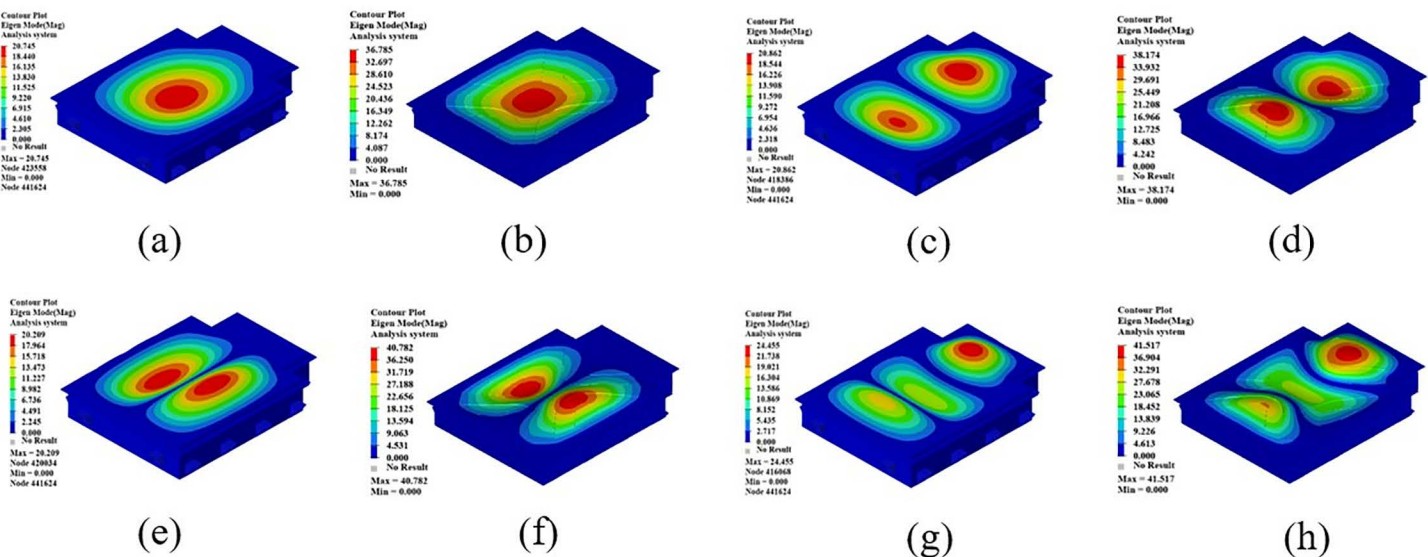

**Fig 51. Comparison cloud of free modes of box assembly.** (a) The first vibration mode before optimization. (b) The first-order vibration mode after optimization. (c) The second-order vibration mode before optimization. (d) The second-order vibration shape after optimization. (e) The third-order vibration mode before optimization. (f) The third-order vibration mode after optimization. (g) The fourth-order vibration shape before optimization.(h) The fourth-order vibration mode after optimization.

**Table 26.  Comparison of the first six orders of modal frequencies of the box assembly.**

| Modal Order | Before optimization/Hz | After optimization/Hz | Rate of change (%) |
|---|---|---|---|
| 1 | 31.59 | 47.39 | 33.34 |
| 2 | 38.79 | 86.31 | 55.06 |
| 3 | 50.42 | 96.21 | 47.59 |
| 4 | 65.37 | 135.13 | 51.62 |
| 5 | 73.42 | 148.25 | 50.48 |
| 6 | 89.37 | 158.84 | 43.74 |

of the box during the test. The power spectral density (PSD) data decomposition of each axis is detailed in the Tables 27–29 below.

The random vibration analysis results in each direction are shown in Figs 52–54. As can be seen from the figure, the random vibration loading point response curves in the X, Y and Z directions are consistent with the national standard input loading curve, indicating that the random vibration simulation analysis results are reliable and the model has good accuracy.

From the RMS stress cloud diagrams of Fig 52 and 53, it can be seen that the maximum RMS stress values in the X and Y directions are 1.24Mpa and 0.28Mpa respectively, which appear around the upper right corner of the connection between the lower box and the side wall, which is much smaller than the yield limit of each material, and the overall stress is at a low level.From the RMS stress cloud diagram in Fig 54, it can be seen that the maximum RMS stress value in the Z

**Table 27.  X-axis PSD values.**

| Frequency (Hz) | PSD($g^2$/Hz) | PSD($(m/s^2)^2$/Hz) |
|---|---|---|
| 5 | 0.0125 | 1.2 |
| 10 | 0.03 | 2.89 |
| 20 | 0.03 | 2.89 |
| 200 | 0.00025 | 0.02 |
| RMS | 0.96g | 9.42 m/s$^2$ |

**Table 28.  Y-axis PSD values.**

| Frequency (Hz) | PSD($g^2$/Hz) | PSD($(m/s^2)^2$/Hz) |
|---|---|---|
| 5 | 0.01 | 0.96 |
| 10 | 0.015 | 1.44 |
| 20 | 0.015 | 1.44 |
| 50 | 0.01 | 0.96 |
| 200 | 0.0004 | 0.04 |
| RMS | 0.95g | 9.32 m/s$^2$ |

**Table 29.  Z-axis PSD values.**

| Frequency (Hz) | PSD($g^2$/Hz) | PSD($(m/s^2)^2$/Hz) |
|---|---|---|
| 5 | 0.05 | 4.81 |
| 10 | 0.06 | 5.77 |
| 20 | 0.06 | 5.77 |
| 200 | 0.0008 | 0.08 |
| RMS | 1.44g | 14.13 m/s$^2$ |

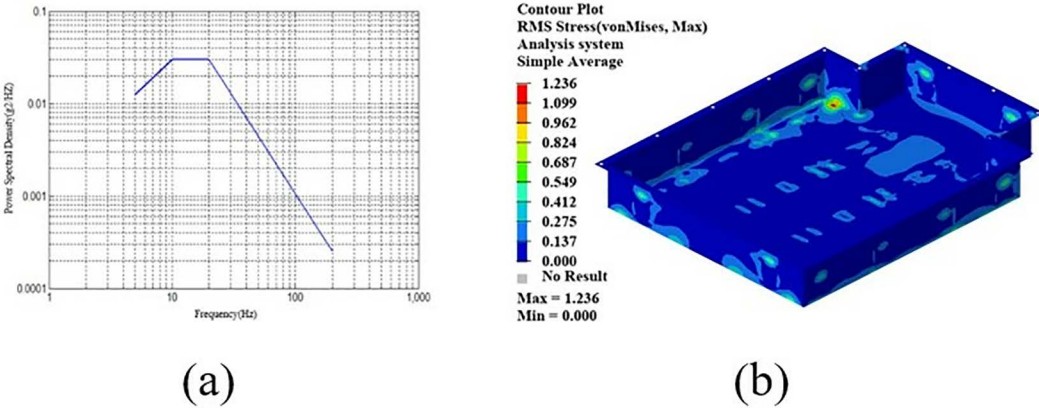

**Fig 52. X-direction random vibration. (a)** Loading point response curve **(b)** RMS stress cloud diagram.

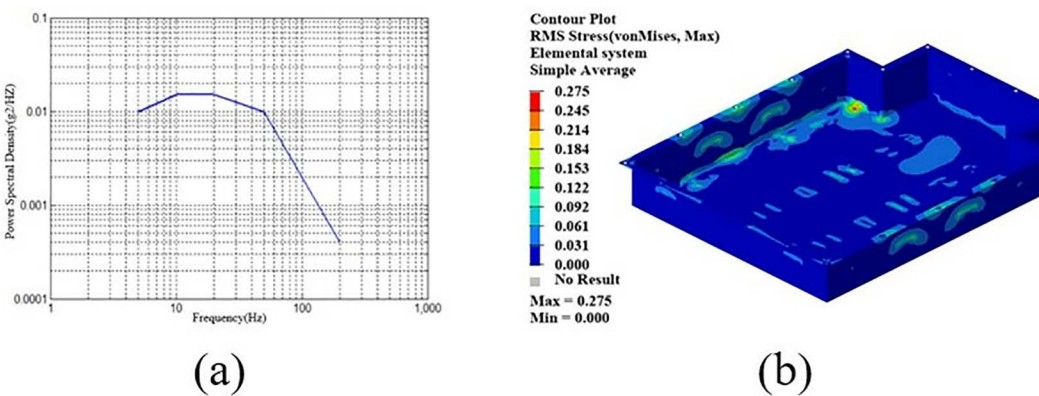

**Fig 53. Y-direction random vibration. (a)** Loading point response curve **(b)** RMS stress cloud diagram.

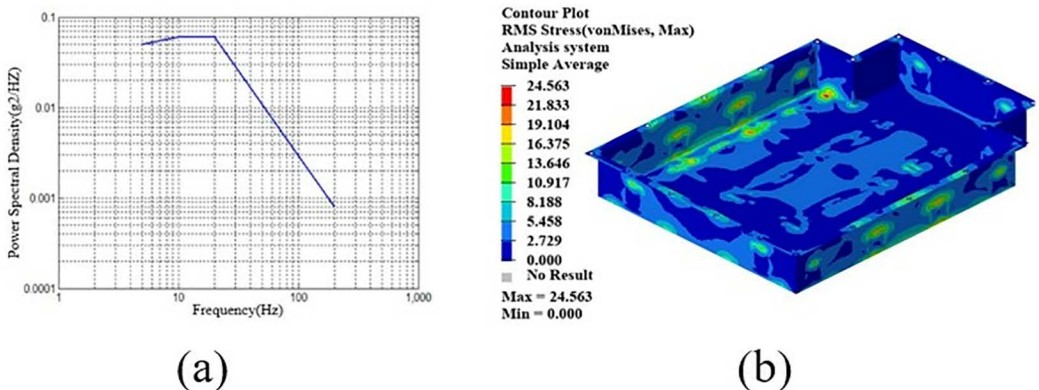

**Fig 54. Z-direction random vibration. (a)** Loading point response curve **(b)** RMS stress cloud diagram.

direction is 24.56MPa, which occurs around the connection between the lifting lug and the side panel, and is much smaller than the yield limit of each material. The overall stress is higher than the RMS stress values in the X and Y directions, and lower than the yield limit of each material. The overall stress is at a relatively low level.

**6.4.2 Battery box extrusion simulation.** The battery box is a key component of new energy vehicles. Its extrusion may cause damage to the internal battery cells, circuit control systems, and high and low voltage wiring harnesses, which may affect the power system at best and cause fire or explosion at worst. During the simulation process, an extrusion table model needs to be established, which consists of two surfaces, as shown in Fig 55. The battery box is placed on the table, and a cylinder is used to apply extrusion along the X-axis and Y-axis directions until the pressure reaches 100KN or the deformation reaches 30% of the box size. The simulation analysis uses LS-DYNA software. First, the finite element model of the battery box is imported into the LS-DYNA module of Hypermesh, all two-dimensional units are set to self-contact, and the pre-built extrusion bench model is loaded, as shown in Fig 55.

Create contact between the battery box and the extrusion stand. The stand surface is a fixed part, and it is in fixed surface contact with the battery box; the semi-cylinder is a moving part, and it is in moving surface contact with the battery box, as shown in the following Fig 56.

The extrusion bench is made of rigid material with a thickness of 1 mm. The material parameters of the moving parts and fixed parts are set separately, and the fixed parts are fully constrained with six degrees of freedom. The cylindrical moving parts are constrained with six degrees of freedom of rotation and five degrees of freedom of translation, and only X-direction movement is allowed.

This simulation uses forced displacement loading to avoid the initial velocity being reduced due to extrusion release, which affects the working condition requirements. The displacement curve is applied through the boundary forced displacement load, and the displacement is set to 200 mm within 20ms. After completing the material properties and load

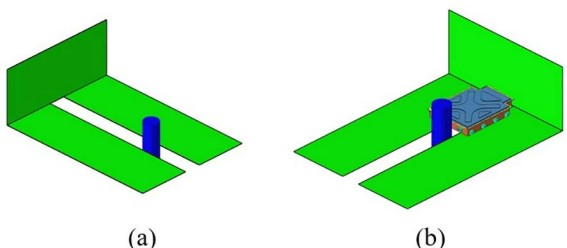

(a)                    (b)

**Fig 55. Finite element model of extrusion simulation conditions. (a)** Finite element model of extrusion platform **(b)** Model diagram of extrusion conditions.

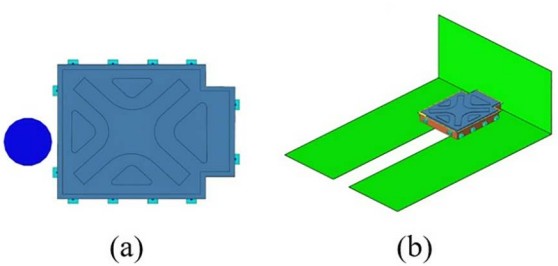

(a)                    (b)

**Fig 56. Contact conditions in extrusion simulation.** (a) moving surface contact (b) fixed surface contact.

condition settings, the extrusion simulation model in the X-axis direction can be established.When modeling in the Y-axis direction, the battery box needs to be rotated 90° along the Z axis and placed on the test bench. The cylindrical moving part still moves in the X-axis direction without adjusting the direction. After completing the extrusion modeling in the X and Y directions, the model is imported into LS-DYNA for analysis.

According to regulatory requirements, when the extrusion force reaches 200kN, the battery cells, circuit control system and high and low voltage harnesses inside the battery box must remain intact. When simulating the extrusion condition, a uniform forced displacement is applied to the moving part within 0.02 seconds to obtain the extrusion force-displacement curve. Simulations are performed in the X-axis and Y-axis directions respectively, and the extrusion condition in the X-axis direction is analyzed first. The extrusion analysis results are obtained using LS-DYNA, and the extrusion force-time and extrusion force-displacement curves are displayed in Hyperview. The results are shown in Fig 57.

According to the curve data, when the extrusion force reaches 200kN, the displacement of the cylinder is 50 mm, and the corresponding extrusion time is 0.01ms. Import the LS-DYNA simulation results into Hyperview, and extract the stress and displacement cloud diagrams. The extrusion and intrusion of the battery box at 0.01ms can be clearly observed, as shown in Fig 58. The maximum deformation of the battery pack structure at the moment when the extrusion force in the X direction reaches 200kN is 10.60 mm, and the position appears at the connection between the upper and lower boxes

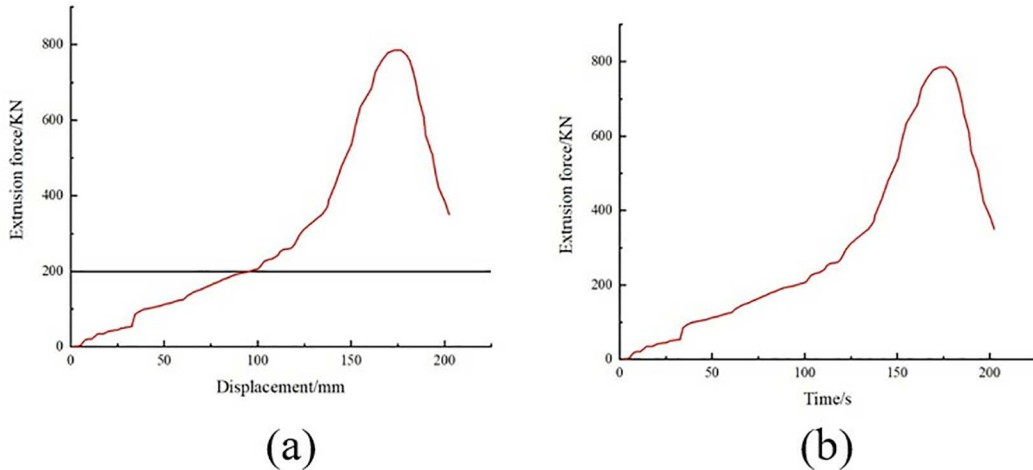

**Fig 57. Extrusion curve in the X-axis direction. (a)** Forced displacement-stress diagram **(b)** Time-stress relationship diagram.

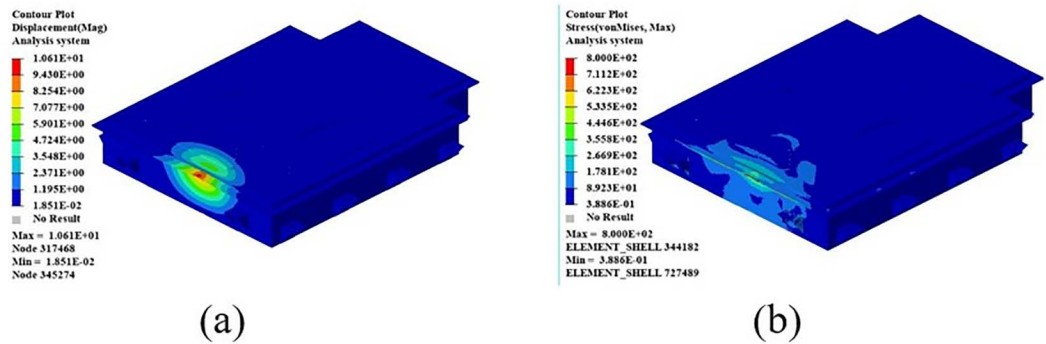

**Fig 58. Extrusion cloud map in the x-axis direction.** (a) displacement cloud map (b) stress cloud map.

at the front end. The distance between the connection and the battery module is about 100 mm, and the deformation is much smaller than this value, so the battery module will not be squeezed.

Then, the extrusion simulation condition of the battery box in the Y-axis direction is analyzed. The extrusion condition analysis results in LS-Dyna, the extrusion force and action time, and the extrusion force and displacement curves are displayed in Hyperview, and the analysis results can be obtained as shown in Fig 59.

According to the curve analysis, when the extrusion force reaches 200kN, the displacement of the cylinder is 7 mm, 10 mm, and 76 mm, and the corresponding extrusion time is 0.0014ms, 0.002ms, and 0.015ms. Comprehensively comparing the three sets of data, when the displacement reaches 76 mm and the action time is 0.015ms, the battery box is most affected. By reading the stress and displacement cloud map of the extrusion condition analysis of LS-DYNA through Hyperview, the extrusion intrusion of the battery box at 0.015ms can be clearly observed, as shown in Fig 60.

As for the Y direction, the maximum deformation of the battery pack structure when the extrusion force reaches 200kN is 11.05 mm, which occurs at the connection between the upper and lower boxes on the side. The minimum distance between the side of the battery pack box and the battery module is 20 mm, and the deformation is much smaller than

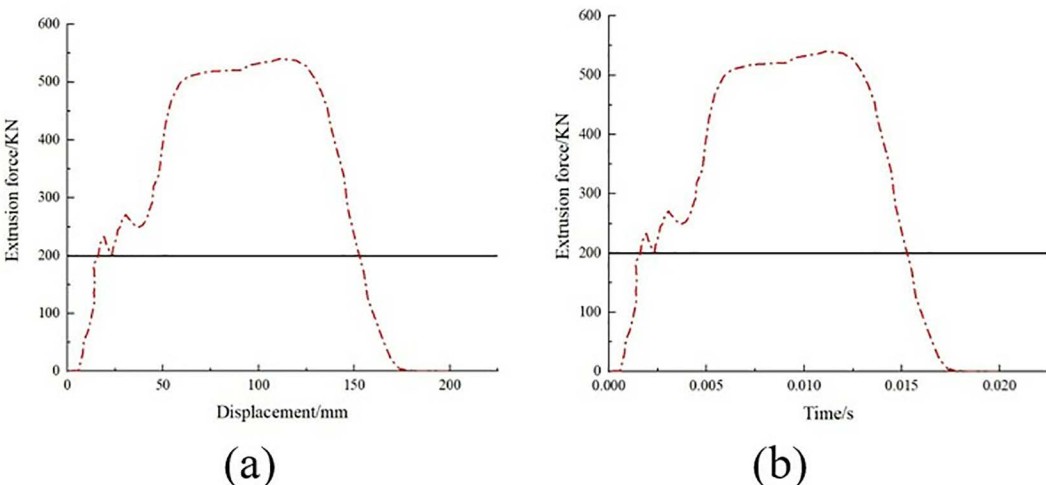

**Fig 59. Y-axis extrusion curve.** (a) forced displacement-stress diagram (b) time-stress relationship diagram.

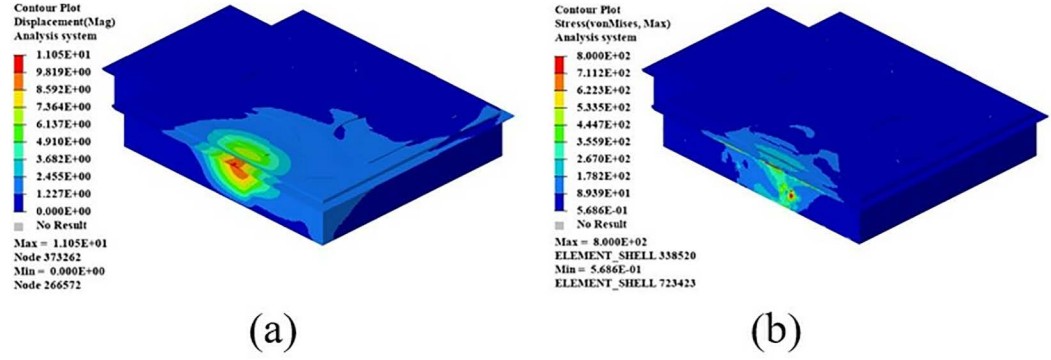

**Fig 60. Y-axis extrusion cloud map.** (a) displacement cloud map (b) stress cloud map.

this value, so the battery module will not be squeezed. Based on the extrusion results in the X and Y directions, it can be judged that the carbon fiber composite material used has excellent specific stiffness performance. When the extrusion force reaches 200kN, the deformation of the box structure is at a low level, which meets the standard regulations and usage requirements.

## 7. Conclusion

In order to improve the overall lightweight level and safety performance of the automotive battery box. This paper takes material replacement and structural optimization design as the means, based on digital twin technology and combined with the improved PSO-BFO algorithm, proposes a twin optimization method for fiber reinforced composite material layup design. A finite element model of the original aluminum alloy battery box was established, and the correctness of the simulation model was verified by combining modal tests and comparative simulation analysis, laying the foundation for subsequent battery box modifications. The original battery box was modified and a finite element model was established. On this basis, the battery box is simulated and analyzed under three working conditions: bumping, bumping+turning, bumping+braking, and constrained modal conditions. The SMC composite material box upper cover was designed and optimized using morphology optimization and size optimization methods. An improved PSO-BFO algorithm is proposed, which integrates the digital twin technology to construct a multi-level and adaptive correction optimization method for the carbon fiber composite lower box. The results show that the first-order mode of the new battery box assembly increased by 33.34% under the condition of 40.63% weight reduction of the battery box, and the performance of each part was improved under the three working conditions of bumping, bumping+turning, and bumping+braking. The complex dynamic multi-working condition performance verification of the optimized model was carried out, and the analysis results showed that the optimized hybrid material battery box structure basically met the use requirements, further proving the effectiveness of the composite material battery box assembly design scheme and optimization strategy proposed in this paper.

(1) A finite element model of the aluminum alloy original battery box assembly was constructed, and the accuracy of the established model was verified by performing free modal simulation analysis and modal tests on the original battery box, providing a reliable modeling environment and analysis method for subsequent finite element modeling of new battery boxes.

(2) The original battery box is redesigned and a new battery box model is reconstructed. The static performance indicators related to the battery box, such as bumpy conditions, bumpy+left/right sharp turn conditions, bumpy+emergency braking conditions and constrained modes, were analyzed. The finite element analysis parameters of carbon fiber composites and SMC composites were constructed through experiments, and the simulation analysis foundation of the battery box assembly under various conditions based on performance-driven optimization was established.

(3) For the upper cover of the battery box with simpler load conditions, consider replacing the material with SMC composite material. The upper cover of the battery box is jointly optimized using morphology optimization and size optimization. Taking the manufacturing process into consideration, the thickness of the upper cover is determined to be 2 mm. After optimizing the design of the upper cover, the mass was reduced by 35.6%. Under the bumpy condition, bumpy+braking condition, and bumpy+turning condition, the displacement improvement rates of the battery box upper cover were 62.11%, 62.66%, and 62.31%, respectively. The maximum stress improvement rates of the battery box upper cover were 7.95%, 4.35%, and 18.70%, respectively. Its first-order modal frequency increased from 20.98 Hz to 35.82 Hz, and its performance improved by 70.73%.

(4) The carbon fiber composite material layup optimization of the lower box of the new battery box was carried out, and the free size optimization, layup block cutting and size optimization methods were carried out based on this, and the

layup thickness, layup shape and number of layup layers of the lower box of the new battery box were determined. The continuous variable discretization and rounding strategy was adopted to obtain the discrete ply numbers of each laying angle of the lower box of the composite battery box. The digital twin structural optimization system is combined with the improved PSO-BFO algorithm to optimize the overall ply sequence of the lower box of the carbon fiber composite battery box. The optimal ply scheme for the lower box of the carbon fiber composite material is determined. The thickness of a single ply is 0.125, the number of plies is 30, and the ply sequence is [90/90/0/0/90/0/0/90/45/-45/0/45/-45/45/-45]s. The optimization results of the carbon fiber composite lower box of the battery box show that under the bumpy working condition, the bumpy + braking working condition and the bumpy + turning working condition, the stress is improved by 4.24%; 12.76%; 9.44% respectively, and the displacement is increased by −9.52%; 8.7%; 10.40% respectively. The first-order modal frequency of the composite lower box is greatly increased from 23.28 to 60.24 Hz, and the first-order modal performance is improved by 61.35%.

(5) A comparative analysis was conducted on the composite battery box assembly built based on the multi-level optimization method. The results show that: while meeting the safety and static performance requirements of the vehicle, the composite battery box assembly reduces weight compared to the original metal battery box assembly. 40.63%, the lightweight effect is significant. Under bumpy conditions, bumpy + braking conditions, and bumpy + turning conditions, the stress of the composite battery box assembly improved by 83.60%, 81.01%, and 84.31%, respectively, and the displacement increased by 62.80%, 63.21%, and 63.14%, respectively. Among them, the first-order mode frequency changes from 31.59 Hz to 47.39 Hz, the performance is improved by 33.34%, the static stiffness performance is significantly enhanced.At the same time, the optimized model was verified for complex dynamic multi-condition performance, covering random vibration and extrusion conditions. The analysis results show that the optimized composite battery box structure basically meets the use requirements and further improves the safety performance of the automotive battery box.

## Supporting information

**S1 File. Original battery box finite element model data.**
(TXT)

**S2 File. New battery box finite element model data.**
(TXT)

**S3 File. Finite element model of new battery box upper cover.**
(TXT)

## Author contributions

**Conceptualization:** Dongzhen Lu, Shuai Zhang, Zhao Li.

**Data curation:** Dongzhen Lu, Shuai Zhang, Jiufeng Chen, Feng Xiong.

**Formal analysis:** Dongzhen Lu, Jiufeng Chen.

**Funding acquisition:** Shuai Zhang, Zhao Li.

**Methodology:** Dongzhen Lu, Shuai Zhang, Zhao Li.

**Project administration:** Shuai Zhang, Zhao Li, Feng Xiong.

**Software:** Dongzhen Lu, Jiufeng Chen.

**Supervision:** Dongzhen Lu, Zhao Li.

**Validation:** Feng Xiong, Jinlong Xiao.

**Visualization:** Yuzhuo Zhang.

**Writing – original draft:** Dongzhen Lu, Shuai Zhang.

**Writing – review & editing:** Dongzhen Lu, Shuai Zhang, Yuzhuo Zhang.

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
