## [Decision Letter · Decision Letter 0]

Dear Dr. Zhang,

Thank you for submitting your manuscript to PLOS ONE. After careful consideration, we feel that it has merit but does not fully meet PLOS ONE’s publication criteria as it currently stands. Therefore, we invite you to submit a revised version of the manuscript that addresses the points raised during the review process.

We look forward to receiving your revised manuscript.

Kind regards,

Khalil Abdelrazek Khalil, Ph.D.

Academic Editor

PLOS ONE

**Journal Requirements:**

1. When submitting your revision, we need you to address these additional requirements. Please ensure that your manuscript meets PLOS ONE's style requirements, including those for file naming. The PLOS ONE style templates can be found at https://journals.plos.org/plosone/s/file?id=wjVg/PLOSOne_formatting_sample_main_body.pdf and https://journals.plos.org/plosone/s/file?id=ba62/PLOSOne_formatting_sample_title_authors_affiliations.pdf 2. Please note that PLOS ONE has specific guidelines on code sharing for submissions in which author-generated code underpins the findings in the manuscript. In these cases, we expect all author-generated code to be made available without restrictions upon publication of the work. Please review our guidelines at https://journals.plos.org/plosone/s/materials-and-software-sharing#loc-sharing-code and ensure that your code is shared in a way that follows best practice and facilitates reproducibility and reuse. 3. We suggest you thoroughly copyedit your manuscript for language usage, spelling, and grammar. If you do not know anyone who can help you do this, you may wish to consider employing a professional scientific editing service.  The American Journal Experts (AJE) (https://www.aje.com/) is one such service that has extensive experience helping authors meet PLOS guidelines and can provide language editing, translation, manuscript formatting, and figure formatting to ensure your manuscript meets our submission guidelines. Please note that having the manuscript copyedited by AJE or any other editing services does not guarantee selection for peer review or acceptance for publication.  Upon resubmission, please provide the following: The name of the colleague or the details of the professional service that edited your manuscript A copy of your manuscript showing your changes by either highlighting them or using track changes (uploaded as a *supporting information* file) A clean copy of the edited manuscript (uploaded as the new *manuscript* file) 4. Thank you for stating in your Funding Statement:  This work is supported by the Open Fund Project of State Key Laboratory of Structural Analysis, Optimization and CAE Software for Industrial Equipment�grant number GZ2024A03-ZZU��the National Key Laboratory of Land and Air Based Information Perception and Control, China (grant number B324009); Science and Technology Research Project of Henan Province(grant number 242102241055); the Industry-University-Research Collaborative Innovation Base Project on Au-tomobile Lightweight of “Science and Technology Innovation in Central Plains”(grant number 2024KCZY315);The Project Supported by National Natural Science Foundation of China (grant number 52302408); the National Natural Science Foundation of China (grant number 52202437) Please provide an amended statement that declares *all* the funding or sources of support (whether external or internal to your organization) received during this study, as detailed online in our guide for authors at http://journals.plos.org/plosone/s/submit-now.  Please also include the statement “There was no additional external funding received for this study.” in your updated Funding Statement. Please include your amended Funding Statement within your cover letter. We will change the online submission form on your behalf. 5. We are unable to open your Supporting Information file “S1 File. Finite element model of primary battery box.fem, S2 File. New battery box finite element model.fem and S3 File. Finite element model of the new battery box upper cover.fem”. Please kindly revise as necessary and re-upload. 6. PLOS requires an ORCID iD for the corresponding author in Editorial Manager on papers submitted after December 6th, 2016. Please ensure that you have an ORCID iD and that it is validated in Editorial Manager. To do this, go to ‘Update my Information’ (in the upper left-hand corner of the main menu), and click on the Fetch/Validate link next to the ORCID field. This will take you to the ORCID site and allow you to create a new iD or authenticate a pre-existing iD in Editorial Manager. 7. Please amend either the abstract on the online submission form (via Edit Submission) or the abstract in the manuscript so that they are identical.

Reviewers' comments:

Reviewer's Responses to Questions

**Comments to the Author**

1. Is the manuscript technically sound, and do the data support the conclusions?

Reviewer #1: Yes

Reviewer #2: Yes

2. Has the statistical analysis been performed appropriately and rigorously?

Reviewer #1: Yes

Reviewer #2: Yes

3. Have the authors made all data underlying the findings in their manuscript fully available?

Reviewer #1: Yes

Reviewer #2: Yes

4. Is the manuscript presented in an intelligible fashion and written in standard English?

Reviewer #1: Yes

Reviewer #2: Yes

**Reviewer #1:**  The reviewed manuscript entitled “Carbon fiber reinforced composite material layup design optimization method and its application in automobile battery box” investigates lightweight design and performance optimization of battery boxes using carbon fiber composites. The article is scientifically and technically sound, with clear practical significance. However, several major concerns should be addressed to enhance readability and clarity:

1- The introduction provides good context but needs a clearer distinction of how this work differs from previous studies.

2- The methodology lacks comparisons with existing optimization algorithms, making it difficult to assess the PSO-BFO algorithm's effectiveness.

3- The finite element modeling section should include a mesh sensitivity analysis for validation.

4- Sources of the 3.67% error in modal frequencies are not discussed and should be clarified.

5- Figures and tables need annotations to highlight key observations clearly.

6- The scalability of the proposed method to other automotive components is not addressed.

7- Statistical validation or repeatability of the experimental results is lacking and should be included.

8- The literature review could be expanded to include recent advancements in digital twin technology.

9- Intermediate optimization results, such as fitness evolution, are not shown and should be added for better visualization.

10- The structural design section could benefit from discussing the durability of the materials in harsh conditions.

11- Details on the choice of excitation point in the experimental setup need clarification.

**Reviewer #2:**  1- The paper presents improvement rates in battery box performance metrics, but no confidence intervals, statistical significance tests, or sensitivity analysis are provided. Given the computational nature of the study, variance in optimization outcomes should be reported.

2- The study primarily relies on finite element simulations to evaluate the optimized battery box's structural behavior. However, experimental validation is missing. The absence of load tests, vibration tests, or impact assessments makes it difficult to confirm the claimed improvements.

3- The influence of different fiber orientations in the composite layup is not systematically investigated. The PSO-BFO hybrid algorithm optimizes layup sequences, but the paper does not analyze how small deviations in fiber angles (±5° or ±10°) affect stress distribution, stiffness, or failure modes.

4- The crashworthiness of the optimized battery box is not discussed. Since the battery box is a critical safety component, it should be evaluated under impact scenarios in addition to static and modal analyses.

5- The transition between the SMC composite upper panel and the carbon fiber composite lower panel could lead to localized stress concentration. The paper lacks stress contour plots highlighting these transition regions and does not mention how delamination or interface debonding risks were mitigated.

6- The study does not compare the optimized hybrid composite battery box with other alternative lightweight designs to contextualize the improvements.

7- The paper does not include a convergence plot for the hybrid PSO-BFO algorithm. Given that hybrid swarm-based optimizers may experience premature convergence, it is essential to demonstrate whether the objective function stabilizes and how many iterations were required.

8- While the PSO-BFO hybrid approach is novel, the paper does not compare its performance with other optimization algorithms. Benchmarking against existing approaches would strengthen the validity of its effectiveness.

9- The study does not assess how variations in material properties (e.g., fiber volume fraction, matrix stiffness, interlaminar shear strength) influence the optimization results. Given manufacturing inconsistencies, this should be addressed.

10- The shear properties of the optimized composite layup are not analyzed. Battery boxes may be subjected to out-of-plane shear loads, which could lead to delamination. A short-beam shear test or interlaminar tension test should be conducted.

11- The paper does not discuss the practical challenges in manufacturing the hybrid composite battery box. Issues like tooling cost, layup reproducibility, curing cycle optimization, and adhesive bonding techniques should be addressed.

12- Given the scope and content of this paper, it may benefit from considering the following related works:

https://doi.org/10.1007/s00521-024-09494-4

https://doi.org/10.1016/B978-0-443-13191-2.00015-8

**Do you want your identity to be public for this peer review?** For information about this choice, including consent withdrawal, please see our Privacy Policy

Reviewer #1: No

Reviewer #2: No

---

## [Author Response · Author response to Decision Letter 1]

21 Mar 2025

Dear Editors and Reviewers.

Regarding our response to your review comments, we have uploaded the file “Response to Reviewers”, which contains the comments from reviewer 1 and reviewer 2, and this contains a detailed response to each reviewer, which contains the images and tables needed to respond to the reviewers' comments. Since only text information can be uploaded on this page, and not tables and images, please review the “Response to Reviewers” file, which contains our detailed responses to the reviewers' comments. Thank you again for your valuable comments, which have helped us a lot in our research work. Thank you.

Reviewer #1:

1- The introduction provides good context but needs a clearer distinction of how this work differs from previous studies.

By combing through the above literature, the researchers have studied and analyzed the performance of automotive battery box assemblies from different angles, including structural optimization, high-strength and lightweight materials, and optimization algorithms, but have ignored manufacturing costs and universality. The lightweight design of the automotive bat-tery box assembly requires comprehensive consideration of factors such as safety, structural optimization, material application and cost control.

In terms of safety, the design and manufacture of the automotive battery box assembly requires comprehensive consideration of multiple factors to ensure the safe driving of the vehicle under various operating conditions. In terms of structural optimization, factors such as the ply angle, shape, number of layers and sequence of carbon fiber composite materials have an important influence on structural performance and strength. At the same time, facing the problem of multi-variable optimization, the convergence and optimization efficiency of the optimization algorithm need to be improved, and the quantitative ranking of the optimized compromise solutions will help to comprehensively balance the performance requirements of all parties. In terms of cost control, it is difficult for an automotive battery box assembly made of a single lightweight material to have both performance and cost advantages. In response to this situation, this paper proposes a hybrid particle swarm optimization and bacterial foraging optimization strategy (PSO-BFO) based on mathematical twin technology, which can signif-icantly improve the lightweight optimization effect of automotive carbon fiber composite battery boxes and achieve an integrated lightweight design of structure, materials and per-formance. However, no relevant methods have been applied to the lightweight design of hy-brid material automotive battery box assemblies.

Therefore, this paper proposes a novel optimization design method for composite battery cases to address the above issues. Based on structural optimization, material substitution and experimental testing, Digital Twinning technology is used in combination with a hybrid par-ticle swarm optimization and bacterial foraging optimization strategy to improve the global optimization capability and convergence speed of composite material battery cases in the lightweight design process. A real-time simulation and monitoring system is used to provide more physical feedback during the PSO-BFO optimization process provides more physical feedback while jointly optimizing key indicators such as strength, stiffness, and manufac-turing cost under multi-objective comprehensive constraints, ensuring that in the pursuit of Lightweight, structural safety and material performance are not compromised, and that an integrated lightweight design of the structure, materials, and performance of the carbon fiber hybrid material battery box can be achieved. This provides an entirely new solution for ad-vanced lightweight design with broad application prospects.

2- The methodology lacks comparisons with existing optimization algorithms, making it difficult to assess the PSO-BFO algorithm's effectiveness.

In order to verify the effectiveness of the hybrid optimization algorithm proposed in this paper, under the same conditions, a single optimization algorithm and a hybrid optimization algorithm are used to optimize the ply design of the lower box of the battery box, and the results are compared and analyzed. Table 1 is a comparison of the results of using different optimization algorithms. From the results, it can be seen that different optimization methods can improve the performance of the battery box and reduce its quality to a certain extent, but using a single optimization algorithm is prone to fall into a local optimal solution. In contrast, the hybrid optimization algorithm can produce more perfect results. The proposed PSO-BFO algorithm improves the mass M of the battery box assembly, the first-order modal frequency of the battery box, the stress under bumpy conditions, and the displacement under bumpy conditions by 40.63%, 33.34%, 83.60%, and 62.80%, respectively. The PSO-GA algorithm improves the mass M of the battery box assembly, the first-order modal frequency of the battery box, the stress under bumpy conditions, and the displacement under bumpy conditions by 33.50%, 29.31%, 83.04%, and 57.87%, respectively. Although both the proposed PSO-BFO algorithm and PSO-GA algorithm can obtain the global optimal solution of complex optimization problems, the results obtained by the PSO-BFO algorithm are more balanced than those of the PSO-GA algorithm. Therefore, the PSO-BFO algorithm proposed in this paper can effectively solve the multi-objective optimization problem of the ply layup sequence of the lower box of the carbon fiber composite battery box

3- The finite element modeling section should include a mesh sensitivity analysis for validation.

According to the relevant national standard GB/T 33582-2017 “General Rules for Finite Element Mechanical Analysis of Mechanical Product Structures”, the mesh quality of the finite element model of the battery box assembly established in this paper was checked, and the results are shown in Table 1. It can be seen from Table 2 that the mesh quality of the battery box finite element model established in this paper meets the quality standards specified by the regulations. Therefore, the established battery box assembly finite element model can accurately model the actual situation of the battery box assembly.

4- Sources of the 3.67% error in modal frequencies are not discussed and should be clarified.

Thank you very much for your suggestion. We are deeply sorry for our carelessness. We have made additional explanations in the original text, and the additional explanations have been marked in red text.

The maximum error between the simulated modal frequency and the experimental modal frequency is 3.67%. The reason for this may be that there is a deviation between the true coordinate angle of the triaxial accelerometer and the modeling angle, which leads to the maximum error of 3.67% between the simulated modal frequency and the experimental modal frequency. We have already made additional explanations in the article.

5- Figures and tables need annotations to highlight key observations clearly.

Dear reviewer, thank you for your valuable comments! We have optimized the chart captions in the paper to highlight the key observations more clearly. For each chart, we have added more detailed descriptions to more intuitively present the trends and conclusions reflected by the data. In addition, we have further improved the interpretation of key data points so that readers can more easily understand the information conveyed by the charts and strengthen their connection with the core content of the paper. I hope these modifications can improve the readability and accuracy of the paper. Thank you again for your review and suggestions, and look forward to your further feedback. For example, Figures 5, 38, 39, 40, 41, 44, 46, and Table 12. We have provided additional explanations in the manuscript to make it easier to see the key results.

6- The scalability of the proposed method to other automotive components is not addressed.

Regarding the question you raised about applying the method in this article to other automotive parts, other researchers in our research group have successfully applied this method to the front floor and B-pillar of the car, and have carried out physical verification. The relevant research work will be released in the near future.

7- Statistical validation or repeatability of the experimental results is lacking and should be included.

Thank you very much for your suggestion. We are deeply sorry for the inaccurate expression. We have made additional clarifications in the original text, and the additional clarifications have been marked in red text in the manuscript.During the experiment, we conducted multiple groups of experimental controls for each test to ensure the credibility of the test results.

Among them: In the free modal test, the acquisition of modal vibration signal is completed through multiple point measurements. At the same time, in order to reduce the test error and ensure the test accuracy, 20 groups of tests are carried out each time, and then the pathological samples are removed and the average value is taken as the result of each measurement.

In the process of obtaining carbon fiber composite material parameters, we use 6 groups of repeated tests for each test and take the average value of the test results to ensure the credibility of the test results obtained.

8- The literature review could be expanded to include recent advancements in digital twin technology.

Thank you for your suggestions, which are crucial to enriching our work. We have added the latest articles on the progress of digital twin technology to the text, and the added content has been marked with red text.

9- Intermediate optimization results, such as fitness evolution, are not shown and should be added for better visualization.

After repeated tests, the PSO-BFO hybrid algorithm can effectively avoid falling into the local optimum and accelerate the convergence speed of particles by combining the global and local search capabilities of the BFO algorithm and the PSO algorithm. Its convergence effect is better than that of the BFO algorithm and the PSO algorithm. The convergence effect test is shown in Figure 1.

The algorithm iteration process is shown in Figure 1. The PSO-BFO algorithm has a fast convergence speed and high precision. After reaching convergence, the PSO-BFO curve shows slight fluctuations, indicating that it still has a certain global search capability, which helps to refine the optimal solution. The PSO algorithm converges faster, but slightly slower than the PSO-BFO algorithm. The final fitness value is higher than the PSO-BFO algorithm, indicating that its optimization effect is inferior to the PSO-BFO algorithm, but it still has good global optimization capabilities. The BFO algorithm has the slowest convergence speed and the highest final fitness value, indicating that under the same computing conditions, its search efficiency and accuracy are not as good as the other two algorithms.

10- The structural design section could benefit from discussing the durability of the materials in harsh conditions.

Thank you for your suggestions, which are of great importance to our work. Due to the length of the paper and the limitations of the research, this paper does not study the durability of materials under harsh conditions, so we have added relevant literature in the introduction to understand the research content in this area. We have updated these contents in the introduction, and the additions are marked in red text. Thank you very much for your valuable comments and suggestions, which have played a positive role in promoting the improvement of our research. If there are further revisions, we are also very happy to make corresponding adjustments.

11- Details on the choice of excitation point in the experimental setup need clarification.

In the battery box modal test, the optimal excitation point is the one that can maximize the excitation force to excite the various modes of the battery box. The selection criteria for the excitation point are as follows:

1. The excitation point should be far away from the nodes or nodal lines of each mode to avoid mode loss due to insufficient energy input. If the excitation point is located at the node of a certain mode, the mode will not be effectively excited.

2. Choose a place with high rigidity and convenient for excitation. In addition, it is also necessary to consider whether the exciter is easy to install at the selected excitation point. Otherwise, it will not be able to effectively excite and other candidate points need to be re-evaluated. Therefore, the middle edge of the bottom of the battery box is selected as the excitation point in this test.

Reviewer #2:

1- The paper presents improvement rates in battery box performance metrics, but no confidence intervals, statistical significance tests, or sensitivity analysis are provided. Given the computational nature of the study, variance in optimization outcomes should be reported.

Dear reviewer, thank you for your review and valuable comments on our paper. Our research is based on the optimization design of the test and simulation model of the original battery box, so the credibility of the optimization results is mainly supported by the consistency of simulation and test data. Before optimization, the maximum error rate between the free modal test frequency and the simulated modal frequency of the original battery box was 3.67%, indicating that the simulation model reflects the actual situation well. The optimized battery box uses SMC upper cover, and the maximum error between its free modal simulation analysis and modal test frequency is 3.15%, which further proves the reliability of the simulation method. Therefore, although the confidence interval is not provided in the study, the high consistency between simulation and experiment can serve as a support for the reliability of the optimization results.

Given the computational nature of the study, the change in the modal frequency of the battery box upper cover before and after optimization can intuitively reflect the effectiveness of the optimization design. The first-order modal frequency increased from 20.98 Hz to 35.82 Hz, and the performance was improved by 70.73%. The comparison results before and after optimization show that the stiffness of the battery box upper cover has been significantly enhanced, and the optimization scheme has practical value in engineering applications.

Regarding sensitivity analysis, the reliability of the optimization scheme mainly depends on the high consistency between simulation and test data, which is verified by the significant improvement of modal frequency before and after optimization. Since the optimization design is based on a simulation model that has been verified by experiments, and the optimization results show obvious performance improvement, the optimization trend and improvement are sufficient to prove the effectiveness of the optimization design, and the study did not conduct further sensitivity analysis. Thank you for your review and valuable suggestions, we look forward to your further feedback!

2- The study primarily relies on finite element simulations to evaluate the optimized battery box's structural behavior. However, experimental validation is missing. The absence of load tests, vibration tests, or impact assessments makes it difficult to confirm the claimed improvements.

Thank you very much for your suggestions, which are of vital importance to our work. We have included the relevant trials in the article, and the additions are marked in red text. When we submitted the manuscript, the finished SMC composite material upper cover was still in production. Now we have completed the production of the finished SMC composite material upper cover. We supplemented the free modal test of the battery box SMC upper cover and compared it with the experimental modal frequency to verify the correctness of its structure. Due to time constraints, the finished carbon fiber composite lower box has not yet been completed. We will supplement this aspect

---

## [Decision Letter · Decision Letter 1]

Carbon fiber reinforced composite material layup design optimization method and its application in automobile battery box

PONE-D-25-01443R1

Dear Dr. Zhang,

We’re pleased to inform you that your manuscript has been judged scientifically suitable for publication and will be formally accepted for publication once it meets all outstanding technical requirements.

Kind regards,

Khalil Abdelrazek Khalil, Ph.D.

Academic Editor

PLOS ONE

Additional Editor Comments (optional):

Reviewers' comments:

Reviewer's Responses to Questions

**Comments to the Author**

Reviewer #1: All comments have been addressed

Reviewer #2: (No Response)

2. Is the manuscript technically sound, and do the data support the conclusions?

Reviewer #1: Yes

Reviewer #2: (No Response)

3. Has the statistical analysis been performed appropriately and rigorously?

Reviewer #1: Yes

Reviewer #2: (No Response)

4. Have the authors made all data underlying the findings in their manuscript fully available?

Reviewer #1: Yes

Reviewer #2: (No Response)

5. Is the manuscript presented in an intelligible fashion and written in standard English?

Reviewer #1: Yes

Reviewer #2: (No Response)

Reviewer #1: The authors have provided amendments to all the suggested queries, therfore, I recommend this work for publication in PLOS One

Reviewer #2: After thoroughly reviewing the revised manuscript and the authors' detailed responses to the initial comments, the improvements made are satisfactory. All concerns have been adequately addressed, and the manuscript now meets the standards for publication. Acceptance of this article is recommended.

**Do you want your identity to be public for this peer review?** For information about this choice, including consent withdrawal, please see our Privacy Policy

Reviewer #1: No

Reviewer #2: No

---

## [Editor Report · Acceptance letter]

PONE-D-25-01443R1

PLOS ONE

Dear Dr. Zhang,

I'm pleased to inform you that your manuscript has been deemed suitable for publication in PLOS ONE. Congratulations! Your manuscript is now being handed over to our production team.

Kind regards,

on behalf of

Dr. Khalil Abdelrazek Khalil

Academic Editor

PLOS ONE